# Task-Relevant Feature Selection with Prediction Focused Mixture Models

**Abhishek Sharma**                                    *abhisheksharma@g.harvard.edu*
*School of Engineering and Applied Sciences,*
*Harvard University*

**Catherine Zeng**                                    *catherinezeng@college.harvard.edu*
*School of Engineering and Applied Sciences,*
*Harvard University*

**Sanjana Narayanan**                                    *sanjana.books@gmail.com*
*School of Engineering and Applied Sciences,*
*Harvard University*

**Sonali Parbhoo**                                    *s.parbhoo@imperial.ac.uk*
*Imperial College London*

**Roy Perlis**                                    *rperlis@mgh.harvard.edu*
*Massachusetts General Hospital*

**Finale Doshi-Velez**                                    *finale@seas.harvard.edu*
*School of Engineering and Applied Sciences,*
*Harvard University*

**Reviewed on OpenReview:** *https://openreview.net/forum?id=voHKJOdCNw*

## Abstract

Probabilistic models, such as mixture models, can encode latent structures that both explain the data and aid specific downstream tasks. We focus on a constrained setting where we want to learn a model with relatively few components (e.g. for interpretability). Simultaneously, we ensure that the components are useful for downstream predictions by introducing *prediction-focused* modeling for mixtures, which automatically selects data features relevant to a prediction task. Our approach identifies task-relevant input features, outperforms models that are not prediction-focused, and is easy to optimize; most importantly, we also characterize *when* prediction-focused modeling can be expected to work.

## 1 Introduction

The subfield of task-focused generative modeling uses tasks to identify *relevant* structure embedded in complex data. For example, suppose we want to discover HIV subtypes (clusters) from electronic health records. Health records are complex and contain structures irrelevant to our goal (e.g. effects of insurance type). A model small enough for us to inspect for medical insights (that is, one where we underspecify the number of clusters), risks finding irrelevant patterns, while a model that is too large may be too complex to inspect or learn efficiently.

Task-focused approaches to this problem introduce a *downstream prediction task* to separate relevant structure from irrelevant. For example, if the clusters can predict whether a patient will develop a specific mutation, then it is likely those clusters are relevant to identifying HIV subtypes. This motivation results in our desiderata: our goal is to train a mixture model that is (a) small enough for human inspection, and

(b) predictive for some task. Because inspection is our ultimate goal, simply creating a predictive model is not sufficient. Additionally, focusing only on relevant structures also facilitates sample-efficient learning.

Unfortunately, task-focused generative modeling is not easy: identifying clusters relevant to our task is the challenge. As we detail in Section 6, the naive approach of augmenting a mixture model with labels (supervised generative modeling) fails because the signal from the input features overwhelms the signal from the labels. More principled approaches involve challenging optimization (Hughes et al., 2018b). Works like Ren et al. (2020) achieve more tractable optimization by identifying a graphical model whose maximum likelihood can be used as the solution. However, they do not address important theoretical questions, like *if* and *when* their objective makes the desired trade-off between predictive and generative performance.

In this work, we introduce and analyze prediction-focused Gaussian mixture models (pf-GMM) and hidden Markov models (pf-HMM). All of our models identify relevant structure by automatically curating input features as relevant or irrelevant. This curation enables us to find predictive clusters even when the model is misspecified. Unlike the prediction-constrained approach by Hughes et al. (2018b), we have formulated our problem as a graphical model with a single hyperparameter (the probability that an input feature is relevant). Unlike Ren et al. (2020), we provide a precise theoretical analysis identifying data regimes in which this approach will work.

Specifically, we make the following contributions:

1. We develop a probabilistic graphical modeling approach (pf-GMM, pf-HMM) that automatically curates and clusters features relevant to a downstream task. We outperform standard model-based clustering approaches like Gaussian Mixture Models (with and without supervision) at predicting the label and can use existing inference algorithms e.g. Expectation Maximization (EM).

2. We analytically characterize representative scenarios where our auto-curation model achieves the desired relevant structure, and support this analysis with empirical evidence using simulations and real datasets. Unlike Ren et al. (2020), we characterize the true likelihood objective, and not its lower bound—making our analysis applicable irrespective of approximation in inference. In doing so, we show in which data regimes this relatively simple and robust approach will work.

3. We demonstrate that pf-GMM achieves predictive clusters in even in misspecified-cluster settings on several synthetic and real-world datasets. We show that our approach not only yields computational benefits but is also easier to optimize than the prediction-constrained approach (Hughes et al., 2018b). pf-GMMs thus provide the best of both worlds—they achieve a good predictive-generative trade-off and are easy to tune and optimize.

**Outline**

The rest of the paper is organized as follows. In Section 2, we review the literature on semi-supervised learning and task-focused generative modeling. In Section 3, we describe the data setting and the assumptions we make about the data. In Sections 4 and 5, we introduce our prediction-focused models and describe how to perform inference on them. Then in Section 6, we analyze the properties of prediction-focused mixture models and characterize the data regimes in which they will work. In Sections 7 and 8, we present our experiments and results. Finally, in Section 9, we discuss the implications of our work.

## 2 Related Work

**Semi-supervised Learning** Semi-supervised Learning augments the generative modeling of inputs $\mathbf{X}$ with a supervisory target $Y$. Methods that incorporate supervisory labels under the paradigm of "supervised clustering" fail to provide a density of the observed data (Eick et al., 2004) and are not aligned with our goals. For methods that train the joint likelihood $\log f(\mathbf{X}, Y)$ (Nigam et al., 1998; Kingma et al., 2014; Ghahramani & Jordan, 1994; Fraley & Raftery, 2002), the generative process is typically modeled with latent variables $\mathbf{Z}$ as $\mathbf{X} \leftarrow \mathbf{Z} \rightarrow Y$. However, these models fail to recognize the inherent asymmetry between the dimensionalities of $\mathbf{X}$ and $Y$ and tend to ignore $Y$ in favor of allocating the model's capacity for the

much more structured $\mathbf{X}$. We address this issue by incorporating specific input-focused latent variables (called 'switches') that treat the inputs and targets differently.

**Task-focused Generative Modeling**  Most work on learning generative models for specific downstream tasks (Lacoste–Julien et al., 2011; Cobb et al., 2018; Stoyanov et al., 2011; Futoma et al., 2020) train for the discriminative loss only. Closer to our goal of managing the generative-discriminative trade-off, Hughes et al. (2017) proposed prediction-constrained learning in the context of mixture and topic models, formalizing the problem as maximizing the likelihood of $\mathbf{X}$ constrained by the prediction objective. However, this objective is hard to optimize in practice, and unlike our work, does not correspond to the maximum likelihood of a valid graphical model.

**Downstream use cases of Generative Models**  Semi-supervised generative models that use a discriminative task to learn more useful generative representations have found several applications in the literature. The review by (Chen et al., 2021) highlights the importance of generative models for understanding the differences in survival time of patients with melanoma with different demographics for aiding in better planning of disease management. Halpern et al. (2016) describe the importance of generative models for identifying groups of clinical conditions that can subsequently used to predict disease subtypes and clinical tags. Works such as (Prabhakaran & Vogt, 2019) and (Prabhakaran et al., 2012) also show how generative models can be used to learn clusters of similar chemical compounds or proteins in HIV treatment which can subsequently be used to predict response patterns for patients who receive those treatments. Furthermore, mixture models have been used in several classes of downstream tasks. For example, Sharma et al. (2023) use the density $p(X)$ from a mixture model to plan an elevator scheduling decision-making task, Silva & Deutsch (2018) use the gaussian mixture model to impute missing values in a dataset of lateritic Nickel deposits, and Attar et al. (2014) use gaussian mixture models for anomaly detection. Besides these, the interpretation of clusters for exploration and science-oriented tasks is a common use case of the mixture models (Kuyuk et al., 2012; Zhuang et al., 1996). Our work supports these downstream efforts by providing a way to build a mixture model by focusing on the relevant features.

**Switch variables**  In the context of topic models for documents, Chemudugunta et al. (2007) and Ren et al. (2020) use switch variables to include or exclude words based on their predictive relevance. Both approaches rely on independence assumptions between switches and topic-word distributions in the approximate posterior. Furthermore, the authors only study a lower bound to the likelihood—making it unclear whether their results are due to the choice of model or inference. In contrast, we develop a generative-discriminative model for a more general context. More importantly, we show that the prediction-focused properties exist by our model structure alone (independent of the choice of inference), and (unlike previous work) we describe analytically when our approach will work. Finally, some works impose structural constraints on the target variables and covariates of Conditional Random Fields (CRFs) (Wainwright & Jordan, 2008), such as (Jiao et al., 2006; Chen et al., 2015; Zheng et al., 2015). Unlike these works, we focus on the setting where a model does not have enough parameters to fully model the input dimensions.

**Feature selection/attribution methods**  Feature selection (FS) methods pick out a small subset of features from the original feature set, and run subsequent analysis on this small subset of features (Jović et al., 2015). This feature selection can be unsupervised or supervised. Among unsupervised FS methods, clustering FS methods remove the irrelevant features without considering the prediction task (Witten & Tibshirani; Dash & Liu, 2000; Jović et al., 2015). However, their notion of relevance does not consider the alignment of the clusters to a supervisory signal available in the form of labels. This can lead to a clustering of features that are irrelevant to the labels because the irrelevant features can be correlated to each other. In contrast, we utilize the provided labels to cluster the relevant features. This leads to clusters that are predictive of the labels.

Supervised feature selection and attribution methods directly identify the features that are useful for a prediction task (e.g., Random Forest classification, LASSO classification, correlation coefficients) by removing irrelevant features or by providing importance scores. These scores are also an interpretability tool since they help the user understand which features were useful for the task. However, these methods do not directly provide a generative model (i.e. clustering) of the relevant features of the data. In addition, these methods

do not prioritize finding all features that are relevant to the outcome; as a result, the structures they find may be incomplete and thus confusing for domain experts to interpret. In contrast, our prediction-focused modeling provides a model-based clustering using the relevant features *and* the relative importance scores of these features in the joint task of clustering and prediction.

# 3 Data Setting

In this section, we formally describe the data distribution that we assume in the analysis-related sections of our paper. This is done to emphasize that this distribution is much more complex than the model distribution we introduce in Section 4. That said, we emphasize that our prediction-focused model is specifically designed to work well in 'misspecified' settings where the model has fewer components than the true data generating process, regardless of the specific data generating process.

## 3.1 Preliminaries

We denote random variables by capital letters (e.g. $X, Y$), (column) vectors by bold-face letters (e.g. $\mathbf{u}$), and random vectors by bold-faced capital letters (e.g. $\mathbf{X}$). We denote an identity matrix of size $D \times D$ as $I_D$. We use $[\mathbf{u}; \mathbf{v}]$ to denote the concatenation of vectors $\mathbf{u}$ and $\mathbf{v}$ along the last dimension. We use $\mathbf{0}_D$, and $\mathbf{1}_D$ to denote constant vectors of length $D$ with values 0 and 1 respectively. In our discussions, we use the terms 'feature' and 'dimension' interchangeably. The true data-generated densities and parameters are $f^*, \Theta^*$, whereas the estimated quantities are denoted by $f, \Theta$. We use $\phi(\cdot; \Theta)$ to denote the multivariate normal density with parameters $\Theta = \{\mu^\Theta, \Sigma^\Theta\}$.

## 3.2 Assumptions

We assume that we are given the dataset $\mathcal{D} = \{(\mathbf{X}^{(n)}, Y^{(n)})\}_{n=1}^N$, where the input $\mathbf{X}^{(n)} \in \mathbb{R}^D$ and the target $Y^{(n)} \in \{0, 1\}$ when $\mathbf{X}^{(n)}$ and $Y^{(n)}$ are not time-series. In the time-series setting, the input $\mathbf{X}^{(n)} \in \mathbb{R}^{T \times D}$ and the target $Y^{(n)} \in \{0, 1\}^T$ are sequences of length $T$.

We consider the setting where the data is a concatenation of multiple sources with multiple *irrelevant* dimensions each to one source that contains *relevant* dimensions—and *each* source is distributed as an independent Gaussian Mixture Model. The *relevant* mixture is called so because it also generates $Y$. Specifically, we assume that the input $\mathbf{X}^{(n)}$ is a concatenation of $M$ sources:

$$\mathbf{X}^{(n)} = [\mathbf{X}_1^{(n)}; \ldots; \mathbf{X}_M^{(n)}] \quad \forall n \in \{1, \ldots, N\} \tag{1}$$

$$\text{where } \mathbf{X}_m^{(n)} \in \mathbb{R}^{D_m} \quad \forall m \in \{1, \ldots, M\} \text{ and } \sum_{m=1}^M D_m = D. \tag{2}$$

We also assume that there are much fewer relevant dimensions than irrelevant dimensions. This assumption is the main source of the challenge since we only care about the relevant dimensions $\mathbf{X}_1^{(n)}$ and the target $Y^{(n)}$ when explaining the input $\mathbf{X}^{(n)}$ and predicting the target $Y^{(n)}$.

## 3.3 Data distribution

We describe the data-generating process for a single data point $(\mathbf{X}, Y)$ in the non-time-series setting (the description for the time-series setting appears in Supplement Section 3.4). As stated above in the assumptions, the ground truth data density $f^*$ is a product of the relevant source $f_1^*$ and each of the irrelevant sources $f_m^*$ (for $m \geq 2$). The data density is (see Figure 1 for a graphical model depicting this process):

$$f^*(\mathbf{X}, Y; \Theta^*) = f_1^*(\mathbf{X}_1, Y; \Theta_1^*) \prod_{m=2}^M f_m^*(\mathbf{X}_m; \Theta_m^*) \tag{3}$$

where $\Theta^* = \{\Theta_1^*, \ldots, \Theta_M^*\}$ are the parameters of the mixture components. Mixture $m$ has $K_m$ components.

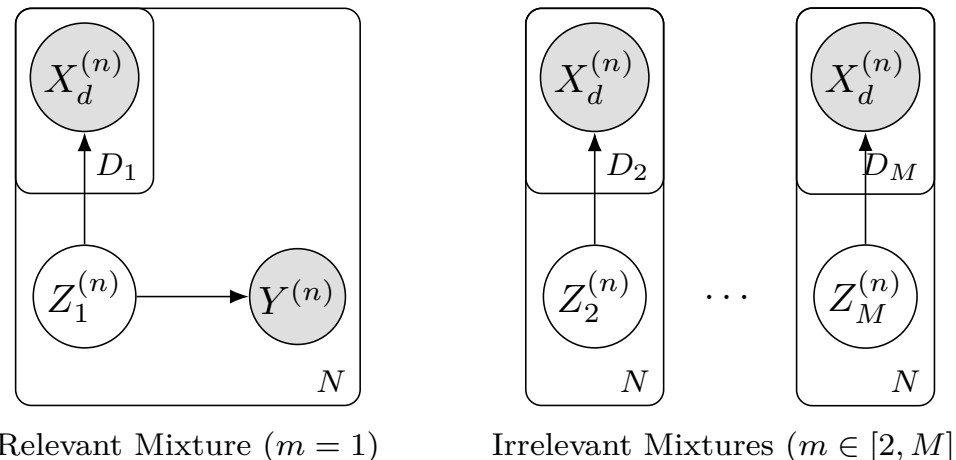

Figure 1: Data generating process concatenes $M$ mixture models. The first mixture model also generates the target variable $Y$ (and hence relevant), while the remaining $M-1$ mixture models are irrelevant.

The first density in the above equation $f_1^*(\mathbf{X}_1, Y; \Theta_1^*)$ is special because correlated with $Y$:

$$f_1^*(\mathbf{X}_1, Y; \Theta_1^*) = \sum_{k=1}^{K_1} \pi_{1,k}^* f_{1,k}^*(\mathbf{X}_1; \beta_{1,k}^*) f_{Y,k}^*(Y; \eta_k^*) \tag{4}$$

where $\pi_{1,k}^*$ are the mixture weights, $\beta_{1,k}^*$ are the component's input data parameters, and $\eta_k^*$ are the component's target parameters.

Every subsequent density $f_m^*(\mathbf{X}_m; \Theta_m^*)$ ($m \geq 2$) is independent of $Y$ in Equation 3:

$$f_m^*(\mathbf{X}_m; \Theta_m^*) = \sum_{k=1}^{K_m} \pi_{m,k}^* f_{m,k}^*(\mathbf{X}_m; \beta_{m,k}^*) \quad \text{for } m \geq 2 \tag{5}$$

where $\pi_{m,k}^*$ and $\beta_{m,k}^*$ are similarly defined as above.

For each mixture density $f_m^*(\cdot)$, the mixture weights sum to one:

$$1 = \sum_k \pi_{m,k}^* \quad \forall m. \tag{6}$$

Note that we consider the case of Gaussian emissions, i.e. $f_{m,k}^*(\cdot; \beta_{m,k}^*) = \phi(\cdot; \beta_{m,k}^*)$ of dimension $D_m$, though our approach applies to any exponential-family distribution. We assume $f_{Y,k}^*(\cdot; \eta_k^*)$ to be the Bernoulli probability mass function, but again our approach extends to any exponential-family distribution.

**Example: HIV subtyping.** As an application of this data-generating process, consider the HIV subtyping example from the introduction (Section 1). The goal of the HIV subtyping problem is to identify data-driven HIV subtypes in the form of patient clusters sharing similar attributes. The data for this problem is the patient's electronic health records, which include several sources of information such as blood count information, genomics, and hospital visit codes during the patient's lifetime. Among these, the relevant features ($\mathbf{X}_1$) would include measurements of immune response and of co-infections expected to be correlated with the mutations ($Y$). The irrelevant features correspond to groups of diagnosis and medicine codes that are unrelated to the patient's HIV condition ($\mathbf{X}_m, m \geq 2$). For instance, a patient's depression diagnosis, along with related medical codes, is irrelevant to the HIV subtyping problem. The relevant features and each group of irrelevant features can be thought of as being distributed as a mixture with $K_m$ components since it is reasonable to expect that each condition (e.g. depression) has multiple subtypes. While it is challenging to reconstruct all of the structure in the data as a mixture with $K_1 \times \prod_{m \geq 2} K_m$ components, our primary focus, the discovery of HIV subtypes, does not necessitate perfect modeling of these irrelevant features.

### 3.4 Data distribution for the time-series case

In this subsection, we describe the case where a single input-target data pair $(\mathbf{X}, \mathbf{Y})$ is a sequence of length $T$, i.e. $\mathbf{X} = [\mathbf{X}_1; \ldots; \mathbf{X}_M] \in \mathbb{R}^{T \times (D_1 + \cdots + D_M)}$ and $\mathbf{Y} \in \{0, 1\}^T$. This case is analogous to the non-time-series setting, except that we assume that the data is generated by $M$ independent Hidden Markov Models (HMMs) instead of $M$ independent Gaussian Mixture Models (GMMs). The data density is:

$$f^*(\mathbf{X}, \mathbf{Y}; \Theta^*) = f_1^*(\mathbf{X}_1, \mathbf{Y}; \Theta_1^*) \prod_{m=2}^{M} f_m^*(\mathbf{X}_m; \Theta_m^*) \tag{7}$$

$$f_1^*(\mathbf{X}_1; \Theta_1^*) = \sum_{k_1=1}^{K_1} \cdots \sum_{k_T=1}^{K_1} \pi_{1,k_1}^* \prod_{t=2}^{T} A_{1,k_{t-1},k_t}^* \prod_{t=1}^{T} f_{1,k_t}^*(\mathbf{X}_{t,1}; \beta_{1,k_t}^*) f_{Y,k_t}^*(\mathbf{Y}_t; \eta_k^*) \tag{8}$$

$$f_m^*(\mathbf{X}_m; \Theta_m^*) = \sum_{k_1=1}^{K_m} \cdots \sum_{k_T=1}^{K_m} \pi_{m,k_1}^* \prod_{t=2}^{T} A_{m,k_{t-1},k_t}^* \prod_{t=1}^{T} f_{m,k_t}^*(\mathbf{X}_{t,m}; \beta_{m,k_t}^*) \tag{9}$$

$$\tag{10}$$

where $\Theta^* = \{\Theta_1^*, \ldots, \Theta_M^*\}$ are the parameters of the HMMs and HMM $m$ has $K_m$ latent states. $\pi_{m,k}^*$ are the initial state probabilities, $A_{m,j,k}^*$ are the transition probabilities:

$$\pi_{m,k}^* = f_{m,k_1}^*(Z_{1,m} = k) \tag{11}$$

$$A_{m,j,k}^* = f_{m,k_t}^*(Z_{t,m} = k | Z_{t-1,m} = j) \quad \forall t \in [2, T] \tag{12}$$

where $Z_{t,m}$ is the latent state of HMM $m$ at time $t$. These probabilities sum to one: $\sum_k \pi_{m,k}^* = 1$, $\sum_k A_{m,j,k}^* = 1$ for all $m, j$. Finally, the emission distributions are parameterized the same way as above: $\beta_{m,k}^*$ are the parameters of the input-emission distribution $f_{m,k}^*(\cdot)$ of HMM $m$'s latent state $k$, and $\eta_k^*$ are the parameters of target-emission $f_{Y,k}^*(\cdot)$ of HMM 1's latent state $k$.

### 3.5 Consequences of our assumptions

**The data may have exponentially many components but we only care about a few.** In the GMM case, $f_m^*(\cdot)$ is the probability density function of a $K_m-$component mixture distribution, and $f^*(\mathbf{X}, Y)$ is the density of a mixture distribution with exponentially many components ($\prod_m K_m$ in total). The HMM case is the same except that the mixture components are replaced by HMM components. However, modeling all the components needed to reconstruct all of the structure in the true data-generating distribution may be unnecessary if we are interested in only the structure relevant for prediction.

**The data-generating process allows for several forms of correlation.** The key consequence of our assumption about the data setting is the form of correlations we allow in relevant and irrelevant features: (a) correlation induced by being distributed as a GMM or an HMM, (b) correlation within each mixture component, and (c) independence of relevant and irrelevant mixtures. Together, these correlations allow the relevant features to have complex structure, while keeping the irrelevant features separable from the relevant ones.

### 3.6 Problem Statement

We want a mixture model (GMM or HMM) that uses a user-specified budget of K ($\ll \prod_m K_m$) components to: (a) predict targets $\mathbf{Y}$ at test time, and (b) maximize the data likelihood $f(\mathbf{X})$ for a given prediction quality of $\mathbf{Y}$. To address these desiderata, we introduce prediction-focused mixture models in the next section. Then in Section 6, we show why a GMM (with or without supervision) does not achieve good predictive performance in this setting, and how our prediction-focused model is preferable.

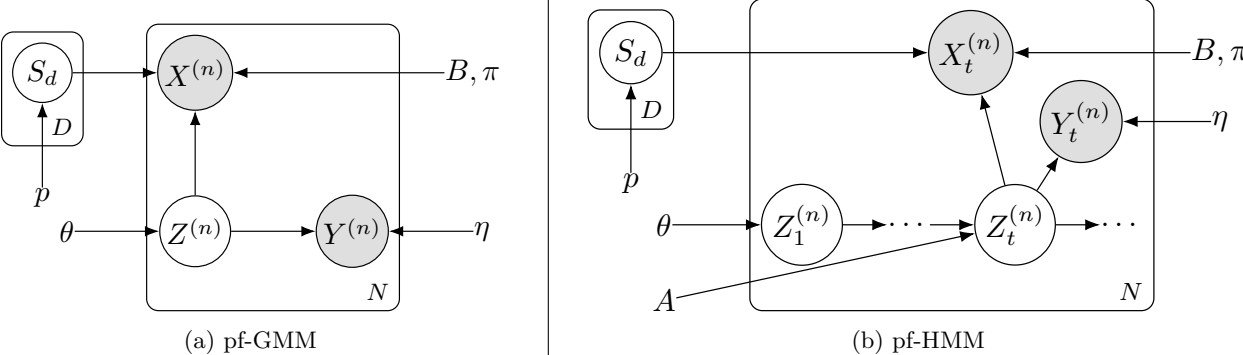

Figure 2: Graphical models for our models pf-GMM (a) and pf-HMM (b). Here, $S_d$ denote the per-dimension switch variables, $X$ denotes the data, $Y$ denotes the target variables. For pf-GMM, $Z$ denotes the cluster assignments, and for pf-HMM, $Z_t$ denotes the hidden states at time $t$.

## 4 Our Models: Prediction-focused Mixtures

In this section, we introduce our prediction-focused mixture models which satisfy the desiderata of maximizing the data likelihood while still performing well on the prediction task. This model is specifically designed to work in the underspecified setting, i.e., when the model is provided much fewer components than the data requires.

Our key assumption is that the input data dimensions can be partitioned as being either 'relevant' or 'irrelevant'. Only 'relevant' dimensions are predictive of the target, $Y$. This assumption matches the data-generating process: we preserve the connection between the relevant dimensions and the target and approximate the irrelevant dimensions with a background distribution. We start by introducing prediction-focused Gaussian Mixture Models (pf-GMM) and then extend them to the time series setting by introducing prediction-focused Hidden Markov Models (pf-HMM) (see Figures 2a and 2b for their graphical models).

### 4.1 Prediction-focused Gaussian Mixture Model (pf-GMM)

Consider the dataset $\mathcal{D} = \{(\mathbf{X}^{(n)}, Y^{(n)})\}_{n=1}^{N}$, where $N$ is the total number of data points, and each input $\mathbf{X}^{(n)} \in \mathbb{R}^D$ has $D$ dimensions. The data-generating process under our model pf-GMM assumes the following way of explaining the dataset $\mathcal{D}$.

First, we draw a 'switch' variable $S_d$ per dimension to indicate whether the dimension is relevant or irrelevant. If $S_d = 1$, then the dimension is relevant, and if $S_d = 0$, then the dimension is irrelevant. We also draw a cluster identity $Z^{(n)}$ per data point to indicate which mixture component generated the data point:

$$S_d \sim \mathbf{Bern}(p) \quad \forall d \in \{1, \ldots, D\}, \qquad Z^{(n)} \sim \mathbf{Cat}(\theta) \quad \forall n \in \{1, \ldots, N\},$$

where $p$ is the prior probability that a dimension is relevant, and $\theta$ is the prior probability of each mixture component.

Next, we draw the input data dimensions $\mathbf{X}_{\mathbf{S}}^{(n)}$ from a mixture distribution with $K$ components, where $\mathbf{S}$ is a binary vector which is 1 for relevant dimensions (i.e. $\mathbf{S} = [S_1; \ldots; S_D]$) and $\mathbf{X}_{\mathbf{S}}^{(n)}$ is the subvector of $\mathbf{X}^{(n)}$ with only the relevant dimensions. We draw the remaining dimensions $\mathbf{X}_{1-\mathbf{S}}^{(n)}$ from a background distribution ($1 - \mathbf{S}$ is the binary vector which is 1 for irrelevant dimensions):

$$Y^{(n)}|Z^{(n)} = j \sim \mathbf{Bern}(\eta_j), \quad \mathbf{X}_{\mathbf{S}}^{(n)}|Z^{(n)} = j \sim \mathcal{N}(\boldsymbol{\mu}_j^B, \Sigma_j^B),$$
$$\mathbf{X}_{1-\mathbf{S}}^{(n)} \sim \mathcal{N}(\boldsymbol{\mu}^\pi, \Sigma^\pi) \quad \forall n \in \{1, \ldots, N\}. \tag{13}$$

Note that we chose specific distributions for $\mathbf{X}$ and $Y$ for the sake of clarity but we could have chosen any exponential family distribution.

The joint distribution for pf-GMM, $f_{\text{pfGMM}}(\mathbf{X}, \mathbf{Y}, \mathbf{Z}, \mathbf{S})$ is:

$$f_{\text{pfGMM}}(\mathbf{X}, \mathbf{Y}, \mathbf{Z}, \mathbf{S}) = \left(\prod_{d=1}^{D} f(S_d)\right) \prod_{n=1}^{N} f(Z^{(n)}) f(\mathbf{X}_{1-\mathbf{s}}^{(n)}) f(\mathbf{X}_{\mathbf{S}}^{(n)}|Z^{(n)}) f(Y^{(n)}|Z^{(n)}) \tag{14}$$

**Switch variables select relevant dimensions by tuning its prior** $p$ The switch variables $S_d$ are the key to our model. They allow us to model the 'irrelevant' dimensions $\mathbf{X}_{1-\mathbf{S}}$ using a background distribution, and to jointly model the 'relevant' dimensions and target $(\mathbf{X}_{\mathbf{S}}, \mathbf{Y})$ using a mixture distribution. Furthermore, our models depend on only one hyperparameter—the switch prior, $p$. As we will see in Section 6, the switch prior $p$ can be *tuned* to trade off generative quality with predictive performance. This trading-off ability allows our model to achieve good downstream performance even when they are (inevitably) misspecified with respect to the data-generating process.

### 4.2 Prediction-focused Hidden Markov Model (pf-HMM)

The generative process for our pf-HMM model is as follows. In addition to the switch variables $S_d$, we draw the state sequence variable $Z_t^{(n)}$ for each data point $n$ and time step $t$:

$$S_d \sim \mathbf{Bern}(p) \quad \forall d \in \{1, \ldots, D\}, \qquad Z_1^{(n)} \sim \mathbf{Cat}(\theta) \quad \forall n \in \{1, \ldots, N\},$$
$$Z_t^{(n)}|Z_{t-1}^{(n)} = j \sim \mathbf{Cat}(A_j) \quad \forall n \in \{1, \ldots, N\}, t \geq 2,$$

where $p$ is the prior probability that a dimension is relevant, $\theta$ is the probability distribution of the first state, and $A_j$ is the probability distribution of the next state given the current state $j$.

For the set of relevant dimensions, we draw the data $\mathbf{X}_{\mathbf{S},t}^{(n)}$ from a hidden Markov model with $K$ Gaussian components. For the set of irrelevant dimensions, we draw the data $\mathbf{X}_{(1-\mathbf{S}),t}^{(n)}$ from a background Gaussian distribution.

$$\mathbf{Y}_t^{(n)}|Z_t^{(n)} = j \sim \mathbf{Bern}(\eta_j), \qquad \mathbf{X}_{\mathbf{S},t}^{(n)}|Z_t^{(n)} = j \sim \mathcal{N}(\boldsymbol{\mu}_j^B, \Sigma_j^B),$$
$$\mathbf{X}_{(1-\mathbf{S}),t}^{(n)} \sim \mathcal{N}(\boldsymbol{\mu}^{\pi}, \Sigma^{\pi}), \quad \forall n, t. \tag{15}$$

The joint distribution for pf-HMM, $f_{\text{pfHMM}}(\mathbf{X}, \mathbf{Y}, \mathbf{Z}, \mathbf{S})$ is:

$$f_{\text{pfHMM}}(\mathbf{X}, \mathbf{Y}, \mathbf{Z}, \mathbf{S}) = \left(\prod_{d=1}^{D} f(S_d)\right) \prod_{n=1}^{N} \left(f(Z_1^{(n)}) \prod_{t=2}^{T} f(Z_t^{(n)}|Z_{t-1}^{(n)})\right) \left(\prod_{t=1}^{T} f(\mathbf{X}_{(1-\mathbf{S}),t}^{(n)}) f(\mathbf{X}_{\mathbf{S},t}^{(n)}|Z_t^{(n)}) f(\mathbf{Y}_t^{(n)}|Z_t^{(n)})\right) \tag{16}$$

The density $f_{\text{pfHMM}}(\mathbf{X}, \mathbf{Y}, \mathbf{Z}, \mathbf{S})$ is similar in form to the pf-GMM case (Equation 14). Both models use a complex structure to explain the relevant dimensions and the target while using a background distribution to explain the irrelevant dimensions. Furthermore, both models use a switch variable $S_d$ to select relevant dimensions. The main difference for pf-HMM is that a Markov chain of variables $\{Z_t^{(n)}\}_{t=1}^{T}$ has replaced the single variable $Z^{(n)}$ to generalize the mixture model to the time-series setting.

## 5 Inference

Since pf-GMM and pf-HMM are valid graphical models, we can use efficient inference algorithms for these models instead of resorting to generic optimization methods. The likelihood of pf-GMM is:

$$L(\Theta) = \sum_{\mathbf{Z}, \mathbf{S}} f_{\text{pfGMM}}(\mathbf{X}, \mathbf{Y}, \mathbf{Z}, \mathbf{S}; \Theta) \tag{17}$$

This objective is intractable to compute for high dimensions because it scales exponentially with the dimensionality of $\mathbf{X}$. Therefore, in practice, we use Variational Inference (VI) to optimize a lower bound

(the ELBO) to the intractable log-likelihood. We choose the following form for the approximate posterior, $q(\mathbf{Z}, \mathbf{S}|\Theta, \varphi)$:

$$q(\mathbf{Z}, \mathbf{S}|\Theta, \varphi) = \prod_{n=1}^{N} q(Z^{(n)}|\mathbf{X}^{(n)}, Y^{(n)}, \varphi, \Theta) \prod_{d=1}^{D} q(S_d^{(n)}|\varphi_d) \tag{18}$$

where $\Theta$ refers to the model parameters. Note that we have added an index $n$ to the switch variable, i.e. $S_d^{(n)}$ is a function of the datum $n$ as well as the dimension $d$. This allows the posterior to factorize over the data. To avoid learning a separate switch posterior for each $n$, we *tie* their parameters across $n$ to obtain $q(S_d^{(n)}|\varphi_d)$. Therefore we still get a per-dimension relevance.

The distribution $q(Z^{(n)}|\mathbf{X}^{(n)}, Y^{(n)}, \varphi, \Theta)$ is simply the posterior distribution conditioned on the variational parameters $\varphi$ and can be computed as:

$$q(Z = k|\mathbf{X}, Y, \varphi) \propto f(Z = k) \cdot f_k(\mathbf{X}|\varphi) \cdot f_{Y,k}(Y)$$

At test time, the distribution $q(Z = k|\mathbf{X}, Y, \varphi)$ is computed without the $P_k(Y)$ term but the *feature selection parameter* $\varphi$ helps select the right features in the term $f_k(\mathbf{X}|\varphi)$.

This results in the following ELBO objective (full derivation in Supplement Section 1):

$$\text{ELBO}(q) = \mathbb{E}_{q(\mathbf{Z},\mathbf{S})} [\log f(\mathbf{X}, \mathbf{Y}|\mathbf{Z}, \mathbf{S}; \Theta)] - \mathbf{KL} \left( q(\mathbf{Z}, \mathbf{S}|\Theta, \varphi) || f(\mathbf{Z}, \mathbf{S}; \Theta) \right) \tag{19}$$

The model is trained using the variational EM algorithm which alternates between learning the parameters $\Theta$ (M-step) and inferring the approximate posterior $q(\mathbf{Z}, \mathbf{S}|\Theta, \varphi)$ (E-step). We also try Gibbs sampling for the E-step since the variables $\mathbf{Z}$ and $\mathbf{S}$ are conditionally conjugate. We provide the conditional distributions for Gibbs sampling in the Supplement Section B.3.

Similarly, we train pf-HMM by variational EM using the following factorization for $q(\mathbf{Z}, \mathbf{S}|\Theta, \varphi)$:

$$q(\mathbf{Z}, \mathbf{S}|\Theta, \varphi) = \prod_{n=1}^{N} \left( \prod_{d=1}^{D} \prod_{t=1}^{T_n} q(\mathbf{S}_{t,d}^{(n)}|\varphi_d) \right) f(\mathbf{Z}^{(n)}|\mathbf{X}^{(n)}, \mathbf{Y}^{(n)}, \varphi, \Theta)$$

where subscript $t$ indexes time within a sequence, and $\mathbf{Z}^{(n)}, \mathbf{X}^{(n)}, \mathbf{Y}^{(n)}$ are variables for each sequence $n$. We use the Forward-Backward algorithm (Bishop, 2006) to infer $f(\mathbf{Z}^{(n)}|\mathbf{X}^{(n)}, \mathbf{Y}^{(n)}, \varphi, \Theta)$ during the E-step (see Supplement Section B for details).

## 6 Analysis of Correctness of Prediction-focused Mixtures

In this section, we show how the switch variable in the pf-GMM is able to trade off generative and discriminative performance when faced with misspecification in the number of components. We explore the limitations of GMMs in such a setting, even with supervision (i.e. for sup-GMM). Importantly, we propose analytical expressions to determine how pf-GMM identifies relevant dimensions in various data scenarios, and conclude the section with an empirical evaluation of the expressions.

### 6.1 What is enabling the generative-discriminative trade-off?

The generative-discriminative trade-off can either be achieved by the choice of model and objective, or by the choice of inference algorithm. Ren et al. (2020) motivated using switch variables to distinguish relevant and irrelevant dimensions by providing a lower bound to the marginal log-likelihood objective which exhibits an explicit generative-discriminative trade-off:

$$\log f(\mathbf{X}, \mathbf{Y}) \geq p \mathbb{E}_{f(\mathbf{Z})} [\log f(\mathbf{X}|\mathbf{Z})] + (1 - p) \log f(\mathbf{X}; \pi) + \mathbb{E}_{f(\mathbf{S})} [\log f(\mathbf{Y}|\mathbf{X}, \mathbf{S})] \tag{20}$$

where $p$ is the switch prior. This suggests that the modeling choice of using switch variables is what enables the generative-discriminative trade-off. However, this bound is not optimized during training (it is

computationally expensive to compute), and is mainly used to justify the use of switch variables. In fact, their trade-off is achieved by the specific inference algorithm they use (variational inference with a specific posterior approximation). Furthermore, favorable properties of a lower bound do not guarantee favorable properties of the model and the objective.

In this section, we demonstrate analytically that the *true* pf-GMM likelihood can balance generation and prediction in certain data regimes through the switch prior $p$. This result implies that the pf-GMM model itself is able to trade off generation and prediction, providing a theoretical justification for the use of switch variables. Furthermore, we precisely characterize the circumstances in which the user can adjust $p$ to achieve the desired trade-off. We note that Hughes et al. (2017) also achieve such a trade-off through their proposed objective, but their objective is hard to optimize. We demonstrate that our objective is easier to optimize than theirs in the experiments section.

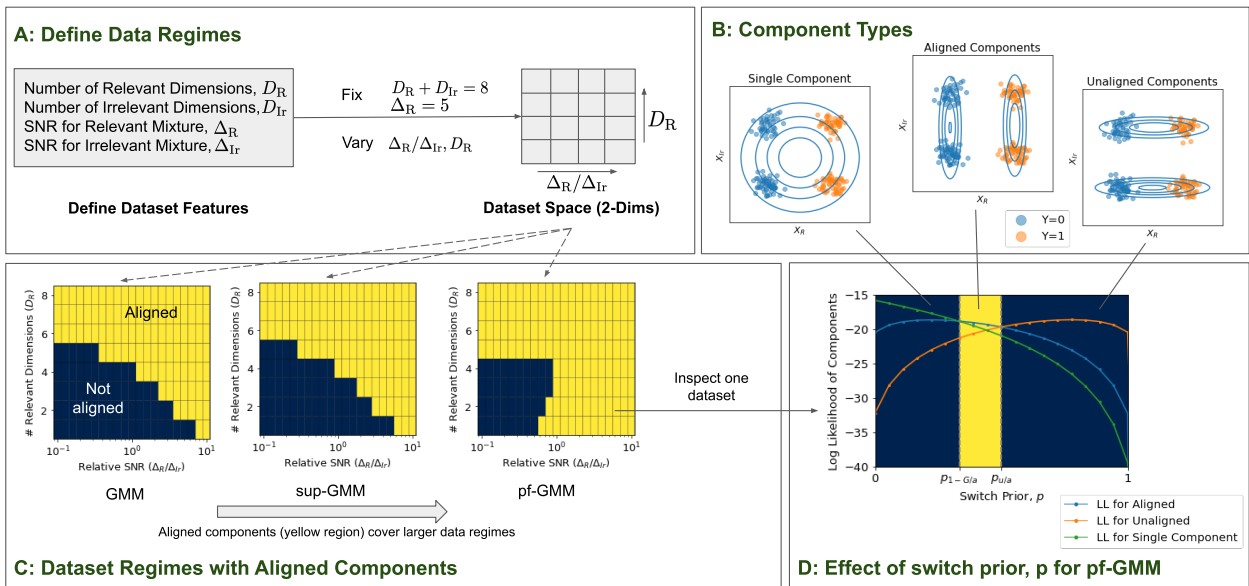

Figure 3: (A) We create representative datasets by varying the number $(D_{\mathrm{R}}, D_{\mathrm{Ir}})$ and the signal-to-noise ratio (SNR; $\Delta_{\mathrm{R}}, \Delta_{\mathrm{Ir}}$) for relevant and irrelevant dimensions. We fix the total dimensions ($D = D_{\mathrm{R}} + D_{\mathrm{Ir}}$) and the SNR for relevant dimensions; the resulting datasets are evaluated by each model to get the figures in panel C. (B) There are three local optima to the Objective 25 in a two-dimensional feature space: aligned components with label $Y$ (left), unaligned components (middle), and just a single component (right). (C) The yellow region in each plot corresponds to the datasets for which the model learns aligned components, and the blue region corresponds to the datasets for which the model learns unaligned components. From left to right, there is an increase in the space of datasets for which aligned components can be selected (i.e. GMM< sup-GMM< pf-GMM). (D) For any dataset, we see how pf-GMM can achieve aligned components by tuning $p$. Setting $p$ to a value in $(p_{1\text{-}\mathrm{G}/a}, p_{u/a})$ (yellow region) learns aligned components, whereas setting $p$ to a value in $(0, p_{1\text{-}\mathrm{G}/a}) \cup (p_{u/a}, 1)$ (blue region) learns unaligned components or a single component.

## 6.2 Defining Pertinent Data Regimes

Consider data formed by concatenating multiple independent mixture distributions, one of which is relevant; the rest are irrelevant. The possible ways in which each mixture distribution (relevant or irrelevant) can vary are (a) separation between the components, (b) variance of components, and (c) relative counts of relevant and irrelevant dimensions in $\mathbf{X}$. Finally, the number of independent irrelevant mixtures can also vary. We first consider the case where the relevant dimensions come from one mixture, and the irrelevant dimensions come from another mixture (see Figure 3(A)). This case generalizes to the analysis for multiple irrelevant mixtures since it highlights the trade-offs the model must make between relevant and irrelevant mixtures.

Consider input data $\mathbf{X} := [\mathbf{X}_{\mathrm{R}}; \mathbf{X}_{\mathrm{Ir}}] \in \mathbb{R}^{D_{\mathrm{R}}+D_{\mathrm{Ir}}}$ with $D_{\mathrm{R}}$ relevant and $D_{\mathrm{Ir}}$ irrelevant dimensions, and a binary output label $Y$. Both the relevant dimensions $\mathbf{X}_{\mathrm{R}}$ and irrelevant dimensions $\mathbf{X}_{\mathrm{Ir}}$ are distributed as equal mixtures of two Gaussians. The data-generating process is:

$$Z_{\mathrm{R}} \sim \mathbf{Bern}(0.5), \quad Z_{\mathrm{Ir}} \sim \mathbf{Bern}(0.5), \tag{21}$$

$$\mathbf{X}_{\mathrm{R}} \sim Z_{\mathrm{R}} \cdot \mathcal{N}(\mathbf{0}_{D_{\mathrm{R}}}, \sigma_{\mathrm{R}}^{*2} I_{D_{\mathrm{R}}}) + (1 - Z_{\mathrm{R}}) \cdot \mathcal{N}(\mu_{\mathrm{R}}^* \cdot \mathbf{1}_{D_{\mathrm{R}}}, \sigma_{\mathrm{R}}^{*2} I_{D_{\mathrm{R}}}), \tag{22}$$

$$\mathbf{X}_{\mathrm{Ir}} \sim Z_{\mathrm{Ir}} \cdot \mathcal{N}(\mathbf{0}_{D_{\mathrm{Ir}}}, \sigma_{\mathrm{Ir}}^{*2} I_{D_{\mathrm{Ir}}}) + (1 - Z_{\mathrm{Ir}}) \cdot \mathcal{N}(\mu_{\mathrm{Ir}}^* \cdot \mathbf{1}_{D_{\mathrm{Ir}}}, \sigma_{\mathrm{Ir}}^{*2} I_{D_{\mathrm{Ir}}}), \tag{23}$$

$$Y = Z_{\mathrm{R}} \tag{24}$$

where $Z_{\mathrm{R}}$ are the cluster identities for the relevant mixture and $Z_{\mathrm{Ir}}$ are the cluster identities for the irrelevant mixture. Note that the output label $Y$ is perfectly correlated with the relevant mixture's cluster identity $Z_{\mathrm{R}}$. Finally, let $\Delta := \mu^*/\sigma^*$ as the signal-to-noise ratio (SNR) for both relevant and irrelevant mixtures (i.e. $\Delta_{\mathrm{R}}, \Delta_{\mathrm{Ir}}$). Note: Our analysis carries over to full-covariance Gaussians; the expressions are simply more involved. We discuss the isotropic case here to convey the core ideas and provide the analogous derivations for the full-covariance case in Supplement Section C.

## 6.3 Formalizing the Question

What are the datasets (i.e. values of $\Delta_{\mathrm{R}}, \Delta_{\mathrm{Ir}}, D_{\mathrm{R}}, D_{\mathrm{Ir}}$) for which a *misspecified model* can still select the relevant components?

We answer this question by characterizing the population log-likelihood objective under the model density $f_{\mathrm{model}}(\mathbf{X}, Y; \Theta)$ and data density $f^*(\mathbf{X}, Y)$:

$$l(\Theta_{\mathrm{model}}) = \underset{f^*}{\mathbb{E}} \left[ \log f_{\mathrm{model}}(\mathbf{X}, Y; \Theta) \right] \tag{25}$$

for three models: GMM, sup-GMM, and our pf-GMM. The GMM learns a Gaussian mixture model on $\mathbf{X}$ alone, whereas the sup-GMM is the supervised variant that also uses $Y$. Upon optimization, the likelihood objective 25 can result in three locally optimal parameters. As illustrated in Figure 3(B), these parameters correspond to:

1. components that model relevant data $(\mathbf{X}_{\mathrm{R}}, Y)$ well ($\Theta^a$ or *aligned* parameters).

2. components that model irrelevant data $(\mathbf{X}_{\mathrm{Ir}})$ well ($\Theta^u$ or *unaligned* parameters).

3. (for pf-GMM only:) a single component ($\Theta^{1\text{-}G}$); this case arises if the pf-GMM considers all dimensions as irrelevant.

Among these, the aligned components have the highest predictive accuracy because they are correlated with $Y$.

## 6.4 Results for the Analysis

### 6.4.1 Likelihood Gaps for GMM, sup-GMM, and pf-GMM

The likelihood gaps for GMM and sup-GMM depend on the data features $(\Delta_{\mathrm{R}}, \Delta_{\mathrm{Ir}}, D_{\mathrm{R}}, D_{\mathrm{Ir}})$:

$$\Delta(\Theta_{\mathrm{GMM}}^a, \Theta_{\mathrm{GMM}}^u) = Q_{\mathrm{R}} - Q_{\mathrm{Ir}} \tag{26}$$

$$\Delta(\Theta_{\mathrm{supGMM}}^a, \Theta_{\mathrm{supGMM}}^u) = \log 2 + Q_{\mathrm{R}} - Q_{\mathrm{Ir}} \tag{27}$$

where $Q_{\mathrm{R}}$ is a shorthand for the following term that only depends on $D_{\mathrm{R}}$ and $\Delta_{\mathrm{R}}$ (and similarly for $Q_{\mathrm{Ir}}$):

$$Q_{\mathrm{R}} = \frac{1}{2} \left[ \frac{D_{\mathrm{R}}(1 - \Delta_{\mathrm{R}}^2)}{D_{\mathrm{R}}\Delta_{\mathrm{R}}^2 + 1} + \log(D_{\mathrm{R}}\Delta_{\mathrm{R}}^2 + 1) \right]$$

$$Q_{\mathrm{Ir}} = \frac{1}{2} \left[ \frac{D_{\mathrm{Ir}}(1 - \Delta_{\mathrm{Ir}}^2)}{D_{\mathrm{Ir}}\Delta_{\mathrm{Ir}}^2 + 1} + \log(D_{\mathrm{Ir}}\Delta_{\mathrm{Ir}}^2 + 1) \right]$$

For pf-GMM, the gaps also depend on the switch prior $p$:

$$\Delta(\Theta^a_{\text{pfGMM}}, \Theta^u_{\text{pfGMM}}) = (D_{\text{R}} - D_{\text{Ir}}) \log \frac{p}{1-p} + \log 2 + Q_{\text{R}} - Q_{\text{Ir}} \tag{28}$$

$$\Delta(\Theta^a_{\text{pfGMM}}, \Theta^{\text{1-G}}_{\text{pfGMM}}) = D_{\text{R}} \log \frac{p}{1-p} + \log 2 + Q_{\text{R}} \tag{29}$$

**When can a misspecified model select relevant components?** The models will select aligned components (over unaligned) when the likelihood gap $\Delta(\Theta^a_{\text{model}}, \Theta^u_{\text{model}})$ is positive. For our pf-GMM, we also need the gap $\Delta(\Theta^a_{\text{pfGMM}}, \Theta^{\text{1-G}}_{\text{pfGMM}})$ to be positive, that is the likelihood associated with the aligned components must be higher than that of collapsing the model to a single Gaussian.

### 6.4.2 Tuning pf-GMM to prefer relevant components

For pf-GMM, we can tune the switch prior $p$ to prefer relevant components in Equations 28 and 29. We can perform this selection of relevant components by identifying values of $p$ for which $\Delta(\Theta^a_{\text{pfGMM}}, \Theta^u_{\text{pfGMM}}) > 0$ and $\Delta(\Theta^a_{\text{pfGMM}}, \Theta^{\text{1-G}}_{\text{pfGMM}}) > 0$. Doing so gives us the following constraints on $p$:

$$p < p_{u/a} \qquad \text{where} \qquad p_{u/a} = \sigma\left(\frac{\log 2 + Q_{\text{R}} - Q_{\text{Ir}}}{|D_{\text{R}} - D_{\text{Ir}}|}\right) \qquad \text{if } D_{\text{R}} < D_{\text{Ir}} \tag{30}$$

$$p > p_{u/a} \qquad \text{where} \qquad p_{u/a} = \sigma\left(-\frac{\log 2 + Q_{\text{R}} - Q_{\text{Ir}}}{|D_{\text{R}} - D_{\text{Ir}}|}\right) \qquad \text{if } D_{\text{R}} > D_{\text{Ir}} \tag{31}$$

$$p > p_{\text{1-G}/a} \qquad \text{where} \qquad p_{\text{1-G}/a} = \sigma\left(-\frac{\log 2 + Q_{\text{R}}}{D_{\text{R}}}\right) \tag{32}$$

where $\sigma(x) = \frac{1}{1+\exp(-x)}$ is the sigmoid function. The feasible range of $p$ exists when $p_{u/a} > p_{\text{1-G}/a}$, which is when:

$$D_{\text{Ir}} \log 2 - D_{\text{R}} Q_{\text{Ir}} + D_{\text{Ir}} Q_{\text{R}} > 0 \text{ if } D_{\text{R}} < D_{\text{Ir}}$$

### 6.5 Empirical Illustrations of Analytical Results

We empirically illustrate when these gaps can be positive by setting $D_{\text{R}} + D_{\text{Ir}} = 8$ and $\Delta_{\text{R}} = 5$ as an example in Figure 3(C). For each model, we vary $\Delta_{\text{Ir}}$ and $D_{\text{Ir}}$ to create datasets with different SNR and a number of irrelevant dimensions. The yellow region in each plot corresponds to the datasets (i.e. values of $(\Delta_{\text{Ir}}, D_{\text{Ir}})$) for which the model learns aligned components as per Equations 26-29. The blue region corresponds to the datasets for which the model fails to learn aligned components. We see that the range of datasets for the aligned parameters (i.e. the yellow area) is largest for pf-GMM, followed by sup-GMM, and smallest for a GMM.

Furthermore, in Figure 3(D), we illustrate how our pf-GMM model selects the aligned components by tuning $p$. By setting $p$ to a value in $(p_{\text{1-G}/a}, p_{u/a})$, we learn aligned components (corresponding to the yellow region in the figure), whereas by setting $p$ to a value in $(0, p_{\text{1-G}/a}) \cup (p_{u/a}, 1)$, we either learn unaligned components or just a single component (corresponding to the blue region in the figure). This figure provides us with guidance on performing hyperparameter tuning for $p$: to learn aligned components, we decrease $p$ if we currently learned unaligned components, and increase $p$ if we learned a single component.

### 6.6 Conclusions from the Analysis

**The pf-GMM allows us to identify relevant structure by changing $p$.** To choose the aligned parameters, the gaps in Equations 26- 29 must be positive. Comparing Equation 26 with 27 shows supervision is useful over just using the GMM—the gap for sup-GMM is $\log 2$ larger than the gap for GMM. However, the pf-GMM does better: its gaps (Equations 28 and 29) also include a term involving $p$, allowing us to *tune* $p$ so that the gaps are positive.

When the relevant dimensions are outnumbered by irrelevant dimensions, learning the unaligned components would be preferred by GMM (and sup-GMM after an extent) since doing so increases the generative performance of the model. However, maximizing predictive performance would require learning the aligned components. Setting $p$ to a value in $(p_{\text{1-G}/a}, p_{u/a})$ allows us to select aligned components, and setting it in $(p_{u/a}, 1)$ allows us to select unaligned components (Equations 30 and 32). **In this manner, we can trade off between predictive and generative performance**: the aligned parameters achieve predictive performance while the unaligned parameters achieve generative performance—and $p$ lets us decide which ones to choose.

**pf-GMM identifies predictive mixtures even with many irrelevant dimensions.** Suppose we fix means $\mu_{\text{R}}^*, \mu_{\text{Ir}}^*$ and variances $\sigma_{\text{R}}^{*2}, \sigma_{\text{Ir}}^{*2}$ (i.e., fix $\Delta_{\text{R}}$ and $\Delta_{\text{Ir}}$). For the interesting case of fewer relevant dimensions $D_{\text{R}} < D_{\text{Ir}}$, lowering $p$ below $p_{u/a}$ forces the aligned parameters to be preferred over the unaligned parameters. The other models cannot achieve this: the gap $\Delta(\Theta_{\text{GMM}}^a, \Theta_{\text{GMM}}^u)$ is negative for the GMM and only increases by $\log 2$ for the sup-GMM (Equations 26, 27). However, there are limits to pf-GMM, too: if we lower $p$ below $p_{\text{1-G}/a}$ when $D_{\text{R}} < D_{\text{Ir}}$, pf-GMM, prefers the single Gaussian solution (as seen in Figure 3(D)).

**pf-GMM is more robust to high irrelevant SNR.** We vary the irrelevant SNR $\Delta_{\text{Ir}}$ by varying the gap between irrelevant component $(\mu_{\text{Ir}}^*)$, while fixing the means of the relevant components $(\mu_{\text{R}}^*)$ and all variances $(\sigma_{\text{R}}^{*2}, \sigma_{\text{Ir}}^{*2})$. The case of $\sigma_{\text{R}}^* >> \sigma_{\text{Ir}}^*$ is analogous. For all models, the likelihood gap decreases on increasing $\Delta_{\text{Ir}}$ (see Equations 26-29). However, pf-GMM selects aligned components for a larger range of $\Delta_{\text{Ir}}$. For the interesting case of $D_{\text{R}} < D_{\text{Ir}}$, we can see this by setting $p < p_{u/a}$ in Equation 30—doing so adds a positive bias to the gaps of GMM and sup-GMM, making it easier to select aligned components. Nonetheless, if $\Delta_{\text{Ir}}$ is too high, decreasing $p$ no longer works because $p_{u/a} < p_{\text{1-G}/a}$: we go directly from the unaligned solution $\Theta^u$ to the single Gaussian solution $\Theta^{\text{1-G}}$, bypassing the predictive solution $\Theta^a$.

*This case highlights the properties and limitations of all three models*: our pf-GMM has the widest range of datasets for which it will still select the desired aligned parameters, but there will be a level of signal-to-noise ratio at which all models will fail.

**Cases of $D_{\mathbf{R}} > D_{\mathbf{Ir}}$ and $D_{\mathbf{R}} = D_{\mathbf{Ir}}$.** $D_{\text{R}} > D_{\text{Ir}}$ is the (easy) regime where GMM (and therefore sup-GMM) work as desired and introducing pf-GMM is not necessary. Finally, for $D_{\text{R}} = D_{\text{Ir}}$, the effect of $p$ disappears irrespective of the parameters. This case is an unlikely technicality, which we rarely expect in real datasets.

**Effect of the number of irrelevant mixtures.** When multiple irrelevant mixtures are present, we only need to focus on the irrelevant mixture that yields model parameters (with model components aligned with mixture components) with the highest population log-likelihood, $l(\Theta)$. Then the same analysis applies.

## 7   Experimental Details

In this section, we empirically demonstrate that the pf-GMM successfully identifies relevant input signal, regardless of whether the number of model components is misspecified or not. We show that the switch parameters and their prior $p$ can recover the relevant dimensions and maintain predictive performance even under misspecification. Specifically, we address the following questions:

**Q1:** Can prediction-focused models perform well when we misspecify the number of components?

**Q2:** Can prediction-focused models perform well as the number of irrelevant dimensions increases?

**Q3:** Do prediction-focused models select relevant dimensions, even when irrelevant dimensions are not fully independent of the signal?

**Q4:** Is the optimization stable for these models?

For both pf-GMM and pf-HMM, we constrain the models to have diagonal covariance for faster inference; analogous results for the full-covariance case are in Supplement Section 3.2.

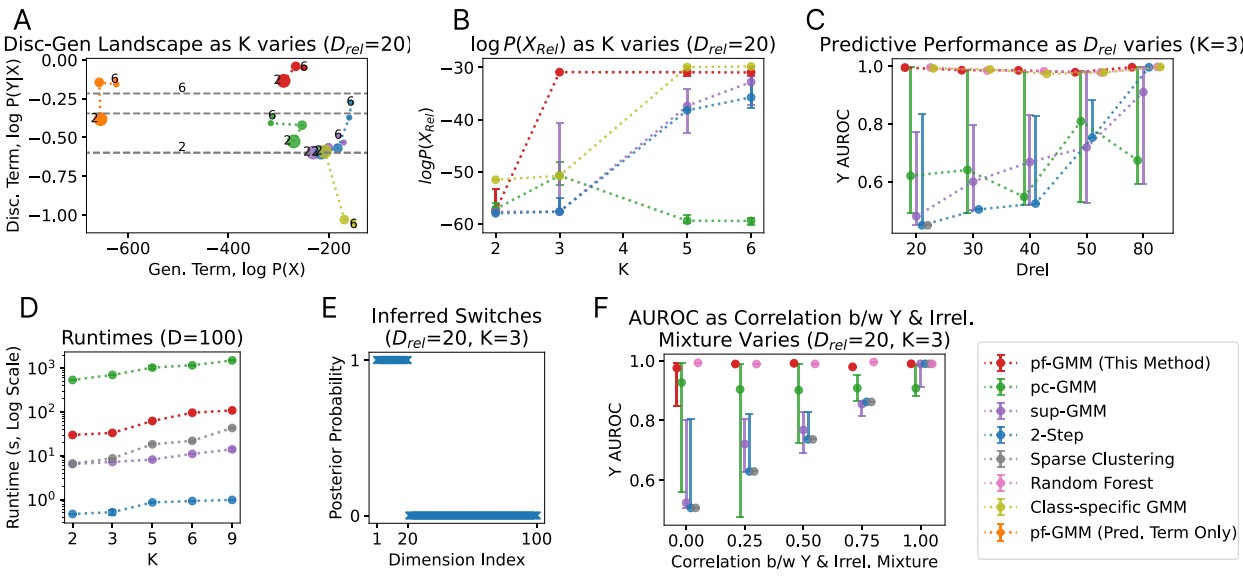

Figure 4: Plots for synthetic data (Section 7). (A) pf-GMM is an order of magnitude faster than pc-GMM, with respect to the average runtimes (log scale) per run. (B, E) pf-GMM trades off discriminative and generative objectives better—it takes a small hit in generative quality to achieve discriminative quality competitive with pf-GMM trained only on the predictive term. The benefits are largest when misspecification is larger (fatter markers). (C) Constrained pf-GMM achieves optimal predictive performance by identifying the relevant dimensions (random forest baseline provides the upper bound). (D) Posterior probabilities $f(S_d = 1|\mathbf{X}, \mathbf{Y})$ for the switch variables correctly identify the first 20 dimensions as relevant.

Table 1: Comparison of discriminative and generative metrics for the synthetic (complex) dataset shows that pf-GMM, outperforms other baselines.

| Model | $\log P(Y|\mathbf{X})$ | $\log P(\mathbf{X})$ | $\log P(\mathbf{X}_{\mathrm{R}})$ | Y AUROC |
|---|---|---|---|---|
| pf-GMM | **-0.05** | -105.65 | **7.38** | **0.99** |
| pc-GMM | -0.50 | -130.41 | -15.51 | 0.66 |
| sup-GMM | -0.66 | -30.81 | -15.48 | 0.58 |
| 2-Step | -0.70 | 5.79 | -26.23 | 0.52 |
| Class-specific GMM | -5.92 | **14.87** | -20.66 | 0.73 |

## 7.1 Baselines

We compare the generative and predictive performance of pf-GMM to (1) a two-step approach that learns a GMM and trains a logistic regression classifier on the posteriors $f(Z|\mathbf{X})$ (2-Step-GMM), (2) supervised GMM trained jointly on $\mathbf{X}, \mathbf{Y}$ (sup-GMM), (3) prediction-constrained GMM (Hughes et al., 2017) (pc-GMM) and (4) sparse K-Means (Witten & Tibshirani) followed by logistic regression on cluster values (sparse clustering). In Supplement Section F, we also discuss a class-specific GMM baseline that learns a different GMM for each class value and shows that it is qualitatively similar to a GMM baseline. We also provide a random forest baseline as a prediction upper bound. We discuss the utility of these baselines in depth in the Supplement Section 4 For pf-HMM, comparable approaches are sup-HMM and 2-Step-HMM.

## 7.2 Datasets

**Synthetic** We generate two synthetic datasets, where the first dataset is simple and the second dataset is complex. For the first dataset, we generate the relevant and irrelevant dimensions of $\mathbf{X}$ from independent

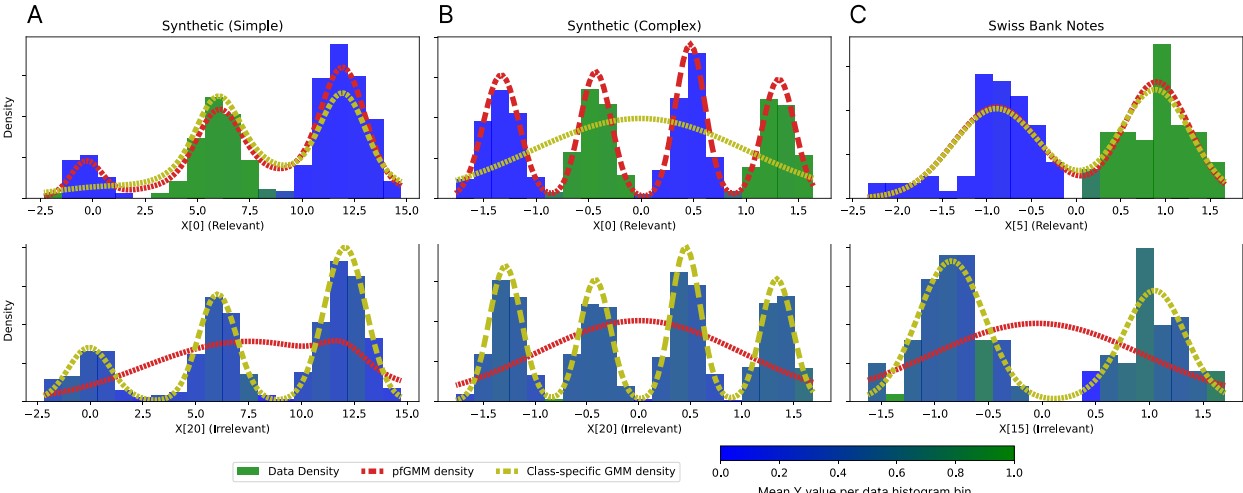

Figure 5: Densities of relevant (top) and irrelevant (bottom) dimensions of inputs **X** for Synthetic (left and center) and Swiss Bank Notes datasets. The Synthetic (complex) in the middle column is *complex* because all the classes have multimodal densities. By this definition, both Synthetic (simple) and Swiss Bank Notes are considered *simple*. We also see that the pf-GMM, can model the relevant dimensions well (red line corresponds to the pf-GMM, model density) in all the datasets. In contrast, class-specific GMM (olive line) prioritizes modeling irrelevant dimensions' data density (bottom rows) and is only able to capture relevant dimensions' data density (top row) when the datasets are simple.

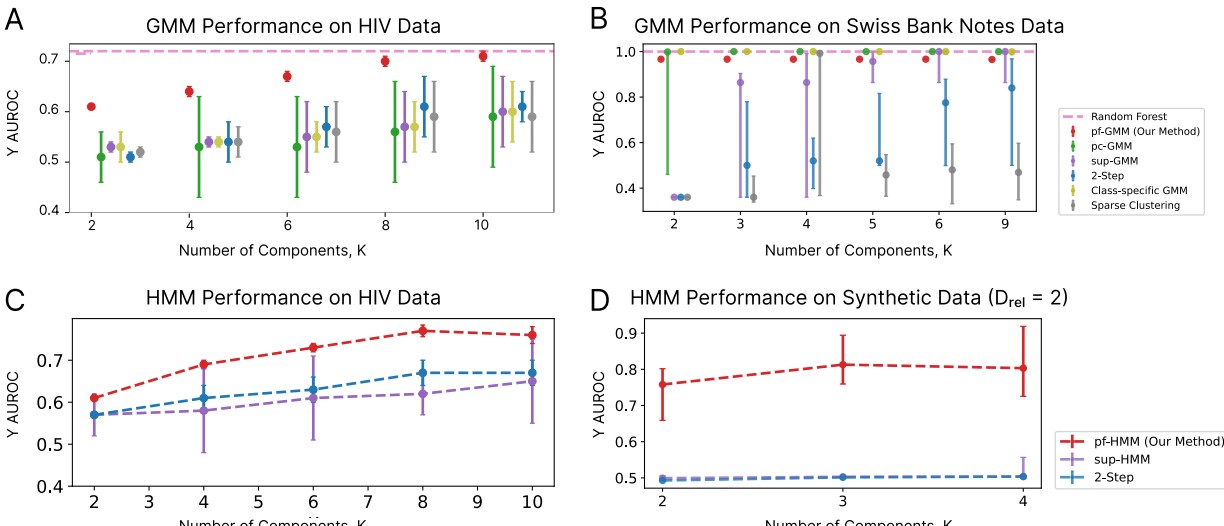

Figure 6: Target AUROC performance as we vary the component budget $K$ in each of the domains for GMMs (Top Row) and HMMs (Bottom Row). pf-GMMs and pf-HMMs outperform methods that do not trade off *prediction and generation* for different choices of $K$. Synthetic data for the HMM models has 2 relevant dimensions (out of 20 total).

$3-$component mixtures of Gaussians. The target $Y$ is binary, with the assignment only depending on the cluster identities in the relevant GMM. The dataset is simple in the sense that the relevant dimension densities for each class are sufficiently unimodal: there is only one mode for the class $Y = 1$ and most of the mass is concentrated in one mode for class $Y = 0$ (see the histograms in the top left panel of Figure 5 for a visual representation). In the second dataset, we generate the relevant and irrelevant dimensions from independent $4-$component mixtures, and again the target is binary and depends on the relevant

GMM's cluster identities. By the same definition of complexity, the second dataset is complex because the relevant dimension densities are multimodal for each class (see the top center panel of Figure 5). The total dimensionality of each observation is fixed at 100. To generate time-series data, we use independent $4-$state HMMs with 20 dimensions. The emission distribution $f_k^*(\mathbf{X}|Z = k)$ given component $k$ is $\mathcal{N}(\cdot; 6k \cdot \mathbf{1}, I)$ for relevant and irrelevant dimensions. The target $Y$ is binary and depends on the relevant HMM (details in Supplement Section 4). We vary the number of model components ($K$) and the ratio of relevant dimensions for different levels of misspecification. For **Q3**, we also simulate irrelevant dimensions that are not completely independent of $Y$; for this experiment, we vary the correlation between $Y$ and the cluster identity.

**Swiss Bank Notes**   The Swiss bank notes dataset (available in the MCLUST library in R (Scrucca et al., 2016)) contains 6 features about Swiss 1000-franc banknotes: length of bill, widths of left, right, top, bottom edges, and length of diagonal. The dataset contains 100 counterfeit and 100 genuine notes. We want the clusters to be predictive of whether a banknote is genuine or counterfeit (label $Y$). However, not all features are relevant to the task of predicting this label. Therefore, prediction-focused modeling is appropriate for this situation. To further demonstrate the robustness of pf-GMM to noisy features, we augment the existing features with 30 irrelevant features generated using a $2-$component mixture of Gaussians with components $\mathcal{N}(-3, I)$ and $\mathcal{N}(3, I)$ to make prediction harder, and finally normalize the dataset. For a method to do well, it must identify clusters that are predictive of the genuineness of the notes even when the cluster count is small (e.g. when only 2 clusters are allowed).

**HIV**   Therapy for HIV involves administering cocktails of antiretroviral treatments to bring the viral load below detection limits ($\leq 40$ copies/ml). These antiretroviral treatments belong to five classes: Non-nucleoside Reverse Transcriptase Inhibitors (nnRTIs), Nucleoside Reverse Transcriptase Inhibitors (nRTIs), Protease Inhibitors (PIs), Fusion Inhibitors (FIs), and Integrase Inhibitors (IIs). Our task is to predict whether a treatment will bring the viral load below detection limits in the next time-step, or whether it will increase the viral load ($Y$). Each input observation ($X$) contains 267 features about the patient, including their CD4+counts, their genetic mutations, their treatments in terms of drug classes, and their lab results. We study $53,236$ patients with HIV from the EuResist Integrated Database. Each person has a time series of an average length of 16 steps where a time step is approximately 4 months between consecutive treatments. Though it is common to have many genetic mutations, only a few of these may be relevant for inducing drug resistance thus increasing the viral load. Therefore, this problem is perfect for prediction-focused modeling.

We compare both HMM and GMM models, ignoring the time dependencies for the latter case. Further details about the experiments are provided in Supplement Section D.

**Psychiatric Medical Records**   We analyzed a dataset of electronic health records from two large academic medical centers and their affiliated community hospitals and outpatient clinics. We selected a cohort of 16,653 patients with at least one diagnosis of major depressive disorder ICD-10 diagnosis code during psychiatric care between 2017 and 2022. We remapped the ICD-10 codes to 423 CCSR codes (i.e. features). We mapped each feature with a value of 1 if the diagnosis code was observed, and 0 otherwise. An independent standard normal observation noise with a standard deviation of 0.1 was added. After excluding codes with high sparsity ($\geq 90\%$ patients with the code) and codes that were highly correlated with another code, we were left with 106 features for a patient. Our prediction task is to ascertain a substance-use disorder outcome for the patient, and the goal of the clustering step is to discover subgroups relevant to the prediction task. This task would benefit from prediction-focused modeling because, without the prediction task, the clusters would correspond to common comorbidities of MDD. However, we wish to discover clusters of substance-use-related comorbidities.

### 7.3   Evaluation

We measure the predictive performance of the models by $\log P(Y|\mathbf{X})$ and area under the receiver operating characteristic (AUROC) for the classification of $Y$ on heldout test data. We measure generative performance by $\log P(\mathbf{X})$. For synthetic datasets, we know which dimensions are relevant, so we can also measure the generative performance of the models on only the relevant dimension, $\log P(\mathbf{X}_R)$. Since we do not have this metric in practice, we use $Y$ AUROC for model selection. Using $Y$ AUROC works well in practice because we

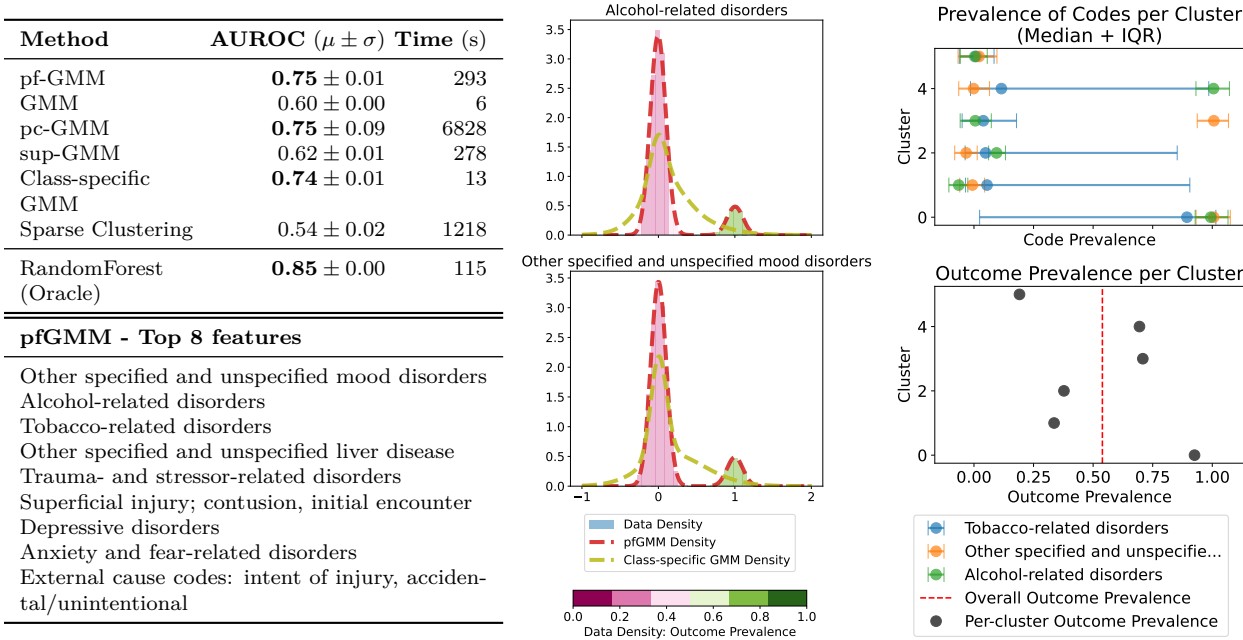

| Method | AUROC ($\mu \pm \sigma$) | Time (s) |
|---|---|---|
| pf-GMM | **0.75** $\pm$ 0.01 | 293 |
| GMM | 0.60 $\pm$ 0.00 | 6 |
| pc-GMM | **0.75** $\pm$ 0.09 | 6828 |
| sup-GMM | 0.62 $\pm$ 0.01 | 278 |
| Class-specific GMM | **0.74** $\pm$ 0.01 | 13 |
| Sparse Clustering | 0.54 $\pm$ 0.02 | 1218 |
| RandomForest (Oracle) | **0.85** $\pm$ 0.00 | 115 |

| **pfGMM - Top 8 features** |
|---|
| Other specified and unspecified mood disorders |
| Alcohol-related disorders |
| Tobacco-related disorders |
| Other specified and unspecified liver disease |
| Trauma- and stressor-related disorders |
| Superficial injury; contusion, initial encounter |
| Depressive disorders |
| Anxiety and fear-related disorders |
| External cause codes: intent of injury, accidental/unintentional |

Figure 7: Results for psychiatric medical records dataset. **Top Left**: pf-GMM learns clusters that are predictive of the substance use disorder outcome, as seen by a high $Y$ AUROC value. Other clustering methods either produce clusters unrelated to the $Y$ outcome (2-Step-GMM, sup-GMM, Sparse Clustering) or are very slow to run and have higher variation (pc-GMM). **Bottom Left**: The top features that pf-GMM clusters on are indeed related to substance use disorder. **Center**: The class-specific GMM does not model relevant dimension (alcohol and mood disorders) density despite having a high $Y$ AUROC, whereas the pf-GMM, correctly captures the density. **Right**: The pf-GMM clusters show a significant variation in the outcome and the relevant feature values, providing evidence that the clusters are both predictive and relevant.

.

can achieve good predictive performance with the clusters only if the relevant dimensions are well modeled (the class-specific GMM can create false positives with this metric because it has more parameters, as we discuss later).

Model selection is done using a heldout validation set. For Swiss Bank Notes Data, we use 3-fold Stratified Cross-validation because the observations are too few to have a separate validation set.

# 8 Results and Discussion

**Prediction-focused models balance the discriminative and generative objectives effectively.** Figure 4 (A) compares pf-GMM to competing approaches on the discriminative-generative landscape. A higher number of irrelevant dimensions or a lower number of components are exactly the settings in which the trade-off of discriminative and generative objectives is critical and pf-GMM manages to achieve superior predictive performance in these situations. In contrast, sup-GMM and 2-Step-GMM are only useful when the number of relevant dimensions (the number of components respectively) is increased—as predicted by our analysis in Section 6. The class-specific GMM fits independent GMMs on smaller subsets of the data, causing it to overfit its $\log f(Y|\mathbf{X})$ density to the training set. Finally, directly optimizing for $\log f(Y|\mathbf{X})$ (orange) gives up a lot on generative performance.

**Prediction-focused models identify parameters that perform significantly better at the downstream task.** Prediction-focused models identify components that align with the relevant dimensions (Figure 5 top left). This helps them outperform their non-prediction-focused counterparts on predictive

performance on several real and synthetic datasets (Figures 4, 6, and 7). While sup-GMM also maximizes the joint likelihood of inputs and the target, it treats the target as just another input dimension. In contrast, prediction-focused learning identifies the asymmetry in the problem, and it emphasizes dimensions that are predictive of the target. The class-specific GMM can achieve good predictive performance, suggesting it also identifies the relevant dimensions. However, we see in Figure 5 that it aligns its components with the irrelevant dimensions (bottom row), and can not align with relevant dimensions when the datasets are complex (top row, center).

**Prediction-focused models seek out the most correlated dimensions first.** As we show in Figure 4(C and F), our constrained pf-GMM achieves optimal predictive performance (random forest baseline provides the upper bound). Ignoring irrelevant dimensions can be easy if the irrelevant dimensions are independent of the relevant dimensions (i.e. correlation to $Y$ is zero). But the pf-GMM seeks out relevant dimensions even when the irrelevant dimensions 'distract' the model by being weakly correlated to $Y$, as seen in Figure 4(F) The rest of the models do a poor job at this: their performance is proportional to how correlated the irrelevant dimensions are to $Y$. A correlation of one is trivial because the notion of an 'irrelevant' dimension disappears.

**Prediction-focused models are faster to compute and they are easier to tune and optimize.** In Figures 4(D) and 7(top left), we see that pf-GMM is orders of magnitude faster than the pc-GMM— the only other baseline that can trade off generative and discriminative objectives. Moreover, pc-GMM's performance is unstable and it requires more independent trials to run, as can be seen from the large error bars corresponding to [5,95]-percentile intervals (Figure 4(C) and 6(A)). The switch parameter $p$ is the only hyperparameter that must be tuned. In contrast, while pc-GMM also has one hyperparameter $\lambda$—used for trading off generative with predictive performance—the model is still considerably harder to optimize. Training requires a grid search over both learning rates and $\lambda$, and it takes much longer to converge than the EM-based methods. This makes it prohibitively expensive to apply to HMMs.

**Prediction-focused model learns an accurate density of the relevant dimensions even for complex data distributions.** In Figures 4(B) and 5(top row), we see that pf-GMM, learns a better density for the relevant dimensions even when the model is constrained (i.e., for small values of K), as measured by $\log P(\mathbf{X}_{\mathbb{R}})$. In contrast, the class-specific GMM learns a poor density of the data even when it does well at predicting the target $Y$. This phenomenon is obscured on simple data that does not have a multimodal data density in $X$ for the values of output class $Y$. This is because the class-specific GMM can use its components to model the irrelevant dimension density but still do well on the prediction task and model relevant dimension density well if the data set is simple. However, for a complex dataset with multimodal relevant dimension density per class, this is no longer possible and the class-conditional baseline fails. We discuss this further in the Supplement Section F.

**Prediction-focused models identify relevant dimensions in both misspecified and high-noise settings.** In Figure 6, we see that prediction-focused models maintain high AUROC—indicating that the components discovered are predictive of the downstream task. In particular, we see that pf-GMM maintains good performance even when $K$ is small. The model also correctly identifies that the first 20 dimensions contribute to clustering. In contrast, the baselines do not offer ways of inspecting the models, with the exception of Sparse Clustering. Both small $K$ and feature attribution in the misspecified setting are critical in our motivating example of disease subtyping.

**Conclusions from HIV dataset** In HIV data (Figure 6, left column), for patients whose treatment is predicted to fail by pf-GMM and pf-HMM, we observe that several resistance-relevant mutations are identified as 'relevant'. For example, for patients taking nRTIs experiencing virologic failure, prediction-focused models predominantly identify certain mutations such as K65R, I116V, and M184V as relevant. These are consistent with Wensing et al. (2019). Patients who are more likely to achieve treatment success tend to be younger female patients or those with mutations consistent with subtypes that are known to be easier to treat e.g. subtype B. This is consistent with some bodies of work that have examined the success of treatment varies across different HIV subtypes (e.g. Gatell (2011))

**Conclusions from Psychiatric Medical Records dataset** From the relevant features table in Figure 7, we see that pf-GMM correctly identifies features relevant to substance use disorder (SUD) outcome. The model correctly identifies that the presence of diagnosis codes related to alcohol and tobacco disorders means that the patient has SUD. It also identifies that a history of trauma, mood disorders, depression, and liver disease can be indicative of SUD prevalence. These discovered features are consistent with the comorbidities of SUD in the literature (NIDA, 2020). On comparing the pf-GMM clusters on code prevalence (top right panel in Figure 7), we observe that codes for mood, tobacco, and alcohol-related disorders vary significantly between clusters. This variation explains the difference in outcome prevalence (bottom right panel). Cluster 0 consists of patients with a high prevalence of all three codes and high outcome prevalence. Cluster 5 is another pure cluster, consisting of patients with mood, tobacco, and alcohol-related disorders. Cluster 3 only consists of patients with mood disorders but no tobacco or alcohol use, suggesting that these patients used other substances since the outcome prevalence is still high. However, mood disorders are not a necessary condition for high SUD: cluster 4 consists of patients with SUD but without diagnosis of mood disorders. pf-GMM allowed us to recover these clusters despite other significant (but irrelevant) clusters in the population. The other model-based clustering models did not recover a meaningful clustering along these features and did not identify the features they focused on.

# 9   Conclusion

We developed prediction-focused generative modeling, a model-based feature selection method for mixture models and hidden Markov models. More importantly, we developed an understanding about when this type of approach to feature selection will work and when it will fail. Prediction-focused modeling selects the 'right' features to cluster even when the inputs are corrupted with several independent irrelevant features. It achieves this trade-off by maximizing the likelihood of the data directly, instead of resorting to any property of a modified objective as proposed by Hughes et al. (2018a) or of the approximate posterior as done by Ren et al. (2020). Our main assumption is in the types of correlations we assume for the relevant and irrelevant variables (i.e. mixture distribution and hidden Markov distribution). In our analysis, we show that prediction-focused models envelop supervised mixture models in the data space they can model in the constrained-components setting. Finally, we validate our findings on both synthetic and real-world datasets, including a medical dataset with a large amount of noise.

## Acknowledgements

This material is based upon work supported by the National Science Foundation under Grant No. IIS-1750358, Grant No. IIS-2007076 and by NIH award R01MH123804. Any opinions, findings, and conclusions or recommendations expressed in this material are those of the author(s) and do not necessarily reflect the official views of the National Science Foundation or the National Institutes of Health.

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

## A  Supplement Outline

In Section B, we provide the derivation of ELBO and coordinate ascent updates for our prediction-focused models. In Section C, we provide the derivations for the analysis we presented in the main paper. The section is divided into three parts: first, we provide the derivations for the sup-GMM case, then for the GMM case, and finally for the pf-GMM case. For each method, we first derive general expressions, and then expressions for the special case to build intuition for the theory. In Section D and E, we provide additional experiments and further details about the experiments.

## B  Inference Details

In this section, we provide the derivation of ELBO and coordinate ascent updates for our prediction-focused models. We provide the derivation for the pf-HMM case, where $N$ is the number of sequences and $T_n$ signifies the length of sequence $n$. The results for pf-GMM case follow by setting $T_n$ to 1 for all $n$. The parameters of the model are $\Theta = \{\theta, A, B, \pi, \eta\}$.

### B.1  ELBO derivation

The ELBO for one sequence $\mathbf{X}, \mathbf{Y}$ with the posterior distribution $q(\mathbf{Z}, \mathbf{S}|\Theta, \varphi)$ is:

$$
\begin{aligned}
\log f(\mathbf{X}, \mathbf{Y}) &\geq \mathop{\mathbb{E}}_{q(\mathbf{Z}, \mathbf{S})} \left[ \frac{\log f(\mathbf{Z}, \mathbf{S}, \mathbf{X}, \mathbf{Y}|\Theta, p)}{q(\mathbf{Z}, \mathbf{S}|\Theta, \varphi)} \right] \\
&= \mathop{\mathbb{E}}_{q(\mathbf{Z}, \mathbf{S})} \left[ \log f(\mathbf{X}, \mathbf{Y}|\mathbf{Z}, \mathbf{S}, \Theta) \right] - \mathbf{KL}\left( q(\mathbf{Z}, \mathbf{S}|\Theta, \varphi) || f(\mathbf{Z}, \mathbf{S}|A, \theta, p) \right) \\
&= \mathop{\mathbb{E}}_{q} \left[ \log f(\mathbf{X}|\mathbf{Z}, \mathbf{S}, B, \pi) \right] + \mathop{\mathbb{E}}_{q} \left[ \log f(\mathbf{Y}|\mathbf{Z}, \eta^\mu, \eta^V) \right] - \mathbf{KL}\left( \log q(\mathbf{S}|\varphi) || \log f(\mathbf{S}|p) \right) \\
&\quad - \mathbf{KL}\left( \log q(\mathbf{Z}|\mathbf{X}, \mathbf{Y}, \varphi, \Theta) || \log f(\mathbf{Z}|\theta, A) \right)
\end{aligned}
$$

where the individual terms are (define $q(Z_t|\mathbf{X}, \mathbf{Y}, \Theta, \varphi) = q(Z_t)$) :

$$
\begin{aligned}
\mathop{\mathbb{E}}_{q} \left[ \log f(\mathbf{X}|\mathbf{Z}, \mathbf{S}, B, \pi) \right] &= \mathop{\mathbb{E}}_{q(\mathbf{S}|\varphi)q(\mathbf{Z}|\mathbf{X}, \mathbf{Y}, \Theta, \varphi)} \left[ \sum_{n=1}^{N} \sum_{t=1}^{T_n} \sum_{d=1}^{D} S_{t,d}^n \log P_B(X_{td}^n | Z_t^{\ n}, B) + \left(1 - S_{t,d}^n\right) \log P_\pi(X_{td}^n|\pi) \right] \\
&= \sum_{n=1}^{N} \sum_{t=1}^{T_n} \sum_{d=1}^{D} \varphi_d \mathop{\mathbb{E}}_{q(Z_t^n)} \left[ \log P_B(X_{td}^n|Z_t^n, B) \right] + (1 - \varphi_d) \log P_\pi(X_{td}^n|\pi)
\end{aligned}
$$

$$\mathbb{E}_q \left[\log f(\mathbf{Y}|\mathbf{Z}, \mathbf{S}, B, \pi)\right] = \mathbb{E}_{q(\mathbf{Z}|\mathbf{X}, \mathbf{Y}, \Theta, \varphi)} \left[\sum_{n=1}^{N} \sum_{t=1}^{T_n} \log f(Y_t^n | Z_t^n, \eta^\mu, \eta^V)\right]$$

$$= \sum_{n=1}^{N} \sum_{t=1}^{T_n} \mathbb{E}_{q(Z_t^n)} \left[\log p(Y_t^n | Z_t^n, \eta^\mu, \eta^V)\right]$$

$$-\mathbf{KL}\left(\log q(\mathbf{Z}|\mathbf{X}, \mathbf{Y}, \varphi, \Theta) || \log f(\mathbf{Z}|\theta, A)\right) = \sum_{n=1}^{N} \mathbb{E}_{q(Z_1^n)} \left[\log \frac{f(Z_1^n|\theta)}{q(Z_1^n|\mathbf{X}, \mathbf{Y})}\right] + \sum_{n=1}^{N} \sum_{t=2}^{T_n} \mathbb{E}_{q(Z_{t-1,t}^n|\mathbf{X}, \mathbf{Y})} \left[\log \frac{f(Z_t^n|Z_{t-1}^n, A)}{q(Z_t^n|Z_{t-1}^n, \mathbf{X}, \mathbf{Y})}\right]$$

$$-\mathbf{KL}\left(\log q(\mathbf{S}|\varphi) || \log f(\mathbf{S}|p)\right) = \mathbb{E}_{q(\mathbf{S}|\varphi)} \left[\sum_{n=1}^{N} \sum_{t=1}^{T_n} \sum_{d=1}^{D} \log \frac{f(S_{t,d}^n|p)}{q(S_{t,d}^n|\varphi_d)}\right]$$

$$= \sum_{n=1}^{N} \sum_{t=1}^{T_n} \sum_{d=1}^{D} \varphi_d \log \frac{p}{\varphi_d} + (1 - \varphi_d) \log \frac{1-p}{1-\varphi_d}$$

## B.2 Coordinate Ascent Updates

Rewriting the ELBO again:

$$L = \sum_t \left(\sum_{d=1}^{D} \varphi_d \mathbb{E}_{q(Z_t|\mathbf{X}, \mathbf{Y})} \left[\log P_B(X_{td}|Z_t)\right] + (1 - \varphi_d) \log p_\pi(X_{td})\right) + \mathbb{E}_{q(Z_T|\mathbf{X}, \mathbf{Y})} \left[\sum_{t=0}^{T_n} \log p(Y_t|Z_t, \eta^\mu, \eta^V)\right]$$

$$- \sum_d \mathbb{E}_{q(S_d|\varphi_d)} \left[\log \frac{q(S_d|\varphi_d)}{f(S_d|p_d)}\right] - \mathbb{E}_{q(Z_0|\mathbf{X}, \mathbf{Y})} \left[\log \frac{q(Z_0|\mathbf{X}, \mathbf{Y})}{f(Z_0|\theta)}\right] - \sum_t \mathbb{E}_{q(Z_{t-1,t}|\mathbf{X}, \mathbf{Y})} \left[\log \frac{q(Z_t|Z_{t-1}, \mathbf{X}, \mathbf{Y})}{f(Z_t|Z_{t-1}, A)}\right]$$

$$= \mathbb{E}_q \left[\log f(\mathbf{X}|\mathbf{Z}, \mathbf{S}, B, \pi)\right] + \mathbb{E}_q \left[\log f(\mathbf{Y}|\mathbf{Z}, \eta^\mu, \eta^V)\right] + \mathbb{E}_q \left[\log f(\mathbf{Z}|\theta, A)\right] + \mathbb{E}_q \left[\log f(\mathbf{S}|p)\right]$$

$$- \mathbb{E}_q \left[\log q(\mathbf{S}|\varphi)\right] - \mathbb{E}_q \left[\log q(\mathbf{Z}|\mathbf{X}, \mathbf{Y}, \varphi, \Theta)\right]$$

### B.2.1 $\varphi$ update

Relevant terms:

$$L_\varphi = \mathbb{E}_q \left[\log f(\mathbf{X}|\mathbf{Z}, \mathbf{S}, B, \pi)\right] + \mathbb{E}_q \left[\log f(\mathbf{S}|p)\right] - \mathbb{E}_q \left[\log q(\mathbf{S}|\varphi)\right]$$

$$= \sum_{n=1}^{N} \sum_{t=1}^{T_n} \sum_{d=1}^{D} \varphi_d \mathbb{E}_{q(Z_t)} \left[\log P_B(X_{td}^n|Z_t, B)\right] + (1 - \varphi_d) \log P_\pi(X_{td}^n|\pi) + \varphi_d \log \frac{p}{\varphi_d} + (1 - \varphi_d) \log \frac{1-p}{1-\varphi_d}$$

The gradient of the loss terms:

$$\nabla_{\varphi_{d'}} L_\varphi = \sum_{n=1}^{N} \sum_{t=1}^{T_n} \mathbb{E}_{q(Z_t)} \left[\log P_B(X_{td'}^n|Z_t, B)\right] - \log P_\pi(X_{td'}^n|\pi) + \log \frac{p}{1-p} - \log \frac{\varphi_{d'}}{1-\varphi_{d'}}$$

Update:

$$\varphi_d \leftarrow \sigma \left(\log \frac{p}{1-p} + \frac{\sum_{n=1}^{N} \sum_{t=1}^{T_n} \mathbb{E}_{q(Z_t^n|\mathbf{X}, \mathbf{Y}, \varphi)} \left[\log P_B(X_{td}^n|Z_t^n, B) - \log P_\pi(X_{td}^n|\pi)\right]}{\sum_{n=1}^{N} T_n}\right)$$

### B.2.2 $B$ update

$\underline{\mu^B}$

Relevant terms:

$$L_{\mu^B} = \sum_{n=1}^{N} \sum_{t=1}^{T_n} \sum_{d=1}^{D} \varphi_d \underset{q(Z_t^n)}{\mathbb{E}} \left[\log P_B(X_{td}^n | Z_t^n, B)\right]$$

$$= \sum_{n=1}^{N} \sum_{t=1}^{T_n} \sum_{d=1}^{D} \varphi_d \underset{q(Z_t^n)}{\mathbb{E}} \left[\frac{-1}{2\sigma^B{}_{Z_t^n,d}^2} \left(X_{td}^n - \mu_{Z_t^n,d}^B\right)^2 + c\right]$$

The gradient of the loss terms:

$$\nabla_{\mu_{j,d'}^B} L_{\mu^B} = \sum_{n=1}^{N} \sum_{t=1}^{T_n} \varphi_{d'} \nabla_{\mu_{j,d'}^B} \sum_{k=1}^{K} q(Z_t^n = k) \cdot \frac{-1}{2\sigma^B{}_{k,d'}^2} \left(X_{td'}^n - \mu_{k,d}^B\right)^2$$

$$= \sum_{n=1}^{N} \sum_{t=1}^{T_n} \varphi_{d'} q(Z_t^n = j) \cdot \frac{\left(X_{td'}^n - \mu_{j,d'}^B\right)}{\sigma^B{}_{k,d'}^2}$$

Update:

$$\mu_{j,d'}^B \leftarrow \frac{\sum_{n=1}^{N} \sum_{t=1}^{T_n} q(Z_t^n = j) X_{td'}^n}{\sum_{n=1}^{N} \sum_{t=1}^{T_n} q(Z_t^n = j)}$$

where the $q(Z_t^n = j)$ term is computed using the forward-backward algorithm.

$\underline{\sigma^{B2}}$

Relevant terms:

$$L_{\sigma^{B2}} = \sum_{n=1}^{N} \sum_{t=1}^{T_n} \sum_{d=1}^{D} \varphi_d \underset{q(Z_t^n)}{\mathbb{E}} \left[-\frac{1}{2} \log \sigma^B{}_{Z_t^n,d}^2 - \frac{1}{2\sigma^B{}_{Z_t^n,d}^2} \left(X_{td}^n - \mu_{Z_t^n,d'}^B\right)^2 + c\right]$$

Gradient of the loss terms:

$$\nabla_{\sigma^{B2}_{j,d'}} L_{\sigma^{B2}} = \sum_{n=1}^{N} \sum_{t=1}^{T_n} \varphi_{d'} q(Z_t^n = j) \left(-\frac{1}{2\sigma^B{}_{j,d'}^2} + \frac{1}{2\left(\sigma^B{}_{j,d'}^2\right)^2} \left(X_{td'}^n - \mu_{j,d'}^B\right)^2\right)$$

Update (if $\varphi_{d'} \neq 0$):

$$\sigma^B{}_{j,d'}^2 \leftarrow \frac{\sum_{n=1}^{N} \sum_{t=1}^{T_n} q(Z_t^n = j) \left(X_{td'}^n - \mu_{j,d'}^B\right)^2}{\sum_{n=1}^{N} \sum_{t=1}^{T_n} q(Z_t^n = j)}$$

where the $q(Z_t^n = j)$ term is computed using the forward-backward algorithm.

### B.2.3 $\pi$ update

$\underline{\mu^\pi}$

Relevant terms:

$$L_{\mu^\pi} = \sum_{n=1}^{N} \sum_{t=1}^{T_n} \sum_{d=1}^{D} (1 - \varphi_d) \log P_\pi(X_{td}|\pi)$$

$$= \sum_{n=1}^{N} \sum_{t=1}^{T_n} \sum_{d=1}^{D} (1 - \varphi_d) \frac{-1}{2\sigma^\pi{}_d^2} (X_{td}^n - \mu_d^\pi)^2 + c$$

The gradient of the loss terms:

$$\nabla_{\mu_{d'}^\pi} L_{\mu^\pi} = \sum_{n=1}^{N} \sum_{t=1}^{T_n} (1 - \varphi_{d'}) \cdot \frac{(X_{td'}^n - \mu_{d'}^\pi)}{\sigma^\pi{}_{d'}^2}$$

Update (if $\varphi_{d'} \neq 1$):

$$\mu_{d'}^\pi \leftarrow \frac{\sum_{n=1}^{N} \sum_{t=1}^{T_n} X_{td'}^n}{\sum_{n=1}^{N} \sum_{t=1}^{T_n} 1}$$

$$= \frac{\sum_{n=1}^{N} \sum_{t=1}^{T_n} X_{td'}^n}{\sum_{n=1}^{N} T_n}$$

$\underline{\sigma^{\pi 2}}$

Relevant terms:

$$L_{\sigma^{\pi 2}} = \sum_{n=1}^{N} \sum_{t=1}^{T_n} \sum_{d=1}^{D} (1 - \varphi_d) \left( -\frac{1}{2} \log \sigma^\pi{}_d^2 - \frac{1}{2\sigma^\pi{}_d^2} (X_{td}^n - \mu_{d'}^\pi)^2 + c \right)$$

The gradient of the loss terms:

$$\nabla_{\sigma^\pi{}_{d'}^2} L_{\sigma^{\pi 2}} = \sum_{n=1}^{N} \sum_{t=1}^{T_n} (1 - \varphi_{d'}) \left( -\frac{1}{2\sigma^\pi{}_{d'}^2} + \frac{1}{2(\sigma^\pi{}_{d'}^2)^2} (X_{td'}^n - \mu_{d'}^\pi)^2 \right)$$

Update:

$$\sigma^\pi{}_{d'}^2 \leftarrow \frac{\sum_{n=1}^{N} \sum_{t=1}^{T_n} (X_{td'}^n - \mu_{d'}^\pi)^2}{\sum_{n=1}^{N} \sum_{t=1}^{T_n} 1}$$

$$= \frac{\sum_{n=1}^{N} \sum_{t=1}^{T_n} (X_{td'}^n)^2}{\sum_{n=1}^{N} T_n} - (\mu^\pi)^2$$

### B.2.4 $\theta$ update

Relevant terms (including the constraint and the regularizer):

$$L_\theta = \log f(\theta|\alpha) + \left\{ \sum_{n=1}^{N} \sum_{k=1}^{K} q(Z_1^n = k) \log f(Z_1^n = k|\theta) \right\} - \lambda_\theta \left( \sum_{k=1}^{K} \theta_k - 1 \right)$$

The gradient of the loss terms:

$$\nabla_{\theta_{k'}} L_\theta = -\lambda_\theta + (\alpha - 1) \frac{1}{\theta_{k'}} + \sum_{n=1}^{N} q(Z_1^n = k') \frac{1}{\theta_{k'}}$$

$$\nabla_{\lambda_\theta} L_\theta = -\sum_{k=1}^{K} \theta_k$$

Update:

$$\theta_{k'} \leftarrow \frac{\alpha - 1 + \sum_{n=1}^{N} q(Z_1^n = k')}{K\alpha - K + N}$$
$$= \frac{\alpha - 1 + \mathbb{E}\left[N_{k'}^1\right]}{K\alpha - K + N}$$

where the terms $q(Z_1^n = k')$ are computed using the forward-backward algorithm.

### B.2.5  $A$ **update**

Relevant terms:

$$L_A = \left\{ \sum_{n=1}^{N} \sum_{t=2}^{T_n} \sum_{j=1}^{J} \sum_{k=1}^{K} q(Z_{t-1}^n = j, Z_t^n = k) \log f(Z_t^n = k | Z_{t-1}^n = j, A) \right\} - \sum_{l=1}^{K} \lambda_{A_l} \left( \sum_{k=1}^{K} A_{lk} - 1 \right)$$

The gradient of the loss terms:

$$\nabla_{A_{j',k'}} L_A = -\lambda_{A_{j'}} + \sum_{n=1}^{N} \sum_{t=2}^{T_n} q(Z_{t-1}^n = j', Z_t^n = k') \frac{1}{A_{j',k'}}$$

$$\nabla_{\lambda_{A_{l'}}} L_A = -\sum_{k=1}^{K} A_{l'k}$$

Update:

$$A_{j',k'} \leftarrow \frac{\sum_{n=1}^{N} \sum_{t=2}^{T_n} q(Z_{t-1}^n = j', Z_t^n = k')}{\sum_{n=1}^{N} \sum_{t=2}^{T_n} \sum_{k=1}^{K} q(Z_{t-1}^n = j', Z_t^n = k)} = \frac{\sum_{n=1}^{N} \sum_{t=2}^{T_n} q(Z_{t-1}^n = j', Z_t^n = k')}{\sum_{n=1}^{N} \sum_{t=2}^{T_n} q(Z_{t-1}^n = j')}$$
$$= \frac{\mathbb{E}\left[N_{j',k'}\right]}{\sum_{k'=1}^{K} \mathbb{E}\left[N_{j',k'}\right]}$$

where the terms $q(Z_{t-1}^n = j', Z_t^n = k')$ are computed using the forward-backward algorithm.

### B.3  Gibbs Sampling Conditional Distributions

While variational EM is quite stable when working with diagonal covariance matrices for $X$ in $f_\pi$ and $f_k$, we require Gibbs sampling if we are working with full covariance matrices:

$$f(Z^{(n)} | Z^{(-n)}, \mathbf{X}, \mathbf{Y}, \mathbf{S}) \propto f(Z_n) f(Y^{(n)} | Z^{(n)}) f_k(\mathbf{X}_{\mathbf{S}}^{(n)} | Z^{(n)})$$

$$f(S_d | \mathbf{S}_{-d}, \mathbf{X}, \mathbf{Y}, \mathbf{Z}) = \frac{f(S_d) \prod_{n=1}^{N} f_\pi(\mathbf{X}_{1-\mathbf{S}}^{(n)} | S_d, \mathbf{S}_{-d}) f_k(\mathbf{X}_{\mathbf{S}}^{(n)} | Z^{(n)}, S_d, \mathbf{S}_{-d})}{\sum_{s \in \{0,1\}} f(S_d = s) \prod_{n=1}^{N} f_\pi(\mathbf{X}_{1-\mathbf{S}}^{(n)} | S_d = s, \mathbf{S}_{-d}) f_k(\mathbf{X}_{\mathbf{S}}^{(n)} | Z^{(n)}, S_d = s, \mathbf{S}_{-d})}$$

# C    Derivations for Section 6 (Analysis)

## C.1    Preliminaries

We first state some useful notation and results that we will use in the proofs.

1. We use $\phi(\cdot; \mu, \Sigma)$ to denote the density of a multivariate Gaussian distribution with mean $\mu$ and covariance $\Sigma$.

2. We use $f_{Model}(\cdot)$ to denote the density of the data under a model $Model$.

3. We use an asterisk superscript to denote the *true / data-distributed* quantities (e.g. $f^*(\cdot)$ for data pdf, $\theta^*, \mu^*, \Sigma^*$ for true parameters etc.)

4. We use $\mathbf{X}_{\mathrm{R}} \in \mathbb{R}^{D_{\mathrm{R}}}$ and $\mathbf{X}_{\mathrm{Ir}} \in \mathbb{R}^{D_{\mathrm{Ir}}}$ to denote the relevant and irrelevant components of the data $\mathbf{X} \in \mathbb{R}^D$ respectively. Similarly, we use similar notation for $\mu_{\mathrm{R}}, \mu_{\mathrm{Ir}}, \Sigma_{\mathrm{R}}, \Sigma_{\mathrm{Ir}}$.

**Data Density**    First, we write the density of the data $([\mathbf{X}_{\mathrm{R}}; \mathbf{X}_{\mathrm{Ir}}], Y)$. In our setup, we assume that $(\mathbf{X}_{\mathrm{R}}, Y)$ and $\mathbf{X}_{\mathrm{Ir}}$ are distributed as *independent, equal* mixtures with $K$ and $L$ components respectively:

$$f^*(\mathbf{X}, Y) = \left[ \sum_{k=1}^{K} \frac{1}{K} f^*_{\mathrm{R},k}(\mathbf{X}_{\mathrm{R}}) f^*_{Y,k}(Y) \right] \left[ \sum_{l=1}^{L} \frac{1}{L} f^*_{\mathrm{Ir},l}(\mathbf{X}_{\mathrm{Ir}}) \right] = \frac{1}{KL} \sum_{k=1}^{K} \sum_{l=1}^{L} f^*_{Y,k}(Y) f^*_{\mathrm{R},k}(\mathbf{X}_{\mathrm{R}}) f^*_{\mathrm{Ir},l}(\mathbf{X}_{\mathrm{Ir}})$$

Note that any of the components can be indexed as $m_{kl} = (k, l)$ where $k$ and $l$ are the indices of the 'relevant' and 'irrelevant' components respectively:

$$f^*(\mathbf{X}, Y) = \frac{1}{KL} \sum_{k=1}^{K} \sum_{l=1}^{L} f^*_{m_{kl}}(\mathbf{X}, Y)$$

We will use this notation in the proofs.

We assume that each of the input data components $\mathbf{X}_{\mathrm{R}}$ and $\mathbf{X}_{\mathrm{Ir}}$ are Multivariate Normal variables and the label $Y$ is a Bernoulli variable:

1. $f^*_{\mathrm{R},k}(\cdot) = \phi(\cdot; \mu^*_{\mathrm{R},k}, \Sigma^*_{\mathrm{R},k})$

2. $f^*_{\mathrm{Ir},k}(\cdot) = \phi(\cdot; \mu^*_{\mathrm{Ir},k}, \Sigma^*_{\mathrm{Ir},k})$

3. $f^*_{Y,k}(\cdot) = \mathbf{Bern}(\cdot; \eta^*_k)$

**Lemma C.1.** *To aid in the proofs, we state the following useful results about the expectation of a function $g$ under the data density $f^*$:*

$$\underset{f^*}{\mathbb{E}} [g(\boldsymbol{X}, Y)] = \frac{1}{KL} \sum_{k=1}^{K} \sum_{l=1}^{L} \underset{f^*_{R,k} f^*_{Ir,l} f^*_{Y,k}}{\mathbb{E}} [g(\boldsymbol{X}, Y)] \tag{33}$$

$$\underset{f^*}{\mathbb{E}} [g(\boldsymbol{X})] = \frac{1}{KL} \sum_{k=1}^{K} \sum_{l=1}^{L} \underset{f^*_{R,k} f^*_{Ir,l}}{\mathbb{E}} [g(\boldsymbol{X})] \tag{34}$$

$$\underset{f^*}{\mathbb{E}} [g(\boldsymbol{X}_R)] = \frac{1}{K} \sum_{k=1}^{K} \underset{f^*_{R,k}}{\mathbb{E}} [g(\boldsymbol{X}_R)] \tag{35}$$

$$\underset{f^*}{\mathbb{E}} [g(\boldsymbol{X}_{Ir})] = \frac{1}{L} \sum_{l=1}^{L} \underset{f^*_{Ir,l}}{\mathbb{E}} [g(\boldsymbol{X}_{Ir})] \tag{36}$$

$$\underset{f^*}{\mathbb{E}} [g(Y)] = \frac{1}{K} \sum_{k=1}^{K} \underset{f^*_{Y,k}}{\mathbb{E}} [g(Y)] \tag{37}$$

**Lemma C.2.** *For multivariate Gaussian distributions* $Q(\boldsymbol{X}) = \phi(\boldsymbol{X}; \mu_Q, \Sigma_Q)$ *and* $f(\boldsymbol{X}) = \phi(\boldsymbol{X}; \mu_P, \Sigma_P)$, *the density* $\phi$, *the entropy* $H(Q)$, *cross-entropy* $H(Q, P)$ *and KL-divergence* $\boldsymbol{KL}(Q||P)$ *are:*

$$\phi(\boldsymbol{X}; \mu_Q, \Sigma_Q) = \frac{1}{\sqrt{(2\pi)^k |\Sigma_Q|}} \exp\left[-\frac{1}{2}(\boldsymbol{X} - \mu_Q)^T \Sigma_Q^{-1}(\boldsymbol{X} - \mu_Q)\right] \tag{38}$$

$$H(Q) = \frac{k}{2}\ln(2\pi) + \frac{k}{2} + \frac{1}{2}\ln|\Sigma_Q| \tag{39}$$

$$H(Q, P) = \frac{1}{2}\left[(\mu_P - \mu_Q)^T \Sigma_P^{-1}(\mu_P - \mu_Q) + k\ln(2\pi) + \ln|\Sigma_P| + Tr\left(\Sigma_P^{-1}\Sigma_Q\right)\right] \tag{40}$$

$$\boldsymbol{KL}(Q||P) = H(Q, P) - H(Q) \tag{41}$$

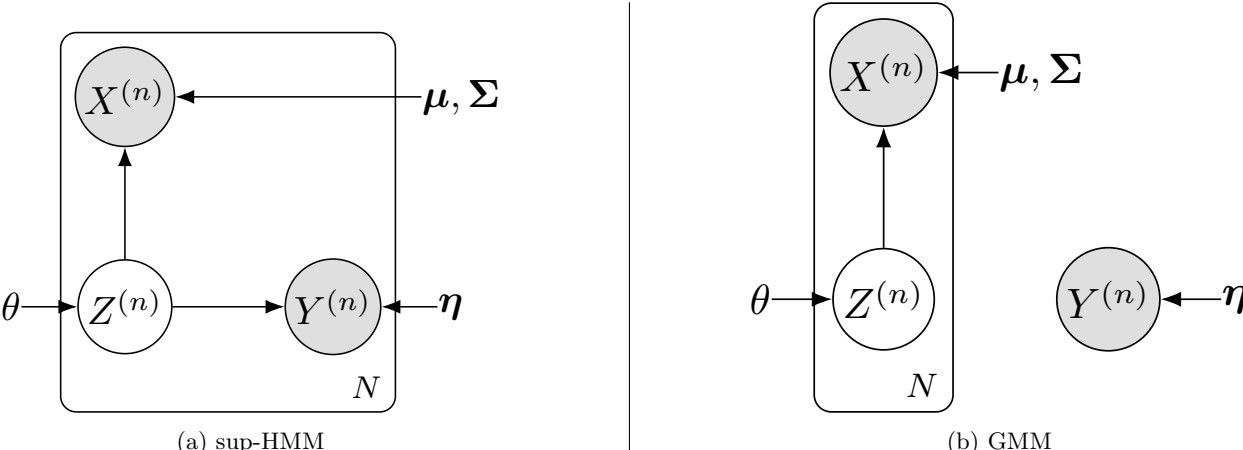

(a) sup-HMM

(b) GMM

Figure 8: Graphical models for the GMM and sup-HMM models.

## C.2 sup-GMM

We have the same setup as in Section 6 with the exception that we don't restrict the data and model to isotropic Gaussians for now. We first state our assumptions about the data first and then derive the population log-likelihoods for sup-GMM and pf-GMM. We will use the sup-GMM case to derive the GMM case.

**Assumption: Model and Density**  We assume that the density of the data under the sup-GMM model is an *equal* mixture of Gaussians with $J$ components (see Figure 8a for the graphical model)

$$Z \sim \mathbf{Cat}(1/J),$$
$$\mathbf{X}|Z = j \sim \mathcal{N}(\mu_j, \Sigma_j),$$
$$Y|Z = j \sim \mathbf{Bern}(\eta_j)$$

The model parameters are $\Theta = \{\mu_j, \Sigma_j, \eta_j\}_{j=1}^{J}$. The density of the data under the model is:

$$f(\mathbf{X}, Y) = \sum_{j=1}^{J} f(Z = j)f(\mathbf{X}, Y|Z = j) = \sum_{j=1}^{J} \frac{1}{J} f(\mathbf{X}, Y|Z = j)$$

**Assumption: Well-separated components**  We assume that the components of the mixture distributions are well-separated. The main reason we require this is to simplify the analysis by allowing the posterior cluster-identity distribution $f(Z|\mathbf{X}, Y)$ to be either 0 or 1 for all $(\mathbf{X}, Y)$. An important consequence of well-separated components is that each data point is assigned to exactly one model component with probability 1. Therefore, each data component will fall under exactly one model component. We will use this fact in the proof of the population log-likelihood below.

**Re-indexing data components**  Due to our "well-separated components" assumption, each data component will fall under exactly one model component. If we call the set of data components that fall under model component $j$ as $M_j$, i.e.

$$M_j = \{m_{kl}| \text{ data component } m_{kl} \text{ falls under model component } j\},$$

then we can partition the set of data components $\{m_{kl}\}$ into $\{M_j|j \in \{1, \ldots, J\}\}$.

**Lemma C.3.** *Assume that the data is generated with density $f^*(\mathbf{X}, Y)$ specified by Equation C.1 and the model is a sup-GMM with $J$ components. Then, the population log-likelihood for the parameters $\Theta_{supGMM}$*

*is:*

$$l(\Theta_{supGMM}) := \underset{f^*}{\mathbb{E}}\left[\log f(\boldsymbol{X}, Y)\right] = \log\frac{1}{J} + l_{\boldsymbol{X}}(\Theta_{supGMM}) + l_Y(\Theta_{supGMM}) + \underset{f^*}{\mathbb{E}}\left[H[Z|\boldsymbol{X}, Y]\right] \tag{42}$$

*where*

$$l_{\boldsymbol{X}}(\Theta_{supGMM}) = \frac{1}{KL}\sum_{j=1}^{J}\sum_{(k,l)\in M_j}\underset{f^*_{m_{kl}}}{\mathbb{E}}\left[\log f_{\boldsymbol{X},j}(\boldsymbol{X};\Theta_j)\right]$$

$$l_Y(\Theta_{supGMM}) = \frac{1}{KL}\sum_{j=1}^{J}\sum_{(k,l)\in M_j}\underset{f^*_{Y,k}}{\mathbb{E}}\left[\log f_j(Y;\Theta_j)\right]$$

$$H[Z|\boldsymbol{X}, Y] := -\underset{f(Z|\boldsymbol{X},Y)}{\mathbb{E}}\left[\log f(Z|\boldsymbol{X}, Y)\right]$$

*Proof.* We derive a general expression for the population log-likelihood with parameters $\Theta_{\text{supGMM}}$. For brevity, we omit $\Theta_{\text{supGMM}}$ in the density functions.

$$l(\Theta_{\text{supGMM}}) := \underset{f^*}{\mathbb{E}}\left[\log f(\mathbf{X}, Y)\right]$$

$$\overset{a}{=} \underset{f^*}{\mathbb{E}}\left[\underset{f(Z|\mathbf{X},Y)}{\mathbb{E}}\left[\log f(\mathbf{X}, Y)\right]\right]$$

$$\overset{b}{=} \underset{f^*}{\mathbb{E}}\left[\underset{f(Z|\mathbf{X},Y)}{\mathbb{E}}\left[\log f(\mathbf{X}, Y, Z) - \log f(Z|\mathbf{X}, Y)\right]\right]$$

$$\overset{c}{=} \underset{f^*}{\mathbb{E}}\left[\underset{f(Z|\mathbf{X},Y)}{\mathbb{E}}\left[\log f(Z) + \log f(\mathbf{X}|Z) + \log f(Y|Z)\right] - \underset{f(Z|\mathbf{X},Y)}{\mathbb{E}}\left[\log f(Z|\mathbf{X}, Y)\right]\right]$$

$$\overset{d}{=} \underset{f^*}{\mathbb{E}}\left[\underset{f(Z|\mathbf{X},Y)}{\mathbb{E}}\left[\log\frac{1}{J} + \log f(\mathbf{X}|Z) + \log f(Y|Z)\right] + H[Z|\mathbf{X}, Y]\right]$$

$$= \log\frac{1}{J} + + \underbrace{\underset{f^*}{\mathbb{E}}\left[\sum_{j=1}^{J} q_j(\mathbf{X}, Y)\log f_{\mathbf{X},j}(\mathbf{X};\Theta_j)\right]}_{l_{\mathbf{X}}(\Theta_{\text{supGMM}})} + \underbrace{\underset{f^*}{\mathbb{E}}\left[\sum_{j=1}^{J} q_j(\mathbf{X}, Y)\log P_j(Y;\Theta_j)\right]}_{l_Y(\Theta_{\text{supGMM}})} + \underset{f^*}{\mathbb{E}}\left[H[Z|\mathbf{X}, Y]\right]$$

$$= \log\frac{1}{J} + l_{\mathbf{X}}(\Theta_{\text{supGMM}}) + l_Y(\Theta_{\text{supGMM}}) + \underset{f^*}{\mathbb{E}}\left[H[Z|\mathbf{X}, Y]\right]$$

where $q_j(\mathbf{X}, Y) := f(Z = j|\mathbf{X}, Y)$ and $\mathbb{E}_{f(Z|\mathbf{X},Y)}\left[\log f(Z|\mathbf{X}, Y)\right] := -H[Z|\mathbf{X}, Y]$. In the above derivation, step (a) follows from the fact that the expectation of a constant doesn't change, step (b) follows from the fact that $f(X, Y) = f(X, Y, Z)/f(Z|X, Y)$, step (c) follows from the factorization of the model and step (d) follows from the fact that $f(Z) = 1/J$.

Expanding the $l_{\mathbf{X}}$ term:

$$l_{\mathbf{X}}(\Theta_{\text{supGMM}}) = \underset{f^*}{\mathbb{E}}\left[\sum_{j=1}^{J} q_j(\mathbf{X}, Y)\log f_{\mathbf{X},j}(\mathbf{X};\Theta_j)\right]$$

$$= \frac{1}{KL}\sum_{k=1}^{K}\sum_{l=1}^{L}\underset{f^*_{m_{kl}}}{\mathbb{E}}\left[\sum_{j=1}^{J} q_j(\mathbf{X}, Y)\log f_{\mathbf{X},j}(\mathbf{X};\Theta_j)\right]$$

$$= \frac{1}{KL}\sum_{j=1}^{J}\sum_{m_{kl}\in M_j}\underset{f^*_{m_{kl}}}{\mathbb{E}}\left[\log f_{\mathbf{X},j}(\mathbf{X};\Theta_j)\right]$$

Similarly, we can expand the $l_Y(\Theta_{\text{supGMM}})$ term:

$$
\begin{aligned}
l_Y(\Theta_{\text{supGMM}}) &= \mathop{\mathbb{E}}_{f^*}\left[\sum_{j=1}^{J} q_j(\mathbf{X}, Y)\log f_{Y,j}(Y;\Theta_j)\right] \\
&= \frac{1}{KL}\sum_{j=1}^{J}\sum_{m_{kl}\in M_j}\mathop{\mathbb{E}}_{f^*_{m_{kl}}}\left[q_j(\mathbf{X}, Y)\log f_{Y,j}(Y;\Theta_j)\right] \\
&= \frac{1}{KL}\sum_{j=1}^{J}\sum_{m_{kl}\in M_j}\mathop{\mathbb{E}}_{f^*_{m_{kl}}}\left[\log f_{Y,j}(Y;\Theta_j)\right]
\end{aligned}
$$

$\square$

### C.2.1 Special case: $J = 2, K = 2, L = 2$

We now consider the special case where the data is generated with hyperparameters $K = L = 2$ and the sup-GMM model has $J = 2$ components. This means that the data density is a mixture of 4 components, while the model has only 2 components. We further assume that true parameters for $Y$ components are $\eta_1^* = 0, \eta_2^* = 1$. Using this case, we determine the conditions that favor relevant components over irrelevant ones.

**Theorem C.4.** *Assume that the data is generated with density $f^*(\boldsymbol{X}, Y)$ and parameters $\Theta^*_{supGMM} = \left\{\mu_j^*, \Sigma_j^*, \eta_j^*\right\}_{j=1}^{J}$ specified by Equation C.1, and has hyperparameters $K = L = 2$. The sup-GMM model with $J = 2$ components, learns parameters $\Theta_{supGMM} = \left\{\hat{\mu}_j, \hat{\Sigma}_j, \hat{\eta}_j\right\}_{j=1}^{J}$. We define $\Theta^a_{supGMM}$ as the parameters the sup-GMM the model learns when its components are aligned with data's relevant components, and $\Theta^u_{supGMM}$ when unaligned. Then, the population log-likelihood for the parameters $\Theta^a_{supGMM}$ and $\Theta^u_{supGMM}$ is:*

$$
l(\Theta^a_{supGMM}) = \log 1/2 + -\frac{1}{4}\sum_{k=1}^{2}\sum_{l=1}^{2}H\left(\phi(\mu^*_{R,k}, \Sigma^*_{R,k})\right) + H\left(\phi(\mu^*_{Ir,l}, \Sigma^*_{Ir,l})\right) + \boldsymbol{KL}\left(\phi(\mu^*_{Ir,l}, \Sigma^*_{Ir,l})||\phi(\bar{\mu}_{Ir}, \bar{\Sigma}_{Ir})\right)
$$

$$
l(\Theta^u_{supGMM}) = \log 1/2 + -\log 2 - \frac{1}{4}\sum_{k=1}^{2}\sum_{l=1}^{2}\left(H\left(\phi(\mu^*_{Ir,l}, \Sigma^*_{Ir,l})\right) + H\left(\phi(\mu^*_{R,k}, \Sigma^*_{R,k})\right) + \boldsymbol{KL}\left(\phi(\mu^*_{R,k}, \Sigma^*_{R,k})||\phi(\bar{\mu}_R, \bar{\Sigma}_R)\right)\right)
$$

*Furthermore, the difference in the population log-likelihoods between the aligned and unaligned cases is:*

$$
l(\Theta^a_{supGMM}) - l(\Theta^u_{supGMM}) = \log 2 + \frac{1}{4}\sum_{k=1}^{2}\sum_{l=1}^{2}\left(\boldsymbol{KL}\left(\phi(\mu^*_{R,k}, \Sigma^*_{R,k})||\phi(\bar{\mu}_R, \bar{\Sigma}_R)\right) - \boldsymbol{KL}\left(\phi(\mu^*_{Ir,l}, \Sigma^*_{Ir,l})||\phi(\bar{\mu}_{Ir}, \bar{\Sigma}_{Ir})\right)\right)
$$

*Proof.*

**Aligned case ($\Theta^a_{\text{supGMM}}$)** Aligned parameters $\Theta^a_{\text{supGMM}}$ correspond to the case when the model components are aligned with the relevant components of the data. Using this information, we first write out the

|  | Description | Values | | | |
|---|---|---|---|---|---|
| $j$ | Model component index | $j = 1$ | | $j = 2$ | |
| $(k, l) \in M_j$ | Data component index assigned to model component $j$ | $k = 1, l = 1$ | $k = 1, l = 2$ | $k = 2, l = 1$ | $k = 2, l = 2$ |
| **Aligned Parameters**, $\Theta^a_{\text{supGMM}}$ | | | | | |
| $f_{\text{R},j} = f^*_{\text{R},k}$ | Learned model density for $\mathbf{X}_\text{R}$ | $f^*_{\text{R},1}$ | $f^*_{\text{R},1}$ | $f^*_{\text{R},2}$ | $f^*_{\text{R},2}$ |
| $f_{\text{Ir},j} = \phi(\bar{\mu}_{\text{Ir}}, \bar{\Sigma}_{\text{Ir}})$ | Learned model density for $\mathbf{X}_\text{Ir}$ | $\phi(\bar{\mu}_{\text{Ir}}, \bar{\Sigma}_{\text{Ir}})$ | $\phi(\bar{\mu}_{\text{Ir}}, \bar{\Sigma}_{\text{Ir}})$ | $\phi(\bar{\mu}_{\text{Ir}}, \bar{\Sigma}_{\text{Ir}})$ | $\phi(\bar{\mu}_{\text{Ir}}, \bar{\Sigma}_{\text{Ir}})$ |
| $f_{Y,j} = f^*_{Y,k}$ | Learned model density for $Y$ | $f^*_{Y,1}$ | $f^*_{Y,1}$ | $f^*_{Y,2}$ | $f^*_{Y,2}$ |
| **Unaligned Parameters**, $\Theta^u_{\text{supGMM}}$ | | | | | |
| $f_{\text{R},j} = \phi(\bar{\mu}_{\text{R}}, \bar{\Sigma}_{\text{R}})$ | Learned model density for $\mathbf{X}_\text{R}$ | $\phi(\bar{\mu}_{\text{R}}, \bar{\Sigma}_{\text{R}})$ | $\phi(\bar{\mu}_{\text{R}}, \bar{\Sigma}_{\text{R}})$ | $\phi(\bar{\mu}_{\text{R}}, \bar{\Sigma}_{\text{R}})$ | $\phi(\bar{\mu}_{\text{R}}, \bar{\Sigma}_{\text{R}})$ |
| $f_{\text{Ir},j} = f^*_{\text{Ir},l}$ | Learned model density for $\mathbf{X}_\text{Ir}$ | $f^*_{\text{Ir},1}$ | $f^*_{\text{Ir},2}$ | $f^*_{\text{Ir},1}$ | $f^*_{\text{Ir},2}$ |
| $f_{Y,j} = \mathbf{Bern}(0.5)$ | Learned model density for $Y$ | $\mathbf{Bern}(0.5)$ | $\mathbf{Bern}(0.5)$ | $\mathbf{Bern}(0.5)$ | $\mathbf{Bern}(0.5)$ |

Table 2: sup-GMM: Summary of the learned model densities for the aligned and unaligned parameter cases.

individual terms in $\Theta^a_{\text{supGMM}} = \left\{ \hat{\mu}^a_j, \hat{\Sigma}^a_j, \hat{\eta}^a_j \right\}^2_{j=1}$:

$$\hat{\mu}^a_j = \begin{bmatrix} \mu^*_{\text{R},j} \\ \bar{\mu}_{\text{Ir}} \end{bmatrix} \qquad \text{for } j \in \{1, 2\} \quad \text{where} \qquad \bar{\mu}_{\text{Ir}} = \frac{1}{L} \sum_{l=1}^2 \mu^*_{\text{Ir},l} \tag{43}$$

$$\hat{\Sigma}^a_j = \begin{bmatrix} \Sigma^*_{\text{R},j} & 0 \\ 0 & \bar{\Sigma}_{\text{Ir}} \end{bmatrix} \qquad \text{for } j \in \{1, 2\} \quad \text{where} \qquad \bar{\Sigma}_{\text{Ir}} = \frac{1}{L} \left( \sum_{l=1}^2 \mu^{*\top}_{\text{Ir},l} \mu^*_{\text{Ir},l} + \Sigma^*_{\text{Ir},l} \right) - \bar{\mu}^\top_{\text{Ir}} \bar{\mu}_{\text{Ir}} \tag{44}$$

$$\hat{\eta}^a_j = \eta^*_j \qquad \text{where} \qquad \eta^*_j = \begin{cases} 0 & \text{if } j = 1 \\ 1 & \text{if } j = 2 \end{cases} \tag{45}$$

We first expand the $l_{\mathbf{X}}$ term:

$$l_{\mathbf{X}}(\Theta^a_{\text{supGMM}}) = \frac{1}{KL} \sum_{j=1}^{J} \sum_{(k,l) \in M_j} \mathbb{E}_{f^*_{m_{kl}}} \left[ \log f_{\mathbf{X},j}(\mathbf{X}; \hat{\mu}^a_j, \hat{\Sigma}^a_j) \right]$$

$$= \frac{1}{KL} \sum_{j=1}^{J} \sum_{(k,l) \in M_j} \mathbb{E}_{f^*_{\text{R},k} f^*_{\text{Ir},l}} \left[ \log f_{\text{R},j}(\mathbf{X}_{\text{R}}; \hat{\mu}^a_{\text{R},j}, \hat{\Sigma}^a_{\text{R},j}) + \log f_{\text{Ir},j}(\mathbf{X}_{\text{Ir}}; \hat{\mu}^a_{\text{Ir},j}, \hat{\Sigma}^a_{\text{Ir},j}) \right]$$

$$= \frac{1}{KL} \sum_{j=1}^{J} \sum_{(k,l) \in M_j} \mathbb{E}_{f^*_{\text{R},k}} \left[ \log f_{\text{R},j}(\mathbf{X}_{\text{R}}; \hat{\mu}^a_{\text{R},j}, \hat{\Sigma}^a_{\text{R},j}) \right] + \mathbb{E}_{f^*_{\text{Ir},l}} \left[ \log f_{\text{Ir},j}(\mathbf{X}_{\text{Ir}}; \hat{\mu}^a_{\text{Ir},j}, \hat{\Sigma}^a_{\text{Ir},j}) \right]$$

$$\overset{a}{=} \frac{1}{KL} \sum_{j=1}^{J} \sum_{(k,l) \in M_j} \mathbb{E}_{f^*_{\text{R},k}} \left[ \log f^*_{\text{R},k}(\mathbf{X}_{\text{R}}) \right] + \mathbb{E}_{f^*_{\text{Ir},l}} \left[ \log \phi(\mathbf{X}_{\text{Ir}}; \bar{\mu}_{\text{Ir}}, \bar{\Sigma}_{\text{Ir}}) \right]$$

$$\overset{b}{=} \frac{1}{KL} \sum_{j=1}^{J} \sum_{(k,l) \in M_j} -H(\phi(\mathbf{X}_{\text{R}}; \mu^*_{\text{R},k}, \Sigma^*_{\text{R},k})) - H(\phi(\mathbf{X}_{\text{Ir}}; \mu^*_{\text{Ir},l}, \Sigma^*_{\text{Ir},l}), \phi(\mathbf{X}_{\text{Ir}}; \bar{\mu}_{\text{Ir}}, \bar{\Sigma}_{\text{Ir}}))$$

$$\overset{c}{=} \frac{1}{KL} \sum_{j=1}^{J} \sum_{(k,l) \in M_j} -\text{H}\left(\phi(\mu^*_{\text{R},k}, \Sigma^*_{\text{R},k})\right) - \text{H}\left(\phi(\mu^*_{\text{Ir},l}, \Sigma^*_{\text{Ir},l}), \phi(\bar{\mu}_{\text{Ir}}, \bar{\Sigma}_{\text{Ir}})\right) \pm \text{H}\left(\phi(\mu^*_{\text{Ir},l}, \Sigma^*_{\text{Ir},l})\right)$$

$$\overset{d}{=} \frac{1}{KL} \sum_{j=1}^{J} \sum_{(k,l) \in M_j} -\text{H}\left(\phi(\mu^*_{\text{R},k}, \Sigma^*_{\text{R},k})\right) - \text{H}\left(\phi(\mu^*_{\text{Ir},l}, \Sigma^*_{\text{Ir},l})\right) - \mathbf{KL}\left(\phi(\mu^*_{\text{Ir},l}, \Sigma^*_{\text{Ir},l}) || \phi(\bar{\mu}_{\text{Ir}}, \bar{\Sigma}_{\text{Ir}})\right)$$

$$\overset{e}{=} \frac{1}{4} \sum_{k=1}^{2} \sum_{l=1}^{2} -\text{H}\left(\phi(\mu^*_{\text{R},k}, \Sigma^*_{\text{R},k})\right) - \text{H}\left(\phi(\mu^*_{\text{Ir},l}, \Sigma^*_{\text{Ir},l})\right) - \mathbf{KL}\left(\phi(\mu^*_{\text{Ir},l}, \Sigma^*_{\text{Ir},l}) || \phi(\bar{\mu}_{\text{Ir}}, \bar{\Sigma}_{\text{Ir}})\right)$$

where equation (a) uses the learned model densities expressions from Table 2, (b) uses the entropy and cross-entropy definitions, (c) and (d) add/subtract the entropy term and use the fact that KL-divergence is the difference between cross-entropy and entropy, and (e) uses the fact that the data is generated with hyperparameters $K = L = 2$ and the model has $J = 2$ components, and $\{(k,l) \in M_j | j \in \{1,2\}\} = \{(1,1),(1,2),(2,1),(2,2)\}$.

Next, we expand the $l_Y(\Theta^a_{\text{supGMM}})$ term:

$$l_Y(\Theta^a_{\text{supGMM}}) = \frac{1}{KL} \sum_{j=1}^{J} \sum_{m_{kl} \in M_j} \mathbb{E}_{f^*_{m_{kl}}} \left[ \log f_{Y,j}(Y; \hat{\eta}^a_j) \right]$$

$$= \frac{1}{KL} \sum_{j=1}^{J} \sum_{m_{kl} \in M_j} \mathbb{E}_{f^*_{Y,k}} \left[ \log f^*_{Y,k}(Y) \right]$$

$$= \frac{1}{KL} \sum_{j=1}^{J} \sum_{m_{kl} \in M_j} -H(f^*_{Y,k}(Y))$$

$$= 0$$

The rest of the terms in Equation 42 are:

$$\log \frac{1}{J} = \log \frac{1}{2}$$

$$\mathbb{E}_{f^*} \left[ H[Z | \mathbf{X}, Y] \right] = 0$$

The likelihood for the aligned case is therefore:

$$l(\Theta^a_{\text{supGMM}}) = \log \frac{1}{2} - \frac{1}{4} \sum_{k=1}^{2} \sum_{l=1}^{2} \text{H}\left(\phi(\mu^*_{\text{R},k}, \Sigma^*_{\text{R},k})\right) + \text{H}\left(\phi(\mu^*_{\text{Ir},l}, \Sigma^*_{\text{Ir},l})\right) + \textbf{KL}\left(\phi(\mu^*_{\text{Ir},l}, \Sigma^*_{\text{Ir},l})||\phi(\bar{\mu}_{\text{Ir}}, \bar{\Sigma}_{\text{Ir}})\right)$$

**Unaligned case ($\Theta^u_{\textbf{supGMM}}$)** Unaligned parameters $\Theta^u_{\text{supGMM}}$ correspond to the case when the model components are *unaligned* with the relevant components of the data. Using this information, we first write out the individual terms in $\Theta^u_{\text{supGMM}} = \left\{ \hat{\mu}^u_j, \hat{\Sigma}^u_j, \hat{\eta}^u_j \right\}^2_{j=1}$:

$$\hat{\mu}^u_j = \begin{bmatrix} \bar{\mu}_{\text{R}} \\ \mu^*_{\text{Ir},j} \end{bmatrix} \qquad \text{for } j \in \{1,2\} \quad \text{where} \qquad \bar{\mu}_{\text{R}} = \frac{1}{K} \sum_{k=1}^{2} \mu^*_{\text{R},k} \tag{46}$$

$$\hat{\Sigma}^u_j = \begin{bmatrix} \bar{\Sigma}_{\text{R}} & 0 \\ 0 & \Sigma^*_{\text{Ir},j} \end{bmatrix} \qquad \text{for } j \in \{1,2\} \quad \text{where} \qquad \bar{\Sigma}_{\text{R}} = \frac{1}{K} \left( \sum_{k=1}^{2} \mu^{*\top}_{\text{R},k} \mu^*_{\text{R},k} + \Sigma^*_{\text{R},k} \right) - \bar{\mu}^\top_{\text{R}} \bar{\mu}_{\text{R}} \tag{47}$$

$$\hat{\eta}^u_j = \frac{1}{2} \tag{48}$$

We can similarly expand the $l_{\textbf{X}}$ term as we did for the aligned case:

$$l_{\textbf{X}}(\Theta^u_{\text{supGMM}}) = \frac{1}{KL} \sum_{j=1}^{2} \sum_{(k,l) \in M_j} \underset{f^*_{m_{kl}}}{\mathbb{E}} \left[ \log f_{\textbf{X},j}(\textbf{X}; \hat{\mu}^u_j, \hat{\Sigma}^u_j) \right]$$

$$= \frac{1}{KL} \sum_{j=1}^{2} \sum_{(k,l) \in M_j} -\text{H}\left(\phi(\mu^*_{\text{Ir},l}, \Sigma^*_{\text{Ir},l})\right) - \text{H}\left(\phi(\mu^*_{\text{R},k}, \Sigma^*_{\text{R},k})\right) - \textbf{KL}\left(\phi(\mu^*_{\text{R},k}, \Sigma^*_{\text{R},k})||\phi(\bar{\mu}_{\text{R}}, \bar{\Sigma}_{\text{R}})\right)$$

$$= \frac{1}{4} \sum_{k=1}^{2} \sum_{l=1}^{2} -\text{H}\left(\phi(\mu^*_{\text{Ir},l}, \Sigma^*_{\text{Ir},l})\right) - \text{H}\left(\phi(\mu^*_{\text{R},k}, \Sigma^*_{\text{R},k})\right) - \textbf{KL}\left(\phi(\mu^*_{\text{R},k}, \Sigma^*_{\text{R},k})||\phi(\bar{\mu}_{\text{R}}, \bar{\Sigma}_{\text{R}})\right)$$

We can also expand the $l_Y(\Theta^u_{\text{supGMM}})$ term as we did for the aligned case:

$$l_Y(\Theta^u_{\text{supGMM}}) = \frac{1}{KL} \sum_{j=1}^{J} \sum_{m_{kl} \in M_j} -H(\textbf{Bern}(0.5))$$

$$= -\log 2$$

The rest of the terms in Equation 42 are the same as for the aligned case: $\log \frac{1}{J} = \log \frac{1}{2}$ and $\mathbb{E}_{f^*}\left[H[Z|\textbf{X}, Y]\right] = 0$.

The likelihood for the unaligned case is therefore (we separate the $\log 2$ term to highlight that it is the $l_Y(\Theta^u_{\text{supGMM}})$ term):

$$l(\Theta^u_{\text{supGMM}}) = \log \frac{1}{2} - \log 2 - \frac{1}{4} \sum_{k=1}^{2} \sum_{l=1}^{2} \left( \text{H}\left(\phi(\mu^*_{\text{Ir},l}, \Sigma^*_{\text{Ir},l})\right) + \text{H}\left(\phi(\mu^*_{\text{R},k}, \Sigma^*_{\text{R},k})\right) + \textbf{KL}\left(\phi(\mu^*_{\text{R},k}, \Sigma^*_{\text{R},k})||\phi(\bar{\mu}_{\text{R}}, \bar{\Sigma}_{\text{R}})\right) \right)$$

This gives us the difference in the population log-likelihoods between the aligned and unaligned cases:

$$l(\Theta^a_{\text{supGMM}}) - l(\Theta^u_{\text{supGMM}}) = \log 2 + \frac{1}{4} \sum_{k=1}^{2} \sum_{l=1}^{2} \left( \textbf{KL}\left(\phi(\mu^*_{\text{R},k}, \Sigma^*_{\text{R},k})||\phi(\bar{\mu}_{\text{R}}, \bar{\Sigma}_{\text{R}})\right) - \textbf{KL}\left(\phi(\mu^*_{\text{Ir},l}, \Sigma^*_{\text{Ir},l})||\phi(\bar{\mu}_{\text{Ir}}, \bar{\Sigma}_{\text{Ir}})\right) \right)$$

$\square$

### C.3 GMM - General Case

The general case for GMM is similar to sup-GMM except that $Y$ is not modeled as part of the mixture distribution. Therefore, we state the model density and the theorem for GMM case by dropping the $\log 2$ term since it is present for both aligned and unaligned parameters.

**Assumption: Model and Density** We assume that the density of the data under the GMM model is an *equal* mixture of Gaussians with $J$ components but $Y$ is generated independently (see Figure 8b for the graphical model):

$$Z \sim \mathbf{Cat}(1/J),$$
$$\mathbf{X}|Z = j \sim \mathcal{N}(\mu_j, \Sigma_j),$$
$$Y \sim \mathbf{Bern}(\eta)$$

The density of the data under the model is:

$$f(\mathbf{X}, Y) = f_Y \sum_{j=1}^{J} f(Z = j)f(\mathbf{X}|Z = j) = f_Y \sum_{j=1}^{J} \frac{1}{J} f(\mathbf{X}|Z = j)$$

**Theorem C.5.** *Assume that the data is generated with density $f^*(\mathbf{X}, Y)$ and parameters $\Theta^*_{GMM} = \left\{\mu_j^*, \Sigma_j^*, \eta_j^*\right\}_{j=1}^{J}$ specified by Equation C.1 and the model is a GMM with learnt parameters $\Theta_{GMM} = \left\{\hat{\mu}_j, \hat{\Sigma}_j, \hat{\eta}_j\right\}_{j=1}^{J}$. Further assume that the data is generated with hyperparameters $K = L = 2$ and the model has $J = 2$ components. Define $\Theta^a_{GMM}$ as the parameters that GMM model learns when its components are aligned with the relevant components of the data, and $\Theta^u_{GMM}$ as the parameters that GMM model learns when its components are unaligned with the relevant components of the data. Then, the population log-likelihood for the parameters $\Theta^a_{GMM}$ and $\Theta^u_{GMM}$ is:*

$$l(\Theta^a_{GMM}) = -\frac{1}{4} \sum_{k=1}^{2} \sum_{l=1}^{2} H\left(\phi(\mu^*_{R,k}, \Sigma^*_{R,k})\right) + H\left(\phi(\mu^*_{Ir,l}, \Sigma^*_{Ir,l})\right) + \mathbf{KL}\left(\phi(\mu^*_{Ir,l}, \Sigma^*_{Ir,l})||\phi(\bar{\mu}_{Ir}, \bar{\Sigma}_{Ir})\right)$$

$$l(\Theta^u_{GMM}) = -\log 2 - \frac{1}{4} \sum_{k=1}^{2} \sum_{l=1}^{2} \left(H\left(\phi(\mu^*_{Ir,l}, \Sigma^*_{Ir,l})\right) + H\left(\phi(\mu^*_{R,k}, \Sigma^*_{R,k})\right) + \mathbf{KL}\left(\phi(\mu^*_{R,k}, \Sigma^*_{R,k})||\phi(\bar{\mu}_R, \bar{\Sigma}_R)\right)\right)$$

*Furthermore, the difference in the population log-likelihoods between the aligned and unaligned cases is:*

$$l(\Theta^a_{GMM}) - l(\Theta^u_{GMM}) = \log 2 + \frac{1}{4} \sum_{k=1}^{2} \sum_{l=1}^{2} \left(\mathbf{KL}\left(\phi(\mu^*_{R,k}, \Sigma^*_{R,k})||\phi(\bar{\mu}_R, \bar{\Sigma}_R)\right) - \mathbf{KL}\left(\phi(\mu^*_{Ir,l}, \Sigma^*_{Ir,l})||\phi(\bar{\mu}_{Ir}, \bar{\Sigma}_{Ir})\right)\right)$$

### C.4 pf-GMM - General case

**Assumption: Model and Density**

$$S_d \sim \mathbf{Bern}(p) \quad \forall d, \qquad Z \sim \mathbf{Cat}(1/J),$$
$$Y|Z = j \sim \mathbf{Bern}(\eta_j), \quad \mathbf{X_S}|Z = j \sim \mathcal{N}(\mu_j, \Sigma_j),$$
$$\mathbf{X_{1-S}} \sim \mathcal{N}(\mu^\pi, \Sigma^\pi) \quad \forall n.$$

**Assumption: Well-separated components and point-mass switch posterior**

1. We again assume that the components of the mixture distributions are 'well-separated'.

2. We also assume that the posterior for the switches is a point-mass at one of the values of $\mathbf{S}$, i.e. $f(\mathbf{S} = \mathbf{s}|\mathbf{X}, Y; \Theta_{\text{pfGMM}}) = 1$ for some $\mathbf{s} \in \{0, 1\}^D$. For the parameters we consider, the following are the values of $\mathbf{s}$:

   (a) $f(\mathbf{S} = \mathbf{s}|\mathbf{X}, Y; \Theta^a_{\text{pfGMM}}) = 1$ for $\mathbf{s} = [\mathbf{1}_{D_{\text{R}}}; \mathbf{0}_{D_{\text{Ir}}}]$
   (b) $f(\mathbf{S} = \mathbf{s}|\mathbf{X}, Y; \Theta^a_{\text{pfGMM}}) = 1$ for $\mathbf{s} = [\mathbf{0}_{D_{\text{R}}}; \mathbf{1}_{D_{\text{Ir}}}]$.
   (c) $f(\mathbf{S} = \mathbf{s}|\mathbf{X}, Y; \Theta^{\text{1-G}}_{\text{pfGMM}}) = 1$ for $\mathbf{s} = [\mathbf{0}_{D_{\text{R}}}; \mathbf{0}_{D_{\text{Ir}}}]$.

The density of the data under the model is (see Figure 2a for the graphical model):

$$f(\mathbf{X}, Y) = \sum_{\mathbf{s} \in \{0,1\}^D} \sum_{j=1}^{J} \left[\prod_{d=1}^{D} f_{S_d}(s_d)\right] f_Z(j) f_{\mathbf{X}_{1-\mathbf{s}}}(\mathbf{X}_{1-\mathbf{s}}|\mathbf{S} = \mathbf{s}) f_{\mathbf{X_s}}(\mathbf{X_s}|Z = j, \mathbf{S} = \mathbf{s}) f_Y(Y|Z = j)$$

$$= \frac{1}{J} \sum_{\mathbf{s} \in \{0,1\}^D} \left[\prod_{d=1}^{D} f_{S_d}(s_d)\right] f_{\mathbf{X}_{1-\mathbf{s}}}(\mathbf{X}_{1-\mathbf{s}}|\mathbf{S} = \mathbf{s}) \sum_{j=1}^{J} f_{\mathbf{X_s}}(\mathbf{X_s}|Z = j, \mathbf{S} = \mathbf{s}) f_Y(Y|Z = j)$$

**Lemma C.6.** *Assume that the data is generated with density $f^*(\mathbf{X}, Y)$ specified by Equation C.1 and the model is a pf-GMM with $J$ components. Then, the population log-likelihood for the parameters $\Theta_{pfGMM}$ is:*

$$l(\Theta_{pfGMM}) := \mathbb{E}_{f^*} [\log f(\mathbf{X}, Y)] = \log \frac{1}{J} + l_{\mathbf{S}}(\Theta_{pfGMM}) + l_{\mathbf{X}}(\Theta_{pfGMM}) + l_Y(\Theta_{pfGMM}) + \mathbb{E}_{f^*} [H[\mathbf{S}, Z|\mathbf{X}, Y]]$$

$$(49)$$

*where*

$$l_{\mathbf{S}}(\Theta_{pfGMM}) = \mathbb{E}_{f^*} \left[\sum_{d=1}^{D} \log f_{S_d}(s_d)\right]$$

$$l_{\mathbf{X}}(\Theta_{pfGMM}) = \frac{1}{KL} \sum_{k=1}^{K} \sum_{l=1}^{L} \mathbb{E}_{f^*_{m_{kl}}} \left[\log f_{\mathbf{X}_{1-\mathbf{s}}}(\mathbf{X}_{1-\mathbf{s}}; \Theta_{pfGMM})\right] + \frac{1}{KL} \sum_{j=1}^{J} \sum_{(k,l) \in M_j} \mathbb{E}_{f^*_{m_{kl}}} \left[\log f_{\mathbf{X}_s, j}(\mathbf{X_s}; \Theta_{pfGMM})\right]$$

$$l_Y(\Theta_{pfGMM}) = \frac{1}{KL} \sum_{j=1}^{J} \sum_{(k,l) \in M_j} \mathbb{E}_{f^*_{Y,k}} \left[\log f_{Y,j}(Y; \Theta_{pfGMM})\right]$$

$$H[\mathbf{S}, Z|\mathbf{X}, Y] = -\mathbb{E}_{f^*} \left[\mathbb{E}_{f(\mathbf{S}, Z|\mathbf{X}, Y)} \left[\log f(\mathbf{S}, Z|\mathbf{X}, Y)\right]\right]$$

$$\psi_{\mathbf{s}}(\mathbf{X}, Y) := f(\mathbf{S} = \mathbf{s}|\mathbf{X}, Y), \quad \text{which is 1 for some } \mathbf{s} \in \{0, 1\}^D \text{ and 0 otherwise.}$$

*Proof.* The derivation is similar to the sup-GMM case:

$$l(\Theta_{\text{pfGMM}}) = \mathop{\mathbb{E}}_{f^*} \left[ \log f(\mathbf{X}, Y) \right]$$

$$= \mathop{\mathbb{E}}_{f^*} \left[ \mathop{\mathbb{E}}_{f(\mathbf{S},Z|\mathbf{X},Y)} \left[ \log f(\mathbf{X}, Y, \mathbf{S}, Z) - \log f(\mathbf{S}, Z|\mathbf{X}, Y) \right] \right]$$

$$= \mathop{\mathbb{E}}_{f^*} \left[ \mathop{\mathbb{E}}_{f(\mathbf{S},Z|\mathbf{X},Y)} \left[ \log \left[ \prod_{d=1}^{D} f_{S_d}(S_d) \right] f_Z(Z) f_{\mathbf{X}_{1-\mathbf{s}}}(\mathbf{X}_{1-\mathbf{s}}; \Theta_{\text{pfGMM}}) f_{\mathbf{X}_{\mathbf{s}},j}(\mathbf{X}_{\mathbf{S}}; \Theta_{\text{pfGMM}}) f_{Y,j}(Y; \Theta_{\text{pfGMM}}) \right] \right]$$

$$\underbrace{- \mathop{\mathbb{E}}_{f^*} \left[ \mathop{\mathbb{E}}_{f(\mathbf{S},Z|\mathbf{X},Y)} \left[ \log f(\mathbf{S}, Z|\mathbf{X}, Y) \right] \right]}_{H[\mathbf{S},Z|\mathbf{X},Y]}$$

Specifically looking at the first term:

$$\overset{a}{=} \log \mathop{\mathbb{E}}_{f^*} \left[ \mathop{\mathbb{E}}_{f(Z|\mathbf{X},Y)} [f_Z(Z)] \right] + \mathop{\mathbb{E}}_{f^*} \left[ \mathop{\mathbb{E}}_{f(\mathbf{S}|\mathbf{X},Y)} \left[ \sum_{d=1}^{D} \log f_{S_d}(S_d) \right] \right] + \mathop{\mathbb{E}}_{f^*} \left[ \mathop{\mathbb{E}}_{f(\mathbf{S},Z|\mathbf{X},Y)} \left[ \log f_{Y,j}(Y; \Theta_{\text{pfGMM}}) \right] \right]$$

$$+ \mathop{\mathbb{E}}_{f^*} \left[ \mathop{\mathbb{E}}_{f(\mathbf{S}|\mathbf{X},Y)f(Z|\mathbf{S},\mathbf{X},Y)} \left[ \log f_{\mathbf{X}_{\mathbf{s}},j}(\mathbf{X}_{\mathbf{S}}; \Theta_{\text{pfGMM}}) + \log f_{\mathbf{X}_{1-\mathbf{s}}}(\mathbf{X}_{1-\mathbf{s}}; \Theta_{\text{pfGMM}}) \right] \right]$$

$$\overset{b}{=} \log \frac{1}{J} + \mathop{\mathbb{E}}_{f^*} \left[ \sum_{\mathbf{s} \in \{0,1\}^D} \psi_{\mathbf{s}}(\mathbf{X}, Y) \sum_{d=1}^{D} \log f_{S_d}(s_d) \right]$$

$$+ \mathop{\mathbb{E}}_{f^*} \left[ \sum_{\mathbf{s} \in \{0,1\}^D} \sum_{j=1}^{J} \psi_{\mathbf{s}}(\mathbf{X}, Y) f(Z = j|\mathbf{S} = \mathbf{s}, \mathbf{X}, Y) \log f_{Y,j}(Y; \Theta_{\text{pfGMM}}) \right]$$

$$+ \mathop{\mathbb{E}}_{f^*} \left[ \sum_{\mathbf{s} \in \{0,1\}^D} \sum_{j=1}^{J} \psi_{\mathbf{s}}(\mathbf{X}, Y) f(Z = j|\mathbf{S} = \mathbf{s}, \mathbf{X}, Y) \left( \log f_{\mathbf{X}_{\mathbf{s}},j}(\mathbf{X}_{\mathbf{S}}; \Theta_{\text{pfGMM}}) + \log f_{\mathbf{X}_{1-\mathbf{s}}}(\mathbf{X}_{1-\mathbf{s}}; \Theta_{\text{pfGMM}}) \right) \right]$$

$$\overset{c}{=} \log \frac{1}{J} + \mathop{\mathbb{E}}_{f^*} \left[ \sum_{d=1}^{D} \log f_{S_d}(s_d) \right]$$

$$+ \frac{1}{KL} \sum_{k=1}^{K} \sum_{l=1}^{L} \mathop{\mathbb{E}}_{f_{m_{kl}}^*} \left[ \sum_{j=1}^{J} f(Z = j|\mathbf{S} = \mathbf{s}, \mathbf{X}, Y) \log f_{Y,j}(Y; \Theta_{\text{pfGMM}}) \right]$$

$$+ \frac{1}{KL} \sum_{k=1}^{K} \sum_{l=1}^{L} \mathop{\mathbb{E}}_{f_{m_{kl}}^*} \left[ \log f_{\mathbf{X}_{1-\mathbf{s}}}(\mathbf{X}_{1-\mathbf{s}}; \Theta_{\text{pfGMM}}) + \sum_{j=1}^{J} f(Z = j|\mathbf{S} = \mathbf{s}, \mathbf{X}, Y) \log f_{\mathbf{X}_{\mathbf{s}},j}(\mathbf{X}_{\mathbf{S}}; \Theta_{\text{pfGMM}}) \right]$$

$$\overset{d}{=} \log \frac{1}{J} + \underbrace{\mathop{\mathbb{E}}_{f^*} \left[ \sum_{d=1}^{D} \log f_{S_d}(s_d) \right]}_{l_{\mathbf{S}}(\Theta_{\text{pfGMM}})} + \underbrace{\frac{1}{KL} \sum_{j=1}^{J} \sum_{(k,l) \in M_j} \mathop{\mathbb{E}}_{f_{m_{kl}}^*} \left[ \log f_{Y,j}(Y; \Theta_{\text{pfGMM}}) \right]}_{l_Y(\Theta_{\text{pfGMM}})}$$

$$\underbrace{+ \frac{1}{KL} \sum_{k=1}^{K} \sum_{l=1}^{L} \mathop{\mathbb{E}}_{f_{m_{kl}}^*} \left[ \log f_{\mathbf{X}_{1-\mathbf{s}}}(\mathbf{X}_{1-\mathbf{s}}; \Theta_{\text{pfGMM}}) \right] + \frac{1}{KL} \sum_{j=1}^{J} \sum_{(k,l) \in M_j} \mathop{\mathbb{E}}_{f_{m_{kl}}^*} \left[ \log f_{\mathbf{X}_{\mathbf{s}},j}(\mathbf{X}_{\mathbf{s}}; \Theta_{\text{pfGMM}}) \right]}_{l_{\mathbf{X}}(\Theta_{\text{pfGMM}})}$$

where equation (a) expands the log terms, (b) expands the expectation over $Z$ and $\mathbf{S}$, (c) uses the assumption that $\psi_{\mathbf{s}}(\mathbf{X}, Y) = 1$ for some $\mathbf{s} \in \{0,1\}^D$, and (d) swaps the order of summations over $j$ over $(k,l)$ in a similar way to the sup-GMM case.

$\square$

| | Description | Values | | | |
|---|---|---|---|---|---|
| | | $j = 1$ | | $j = 2$ | |
| $j$ | Model component index | | | | |
| $(k, l) \in M_j$ | Data component index assigned to model component $j$ | $k=1, l=1$ | $k=1, l=2$ | $k=2, l=1$ | $k=2, l=2$ |
| **Aligned Parameters**, $\Theta^a_{\text{pfGMM}}$ | | | | | |
| $\mathbf{S} = [\mathbf{1}_{D_{\text{R}}}; \mathbf{0}_{D_{\text{Ir}}}]$ | Switches | | | | |
| $f_{\mathbf{X_S}, j} = f^*_{\text{R}, k}$ | Learned model density for $\mathbf{X_S} = \mathbf{X}_{\text{R}}$ | $f^*_{\text{R},1}$ | $f^*_{\text{R},1}$ | $f^*_{\text{R},2}$ | $f^*_{\text{R},2}$ |
| $f_{\mathbf{X_{1-S}}} = \phi(\bar{\mu}_{\text{Ir}}, \bar{\Sigma}_{\text{Ir}})$ | Learned model density for $\mathbf{X_{1-S}} = \mathbf{X}_{\text{Ir}}$ | $\phi(\bar{\mu}_{\text{Ir}}, \bar{\Sigma}_{\text{Ir}})$ | $\phi(\bar{\mu}_{\text{Ir}}, \bar{\Sigma}_{\text{Ir}})$ | $\phi(\bar{\mu}_{\text{Ir}}, \bar{\Sigma}_{\text{Ir}})$ | $\phi(\bar{\mu}_{\text{Ir}}, \bar{\Sigma}_{\text{Ir}})$ |
| $f_{Y, j} = f^*_{Y, k}$ | Learned model density for $Y$ | $f^*_{Y,1}$ | $f^*_{Y,1}$ | $f^*_{Y,2}$ | $f^*_{Y,2}$ |
| **Unaligned Parameters**, $\Theta^u_{\text{pfGMM}}$ | | | | | |
| $\mathbf{S} = [\mathbf{0}_{D_{\text{R}}}; \mathbf{1}_{D_{\text{Ir}}}]$ | Switches | | | | |
| $f_{\mathbf{X_S}, j} = f^*_{\text{Ir}, l}$ | Learned model density for $\mathbf{X_S} = \mathbf{X}_{\text{Ir}}$ | $f^*_{\text{Ir},1}$ | $f^*_{\text{Ir},2}$ | $f^*_{\text{Ir},1}$ | $f^*_{\text{Ir},2}$ |
| $f_{\mathbf{X_{1-S}}} = \phi(\bar{\mu}_{\text{R}}, \bar{\Sigma}_{\text{R}})$ | Learned model density for $\mathbf{X_{1-S}} = \mathbf{X}_{\text{R}}$ | $\phi(\bar{\mu}_{\text{R}}, \bar{\Sigma}_{\text{R}})$ | $\phi(\bar{\mu}_{\text{R}}, \bar{\Sigma}_{\text{R}})$ | $\phi(\bar{\mu}_{\text{R}}, \bar{\Sigma}_{\text{R}})$ | $\phi(\bar{\mu}_{\text{R}}, \bar{\Sigma}_{\text{R}})$ |
| $f_{Y, j} = \mathbf{Bern}(0.5)$ | Learned model density for $Y$ | $\mathbf{Bern}(0.5)$ | $\mathbf{Bern}(0.5)$ | $\mathbf{Bern}(0.5)$ | $\mathbf{Bern}(0.5)$ |
| **Single-Gaussian Parameters**, $\Theta^{\text{1-G}}_{\text{pfGMM}}$ | | | | | |
| $\mathbf{S} = [\mathbf{0}_{D_{\text{R}}}; \mathbf{0}_{D_{\text{Ir}}}]$ | Switches | | | | |
| $f_{\mathbf{X_S}} = 1$ | Learned model density for $\mathbf{X_S} = \{\}$ | 1 | 1 | 1 | 1 |
| $f_{\mathbf{X_{1-S}}} = \phi(\bar{\mu}, \bar{\Sigma})$ | Learned model density for $\mathbf{X_{1-S}} = \mathbf{X}$ | $\phi(\bar{\mu}, \bar{\Sigma})$ | $\phi(\bar{\mu}, \bar{\Sigma})$ | $\phi(\bar{\mu}, \bar{\Sigma})$ | $\phi(\bar{\mu}, \bar{\Sigma})$ |
| $f_{Y, j} = \mathbf{Bern}(0.5)$ | Learned model density for $Y$ | $\mathbf{Bern}(0.5)$ | $\mathbf{Bern}(0.5)$ | $\mathbf{Bern}(0.5)$ | $\mathbf{Bern}(0.5)$ |

Table 3: pf-GMM: Summary of the learned model densities for the aligned and unaligned cases.

**Theorem C.7.** *Assume that the data is generated with density $f^*(\mathbf{X}, Y)$ and parameters $\Theta^*_{pfGMM} = \left\{\mu^*_j, \Sigma^*_j, \eta^*_j\right\}^J_{j=1}$ specified by Equation C.1 and the model is a pf-GMM with learnt parameters $\Theta_{pfGMM} = \left\{\hat{\mu}_j, \hat{\Sigma}_j, \hat{\eta}_j\right\}^J_{j=1}, \hat{\mu}^\pi, \hat{\Sigma}^\pi$. Further assume that the data is generated with hyperparameters $K = L = 2$ and the model has $J = 2$ components. Define $\Theta^a_{pfGMM}$ as the parameters that pf-GMM model learns when its components are aligned with the relevant components of the data, and $\Theta^u_{pfGMM}$ as the parameters that pf-GMM model learns when its components are unaligned with the relevant components of the data. Then, the population log-likelihood for the parameters $\Theta^a_{pfGMM}$ and $\Theta^u_{pfGMM}$ is:*

$$l(\Theta^a_{pfGMM}) = \log\frac{1}{2} + D_R \log p + D_{Ir} \log(1-p)$$

$$- \frac{1}{4}\sum_{k=1}^{2}\sum_{l=1}^{2} H\big(\phi(\mu^*_{R,k},\Sigma^*_{R,k})\big) + H\big(\phi(\mu^*_{Ir,l},\Sigma^*_{Ir,l})\big) + \boldsymbol{KL}\big(\phi(\mu^*_{Ir,l},\Sigma^*_{Ir,l})||\phi(\bar\mu_{Ir},\bar\Sigma_{Ir})\big)$$

$$l(\Theta^u_{pfGMM}) = \log\frac{1}{2} + D_{Ir}\log p + D_R \log(1-p) - \log 2$$

$$+ \frac{1}{4}\sum_{k=1}^{2}\sum_{l=1}^{2} -H\big(\phi(\mu^*_{Ir,l},\Sigma^*_{Ir,l})\big) - H\big(\phi(\mu^*_{R,k},\Sigma^*_{R,k})\big) - \boldsymbol{KL}\big(\phi(\mu^*_{R,k},\Sigma^*_{R,k})||\phi(\bar\mu_R,\bar\Sigma_R)\big)$$

$$l(\Theta^{1\text{-}G}_{pfGMM}) = \log\frac{1}{2} + D\log(1-p) - \log 2$$

$$+ \frac{1}{4}\sum_{k=1}^{2}\sum_{l=1}^{2} -H\big(\phi(\mu^*_{R,k},\Sigma^*_{R,k})\big) - H\big(\phi(\mu^*_{Ir,l},\Sigma^*_{Ir,l})\big)$$

$$+ \frac{1}{4}\sum_{k=1}^{2}\sum_{l=1}^{2} -\boldsymbol{KL}\big(\phi(\mu^*_{R,k},\Sigma^*_{R,k})||\phi(\bar\mu_R,\bar\Sigma_R)\big) - \boldsymbol{KL}\big(\phi(\mu^*_{Ir,l},\Sigma^*_{Ir,l})||\phi(\bar\mu_{Ir},\bar\Sigma_{Ir})\big)$$

*Furthermore, the difference in the population log-likelihoods between the aligned and other cases is:*

$$l(\Theta^a_{pfGMM}) - l(\Theta^u_{pfGMM}) = (D_R - D_{Ir})\log\frac{p}{1-p} + \log 2$$

$$+ \frac{1}{4}\sum_{k=1}^{2}\sum_{l=1}^{2} \boldsymbol{KL}\big(\phi(\mu^*_{R,k},\Sigma^*_{R,k})||\phi(\bar\mu_R,\bar\Sigma_R)\big) - \boldsymbol{KL}\big(\phi(\mu^*_{Ir,l},\Sigma^*_{Ir,l})||\phi(\bar\mu_{Ir},\bar\Sigma_{Ir})\big)$$

$$l(\Theta^a_{pfGMM}) - l(\Theta^{1\text{-}G}_{pfGMM}) = D_R\log\frac{p}{1-p} + \log 2 + \frac{1}{4}\sum_{k=1}^{2}\sum_{l=1}^{2} \boldsymbol{KL}\big(\phi(\mu^*_{R,k},\Sigma^*_{R,k})||\phi(\bar\mu_R,\bar\Sigma_R)\big)$$

*Proof.* Like in the sup-GMM case, we first write out the individual terms for $\Theta^a_{\text{pfGMM}}, \Theta^u_{\text{pfGMM}}, \Theta^{1\text{-}G}_{\text{pfGMM}}$:

**Aligned case ($\Theta^a_{\text{pfGMM}}$)** Aligned parameters $\Theta^a_{\text{pfGMM}}$ correspond to the case when the model components are *aligned* with the relevant components of the data. We first write out the individual terms in $\Theta^a_{\text{pfGMM}}$:

$$\hat\mu^a_j = \begin{bmatrix}\mu^*_{R,j}\\\bar\mu_{Ir}\end{bmatrix} \qquad \text{for } j\in\{1,2\} \quad \text{where} \qquad \bar\mu_{Ir} = \frac{1}{L}\sum_{l=1}^{2}\mu^*_{Ir,l} \tag{50}$$

$$\hat\Sigma^a_j = \begin{bmatrix}\Sigma^*_{R,j} & 0\\ 0 & \bar\Sigma_{Ir}\end{bmatrix} \qquad \text{for } j\in\{1,2\} \quad \text{where} \qquad \bar\Sigma_{Ir} = \frac{1}{L}\left(\sum_{l=1}^{2}\mu^{*\top}_{Ir,l}\mu^*_{Ir,l} + \Sigma^*_{Ir,l}\right) - \bar\mu^\top_{Ir}\bar\mu_{Ir} \tag{51}$$

$$\hat\eta^a_j = \eta^*_j \qquad \qquad \text{where} \qquad \eta^*_j = \begin{cases}0 & \text{if } j=1\\ 1 & \text{if } j=2\end{cases} \tag{52}$$

$$\hat\mu^{\pi a} = \begin{bmatrix}\bar\mu_{R,j}\\\bar\mu_{Ir}\end{bmatrix} \qquad \qquad \text{where} \qquad \bar\mu_R = \frac{1}{K}\sum_{k=1}^{2}\mu^*_{R,k} \tag{53}$$

$$\hat\Sigma^{\pi a} = \begin{bmatrix}\bar\Sigma_R & 0\\ 0 & \bar\Sigma_{Ir}\end{bmatrix} \qquad \qquad \text{where} \qquad \bar\Sigma_R = \frac{1}{K}\left(\sum_{k=1}^{2}\mu^{*\top}_{R,k}\mu^*_{R,k} + \Sigma^*_{R,k}\right) - \bar\mu^\top_R\bar\mu_R \tag{54}$$

Note that the mixture parameters for pf-GMM are the same as sup-GMM and GMM parameters, and the only additional parameters are the last two elements (corresponding to parameters $\hat\mu^{\pi a}, \hat\Sigma^{\pi a}$).

Computing the $l_{\mathbf{S}}$ term (using $\psi_{\mathbf{s}}(\mathbf{X}, Y) = 1$ for $\mathbf{s} = [\mathbf{1}_{D_{\mathrm{R}}}, \mathbf{0}_{D_{\mathrm{Ir}}}]$ and 0 otherwise):

$$
\begin{aligned}
l_{\mathbf{S}}(\Theta_{\mathrm{pfGMM}}^{a}) &= \underset{f^*}{\mathbb{E}} \left[ \sum_{d \in D} \log f_{S_d}(s_d) \right] \\
&= \sum_{d \in D} \log f_{S_d}(s_d) \\
&= D_{\mathrm{R}} \log p + D_{\mathrm{Ir}} \log(1-p)
\end{aligned}
$$

Computing the $l_{\mathbf{X}}$ term:

$$
\begin{aligned}
l_{\mathbf{X}}(\Theta_{\mathrm{pfGMM}}^{a}) &= \frac{1}{KL} \sum_{k=1}^{K} \sum_{l=1}^{L} \underset{f_{m_{kl}}^*}{\mathbb{E}} \left[ \log f_{\mathbf{X}_{1-\mathbf{s}}}(\mathbf{X}_{1-\mathbf{s}}; \Theta_{\mathrm{pfGMM}}) \right] + \frac{1}{KL} \sum_{j=1}^{J} \sum_{(k,l) \in M_j} \underset{f_{m_{kl}}^*}{\mathbb{E}} \left[ \log f_{\mathbf{X}_{\mathbf{s},j}}(\mathbf{X}_{\mathbf{s}}; \Theta_j) \right] \\
&\overset{a}{=} \frac{1}{KL} \sum_{k=1}^{2} \sum_{l=1}^{2} \underset{f_{\mathrm{Ir},l}^*}{\mathbb{E}} \left[ \log \phi(\mathbf{X}_{\mathrm{Ir}}; \bar{\mu}_{\mathrm{Ir}}, \bar{\Sigma}_{\mathrm{Ir}}) \right] + \frac{1}{KL} \sum_{j=1}^{2} \sum_{(k,l) \in M_j} \underset{f_{\mathrm{R},k}^*}{\mathbb{E}} \left[ \log f_{\mathrm{R},k}^*(\mathbf{X}_{\mathrm{R},k}) \right] \\
&\overset{b}{=} \frac{1}{KL} \sum_{k=1}^{2} \sum_{l=1}^{2} \left[ -\mathrm{H}\left( \phi(\mu_{\mathrm{Ir},l}^*, \Sigma_{\mathrm{Ir},l}^*) \right) - \mathbf{KL}\left( \phi(\mu_{\mathrm{Ir},l}^*, \Sigma_{\mathrm{Ir},l}^*) || \phi(\bar{\mu}_{\mathrm{Ir}}, \bar{\Sigma}_{\mathrm{Ir}}) \right) \right] + \frac{1}{KL} \sum_{j=1}^{2} \sum_{(k,l) \in M_j} -\mathrm{H}\left( \phi(\mu_{\mathrm{R},k}^*, \Sigma_{\mathrm{R},k}^*) \right) \\
&= \frac{1}{4} \sum_{k=1}^{2} \sum_{l=1}^{2} -\mathrm{H}\left( \phi(\mu_{\mathrm{R},k}^*, \Sigma_{\mathrm{R},k}^*) \right) - \mathrm{H}\left( \phi(\mu_{\mathrm{Ir},l}^*, \Sigma_{\mathrm{Ir},l}^*) \right) - \mathbf{KL}\left( \phi(\mu_{\mathrm{Ir},l}^*, \Sigma_{\mathrm{Ir},l}^*) || \phi(\bar{\mu}_{\mathrm{Ir}}, \bar{\Sigma}_{\mathrm{Ir}}) \right)
\end{aligned}
$$

which is the same expression as in sup-GMM case. Here, step (a) follows from the fact that $\mathbf{X}_{1-\mathbf{s}} = \mathbf{X}_{\mathrm{Ir}}$ and $\mathbf{X}_{\mathbf{s}} = \mathbf{X}_{\mathrm{R}}$ for the aligned case, and the learned model densities for these cases are given in Table 3. Step (b) follows from the definitions of entropy and KL-divergence.

The $l_Y, \log \frac{1}{J}, \mathbb{E}_{f^*}[H[Z|\mathbf{X}, Y]]$ terms are same as in sup-GMM case:

$$
\begin{aligned}
l_Y(\Theta_{\mathrm{pfGMM}}^{a}) &= 0 \\
\log \frac{1}{J} &= \log \frac{1}{2} \\
\underset{f^*}{\mathbb{E}}[H[Z|\mathbf{X}, Y]] &= 0
\end{aligned}
$$

The likelihood for the aligned case is therefore:

$$
\begin{aligned}
l(\Theta_{\mathrm{pfGMM}}^{a}) = &\log \frac{1}{2} + D_{\mathrm{R}} \log p + D_{\mathrm{Ir}} \log(1-p) \\
&- \frac{1}{4} \sum_{k=1}^{2} \sum_{l=1}^{2} \mathrm{H}\left( \phi(\mu_{\mathrm{R},k}^*, \Sigma_{\mathrm{R},k}^*) \right) + \mathrm{H}\left( \phi(\mu_{\mathrm{Ir},l}^*, \Sigma_{\mathrm{Ir},l}^*) \right) + \mathbf{KL}\left( \phi(\mu_{\mathrm{Ir},l}^*, \Sigma_{\mathrm{Ir},l}^*) || \phi(\bar{\mu}_{\mathrm{Ir}}, \bar{\Sigma}_{\mathrm{Ir}}) \right)
\end{aligned}
$$

**Unaligned case ($\Theta^u_{\mathbf{pfGMM}}$)**

$$\hat{\mu}^u_j = \begin{bmatrix} \bar{\mu}_R \\ \mu^*_{Ir,j} \end{bmatrix} \qquad \text{for } j \in \{1,2\} \quad \text{where} \qquad \bar{\mu}_R = \frac{1}{K} \sum_{k=1}^2 \mu^*_{R,k} \tag{55}$$

$$\hat{\Sigma}^u_j = \begin{bmatrix} \bar{\Sigma}_R & 0 \\ 0 & \Sigma^*_{Ir,j} \end{bmatrix} \qquad \text{for } j \in \{1,2\} \quad \text{where} \qquad \bar{\Sigma}_R = \frac{1}{K} \left( \sum_{k=1}^2 \mu^{*\top}_{R,k} \mu^*_{R,k} + \Sigma^*_{R,k} \right) - \bar{\mu}^\top_R \bar{\mu}_R \tag{56}$$

$$\hat{\eta}^u_j = \frac{1}{2} \tag{57}$$

$$\hat{\mu}^{\pi u} = \begin{bmatrix} \bar{\mu}_{R,j} \\ \bar{\mu}_{Ir} \end{bmatrix} \qquad \text{where} \qquad \bar{\mu}_{Ir} = \frac{1}{L} \sum_{l=1}^2 \mu^*_{Ir,l} \tag{58}$$

$$\hat{\Sigma}^{\pi u} = \begin{bmatrix} \bar{\Sigma}_R & 0 \\ 0 & \bar{\Sigma}_{Ir} \end{bmatrix} \qquad \text{where} \qquad \bar{\Sigma}_{Ir} = \frac{1}{K} \left( \sum_{k=1}^2 \mu^{*\top}_{Ir,k} \mu^*_{Ir,k} + \Sigma^*_{Ir,k} \right) - \bar{\mu}^\top_{Ir} \bar{\mu}_{Ir} \tag{59}$$

Computing the $l_\mathbf{S}$ term (using $\psi_\mathbf{s}(\mathbf{X}, Y) = 1$ for $\mathbf{s} = [\mathbf{0}_{D_R}, \mathbf{1}_{D_{Ir}}]$ and 0 otherwise):

$$l_\mathbf{S}(\Theta^u_{pfGMM}) = \mathbb{E}_{f^*} \left[ \sum_{d \in D} \log f_{S_d}(s_d) \right]$$

$$= D_{Ir} \log p + D_R \log(1-p)$$

The rest of the terms are same as in the unaligned sup-GMM case:

$$l_\mathbf{X}(\Theta^u_{pfGMM}) = \frac{1}{4} \sum_{k=1}^2 \sum_{l=1}^2 -\mathrm{H}\left(\phi(\mu^*_{Ir,l}, \Sigma^*_{Ir,l})\right) - \mathrm{H}\left(\phi(\mu^*_{R,k}, \Sigma^*_{R,k})\right) - \mathbf{KL}\left(\phi(\mu^*_{R,k}, \Sigma^*_{R,k}) || \phi(\bar{\mu}_R, \bar{\Sigma}_R)\right)$$

$$l_Y(\Theta^u_{pfGMM}) = -\log 2$$

$$\log \frac{1}{J} = \log \frac{1}{2}$$

$$\mathbb{E}_{f^*} [H[Z|\mathbf{X}, Y]] = 0$$

to get:

$$l(\Theta^u_{pfGMM}) = \log \frac{1}{2} + D_{Ir} \log p + D_R \log(1-p) - \log 2$$

$$+ \frac{1}{4} \sum_{k=1}^2 \sum_{l=1}^2 -\mathrm{H}\left(\phi(\mu^*_{Ir,l}, \Sigma^*_{Ir,l})\right) - \mathrm{H}\left(\phi(\mu^*_{R,k}, \Sigma^*_{R,k})\right) - \mathbf{KL}\left(\phi(\mu^*_{R,k}, \Sigma^*_{R,k}) || \phi(\bar{\mu}_R, \bar{\Sigma}_R)\right)$$

and the difference in the likelihoods between the aligned and unaligned cases is:

$$l(\Theta^a_{pfGMM}) - l(\Theta^u_{pfGMM}) = (D_R - D_{Ir}) \log \frac{p}{1-p} + \log 2$$

$$+ \frac{1}{4} \sum_{k=1}^2 \sum_{l=1}^2 \mathbf{KL}\left(\phi(\mu^*_{R,k}, \Sigma^*_{R,k}) || \phi(\bar{\mu}_R, \bar{\Sigma}_R)\right) - \mathbf{KL}\left(\phi(\mu^*_{Ir,l}, \Sigma^*_{Ir,l}) || \phi(\bar{\mu}_{Ir}, \bar{\Sigma}_{Ir})\right)$$

**Single-Gaussian case ($\Theta^{\mathbf{1\text{-}G}}_{\mathbf{pfGMM}}$)** The single-Gaussian case is a special case for the pf-GMM model, where the model has only one component (i.e. it considers all the data to be generated from a single

Gaussian distribution). The parameters for this case are:

$$\hat{\mu}_j^{\text{1-G}} = \hat{\mu}^{\pi\text{1-G}} = \begin{bmatrix} \bar{\mu}_{\text{R},j} \\ \bar{\mu}_{\text{Ir}} \end{bmatrix} \qquad \text{for } j \in \{1,2\} \quad \text{where} \quad \bar{\mu}_{\text{R}} = \frac{1}{K} \sum_{k=1}^{2} \mu_{\text{R},k}^* \tag{60}$$

$$\text{and} \qquad \bar{\mu}_{\text{Ir}} = \frac{1}{L} \sum_{l=1}^{2} \mu_{\text{Ir},l}^* \tag{61}$$

$$\hat{\Sigma}_j^{\text{1-G}} = \hat{\Sigma}^{\pi\text{1-G}} = \begin{bmatrix} \bar{\Sigma}_{\text{R}} & 0 \\ 0 & \bar{\Sigma}_{\text{Ir}} \end{bmatrix} \qquad \text{for } j \in \{1,2\} \quad \text{where} \quad \bar{\Sigma}_{\text{R}} = \frac{1}{K} \left( \sum_{k=1}^{2} \mu_{\text{R},k}^{*\top} \mu_{\text{R},k}^* + \Sigma_{\text{R},k}^* \right) - \bar{\mu}_{\text{R}}^\top \bar{\mu}_{\text{R}} \tag{62}$$

$$\text{and} \qquad \bar{\Sigma}_{\text{Ir}} = \frac{1}{K} \left( \sum_{k=1}^{2} \mu_{\text{Ir},k}^{*\top} \mu_{\text{Ir},k}^* + \Sigma_{\text{Ir},k}^* \right) - \bar{\mu}_{\text{Ir}}^\top \bar{\mu}_{\text{Ir}} \tag{63}$$

Computing the $l_{\mathbf{S}}$ term (using $\psi_{\mathbf{s}}(\mathbf{X}, Y) = 1$ for $\mathbf{s} = [\mathbf{0}_{D_{\text{R}}}, \mathbf{0}_{D_{\text{Ir}}}]$ and 0 otherwise):

$$l_{\mathbf{S}}(\Theta_{\text{pfGMM}}^{\text{1-G}}) = \mathbb{E}_{f^*} \left[ \sum_{d \in D} \log f_{S_d}(s_d) \right]$$
$$= D \log(1 - p)$$

Next we compute the $l_{\mathbf{X}}$ term:

$$l_{\mathbf{X}}(\Theta_{\text{pfGMM}}^{\text{1-G}}) = \frac{1}{KL} \sum_{k=1}^{K} \sum_{l=1}^{L} \mathbb{E}_{f_{m_{kl}}^*} \left[ \log f_{\mathbf{X}_{1-\mathbf{s}}}(\mathbf{X}_{1-\mathbf{s}}; \Theta_{\text{pfGMM}}) \right]$$

$$= \frac{1}{4} \sum_{k=1}^{2} \sum_{l=1}^{2} \mathbb{E}_{f_{\text{R},k}^*} \left[ \log \phi(\mathbf{X}_{\text{R}}; \bar{\mu}_{\text{R}}, \bar{\Sigma}_{\text{R}}) \right] + \mathbb{E}_{f_{\text{Ir},l}^*} \left[ \log \phi(\mathbf{X}_{\text{Ir}}; \bar{\mu}_{\text{Ir}}, \bar{\Sigma}_{\text{Ir}}) \right]$$

$$= \frac{1}{4} \sum_{k=1}^{2} \sum_{l=1}^{2} -\text{H}\left( \phi(\mu_{\text{R},k}^*, \Sigma_{\text{R},k}^*) \right) - \mathbf{KL}\left( \phi(\mu_{\text{R},k}^*, \Sigma_{\text{R},k}^*) || \phi(\bar{\mu}_{\text{R}}, \bar{\Sigma}_{\text{R}}) \right)$$

$$+ \frac{1}{4} \sum_{k=1}^{2} \sum_{l=1}^{2} -\text{H}\left( \phi(\mu_{\text{Ir},l}^*, \Sigma_{\text{Ir},l}^*) \right) - \mathbf{KL}\left( \phi(\mu_{\text{Ir},l}^*, \Sigma_{\text{Ir},l}^*) || \phi(\bar{\mu}_{\text{Ir}}, \bar{\Sigma}_{\text{Ir}}) \right)$$

The $l_Y, \log \frac{1}{J}, \mathbb{E}_{f^*}[H[Z|\mathbf{X},Y]]$ terms are same as in the unaligned pf-GMM case: $l_Y(\Theta_{\text{pfGMM}}^{\text{1-G}}) = -\log 2, \log \frac{1}{J} = \log \frac{1}{2}, \mathbb{E}_{f^*}[H[Z|\mathbf{X},Y]] = 0$. Therefore, the likelihood for the single-Gaussian case is:

$$l(\Theta_{\text{pfGMM}}^{\text{1-G}}) = \log \frac{1}{2} + D \log(1 - p) - \log 2$$

$$+ \frac{1}{4} \sum_{k=1}^{2} \sum_{l=1}^{2} -\text{H}\left( \phi(\mu_{\text{R},k}^*, \Sigma_{\text{R},k}^*) \right) - \text{H}\left( \phi(\mu_{\text{Ir},l}^*, \Sigma_{\text{Ir},l}^*) \right)$$

$$+ \frac{1}{4} \sum_{k=1}^{2} \sum_{l=1}^{2} -\mathbf{KL}\left( \phi(\mu_{\text{R},k}^*, \Sigma_{\text{R},k}^*) || \phi(\bar{\mu}_{\text{R}}, \bar{\Sigma}_{\text{R}}) \right) - \mathbf{KL}\left( \phi(\mu_{\text{Ir},l}^*, \Sigma_{\text{Ir},l}^*) || \phi(\bar{\mu}_{\text{Ir}}, \bar{\Sigma}_{\text{Ir}}) \right)$$

and the difference in the likelihoods between the aligned and single-Gaussian cases is:

$$l(\Theta_{\text{pfGMM}}^a) - l(\Theta_{\text{pfGMM}}^{\text{1-G}}) = D_{\text{R}} \log \frac{p}{1 - p} + \log 2 + \frac{1}{4} \sum_{k=1}^{2} \sum_{l=1}^{2} \mathbf{KL}\left( \phi(\mu_{\text{R},k}^*, \Sigma_{\text{R},k}^*) || \phi(\bar{\mu}_{\text{R}}, \bar{\Sigma}_{\text{R}}) \right)$$

$\square$

## Spherical Gaussian case

To develop an intuition for the difference in likelihoods we derived above, we analyze the case where data and model components are isotropic Gaussians. First, we define the data:

$$\mu_{R,1}^* = -\mu_R^* \mathbf{1}_R, \quad \mu_{R,2}^* = \mu_R^* \mathbf{1}_R, \quad \Sigma_{R,1}^* = \Sigma_{R,2}^* = \sigma_R^{*2} I_{D_R}$$
$$\mu_{Ir,1}^* = -\mu_{Ir}^* \mathbf{1}_{Ir}, \quad \mu_{Ir,2}^* = \mu_{Ir}^* \mathbf{1}_{Ir}, \quad \Sigma_{Ir,1}^* = \Sigma_{Ir,2}^* = \sigma_{Ir}^{*2} I_{D_{Ir}}$$
$$\implies \bar{\mu}_R = \mathbf{0}_R, \quad \bar{\Sigma}_R = \mu_R^{*2} \mathbf{1}_{R,R} + \sigma_R^{*2} I_{D_R}$$
$$\text{and } \bar{\mu}_{Ir} = \mathbf{0}_{Ir}, \quad \bar{\Sigma}_{Ir} = \mu_{Ir}^{*2} \mathbf{1}_{Ir,Ir} + \sigma_{Ir}^{*2} I_{D_{Ir}} \tag{64}$$

where $\mathbf{1}_{Ir}$ is a $D_{Ir}$-dimensional vector of all ones and $\mathbf{1}_{Ir,Ir}$ is a $D_{Ir} \times D_{Ir}$ matrix of all ones (similarly for $\mathbf{1}_R, \mathbf{1}_{R,R}$). We also define the signal-to-noise ratio (SNR) for the relevant and irrelevant components as:

$$\Delta_R := \frac{\mu_R^*}{\sigma_R^*}, \quad \Delta_{Ir} := \frac{\mu_{Ir}^*}{\sigma_{Ir}^*}$$

Using the above expressions, we can obtain the model parameters $\Theta^a, \Theta^u$, and $\Theta^{1\text{-}G}$ using the parameter definitions for each of the models.

For each of the models, we can compute the difference in log-likelihoods as follows (we use the notation $\Delta(\Theta^a, \Theta^u) := l(\Theta_{\text{Model}}^a) - l(\Theta_{\text{Model}}^u)$ for each of the models):

$$\Delta(\Theta_{\text{GMM}}^a, \Theta_{\text{GMM}}^u) = Q_R - Q_{Ir}$$
$$\Delta(\Theta_{\text{supGMM}}^a, \Theta_{\text{supGMM}}^u) = \log 2 + Q_R - Q_{Ir}$$
$$\Delta(\Theta_{\text{pfGMM}}^a, \Theta_{\text{pfGMM}}^u) = (D_R - D_{Ir}) \log \frac{p}{1-p} + \log 2 + Q_R - Q_{Ir}$$
$$\Delta(\Theta_{\text{pfGMM}}^a, \Theta_{\text{pfGMM}}^{1\text{-}G}) = D_R \log \frac{p}{1-p} + \log 2 + Q_R$$

where $Q_R$ involves terms in $\Delta(\Theta_{\text{GMM}}^a, \Theta_{\text{GMM}}^u)$ that only depend on $D_R$ and $\Delta_R$ (and similarly for $Q_{Ir}$):

$$Q_R = \frac{1}{2}\left[\frac{D_R(1-\Delta_R^2)}{D_R\Delta_R^2 + 1} + \log(D_R\Delta_R^2 + 1)\right]$$
$$Q_{Ir} = \frac{1}{2}\left[\frac{D_{Ir}(1-\Delta_{Ir}^2)}{D_{Ir}\Delta_{Ir}^2 + 1} + \log(D_{Ir}\Delta_{Ir}^2 + 1)\right]$$

**When are relevant components preferred by each model?** For GMM, the relevant components are preferred when $\Theta_{\text{GMM}}^a$ has a higher likelihood than $\Theta_{\text{GMM}}^u$ (and similarly for sup-GMM). For pf-GMM, the relevant components are preferred when $\Theta_{\text{pfGMM}}^a$ has a higher likelihood than $\Theta_{\text{pfGMM}}^u$ and $\Theta_{\text{pfGMM}}^{1\text{-}G}$.

**Tuning pf-GMM to prefer relevant components** For pf-GMM, we can tune the parameter $p$ to prefer relevant components. We can do this by setting $\Delta(\Theta_{\text{pfGMM}}^a, \Theta_{\text{pfGMM}}^u) > 0$ and $\Delta(\Theta_{\text{pfGMM}}^a, \Theta_{\text{pfGMM}}^{1\text{-}G}) > 0$. This gives us the following constraints on $p$:

$$p < p_{u/a} = \begin{cases} \sigma\left(\frac{\log 2 + Q_R - Q_{Ir}}{|D_R - D_{Ir}|}\right) & \text{if } D_R < D_{Ir} \\ \sigma\left(-\frac{\log 2 + Q_R - Q_{Ir}}{|D_R - D_{Ir}|}\right) & \text{if } D_R > D_{Ir} \end{cases}$$
$$p > p_{1\text{-}G/a} = \sigma\left(-\frac{\log 2 + Q_R}{D_R}\right)$$

where $\sigma(x) = \frac{1}{1+\exp(-x)}$ is the sigmoid function. The feasible range of $p$ exists when $p_{u/a} > p_{1\text{-}G/a}$, which is when:

$$D_{\text{Ir}} \log 2 - D_{\text{R}} Q_{\text{Ir}} + D_{\text{Ir}} Q_{\text{R}} > 0 \text{ if } D_{\text{R}} < D_{\text{Ir}}$$
$$D_{\text{Ir}} \log 2 - D_{\text{R}} Q_{\text{Ir}} + D_{\text{Ir}} Q_{\text{R}} < 0 \text{ if } D_{\text{R}} > D_{\text{Ir}}$$

# D  Additional Experiments

## D.1  pf-HMM Simulated Data Experiments

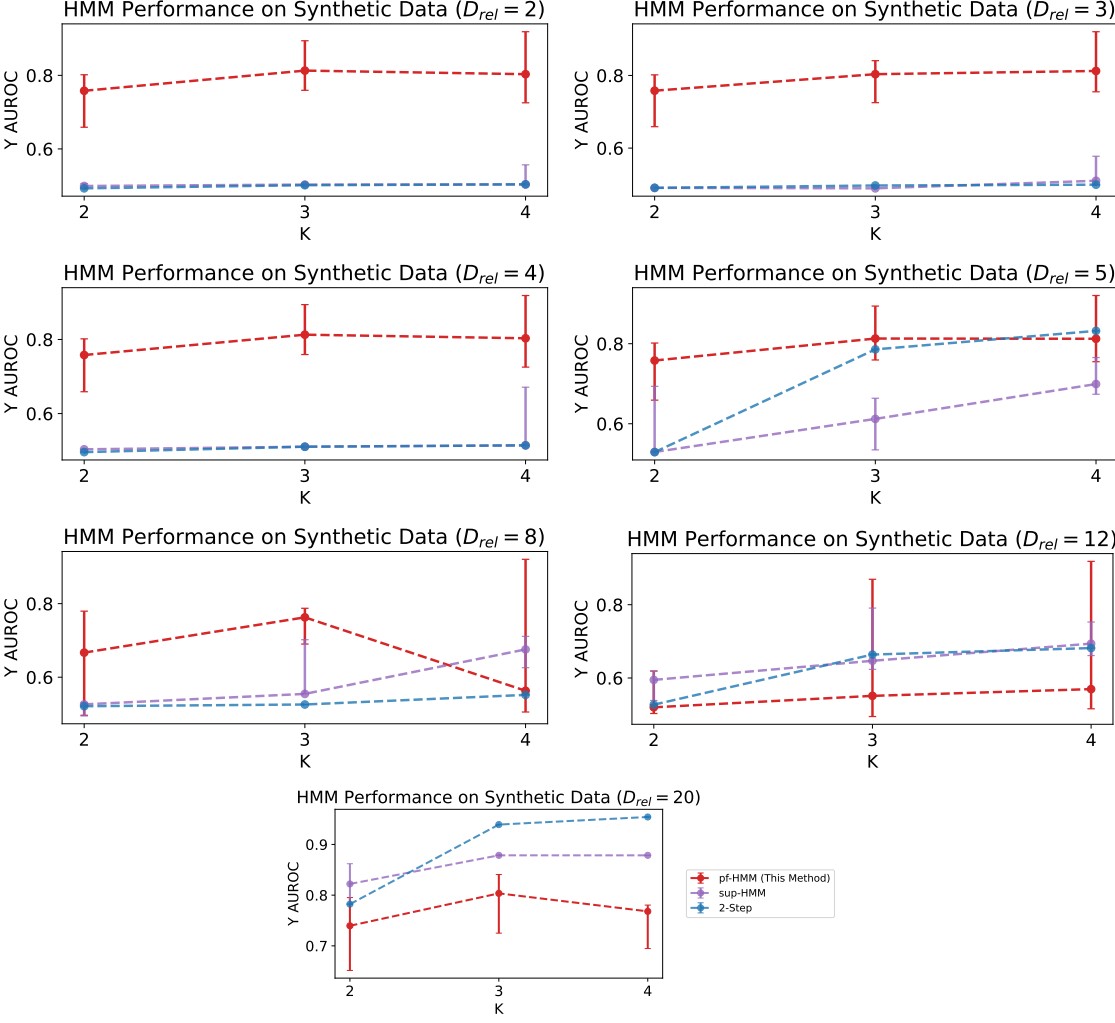

Figure 9: pf-HMM performance with respect to target AUROC on simulated data when we vary the component budget for the mixture models. Each panel corresponds to a dataset with a specified number of relevant dimensions, while keeping the total number of dimensions fixed at 20.

## D.2 pf-GMM Simulated Data Experiments (Full Covariance Parameterization)

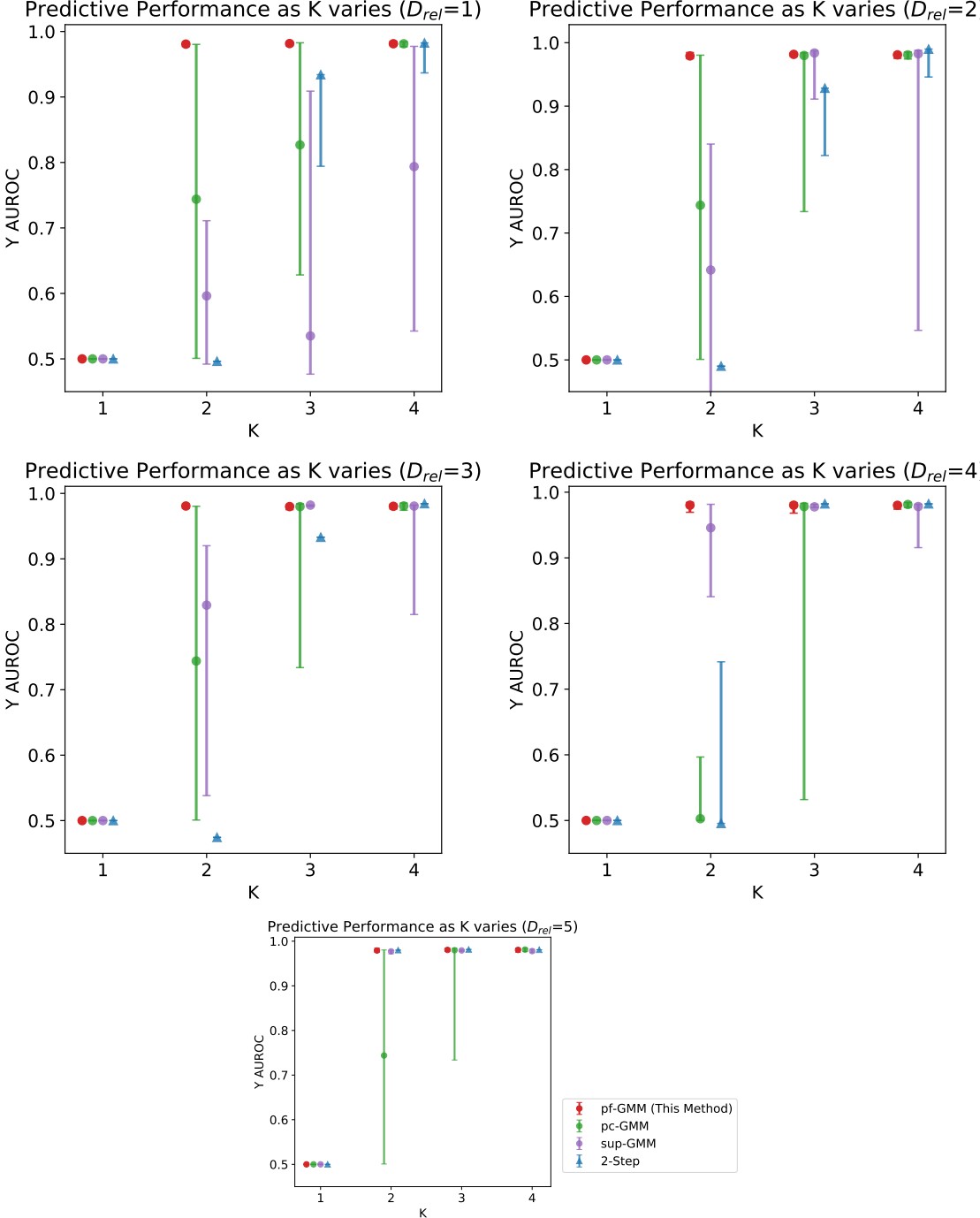

Figure 10: pf-GMM performance (target AUROC) as we vary the component budget (K) for the mixture models on simulated data. We now model the full covariance matrices for all components. Each panel corresponds to a dataset with the number of relevant dimensions fixed, while keeping the total number of dimensions fixed at 8. We see that 2-Step-GMM model under-performs when the $K$ is small, while prediction-focus (in pf-GMM and pc-GMM) helps when the relevant dimensions are fewer. We also see considerable optimization variation in the models, owing to a larger number of parameters needed to learn a full-covariance matrix

| $D_\mathrm{R}$ | Model | $\log P(Y\|\mathbf{X})$ | $\log P(\mathbf{X})$ | $\log P(\mathbf{X}_\mathrm{R})$ |
|---|---|---|---|---|
| 20 | pf-GMM | **-0.05** | -105.65 | **7.38** |
| | pf-GMM (Pred. Term Only) | -0.18 | -150.18 | -20.93 |
| | pc-GMM | -0.50 | -130.41 | -15.51 |
| | sup-GMM | -0.66 | -30.81 | -15.48 |
| | 2-Step | -0.70 | 5.79 | -26.23 |
| | Class Cond. GMM | -5.92 | **14.87** | -20.66 |
| 30 | pf-GMM | **-0.07** | -85.90 | **12.74** |
| | pf-GMM (Pred. Term Only) | -0.19 | -143.28 | -33.05 |
| | pc-GMM | -0.51 | -126.11 | -25.99 |
| | sup-GMM | -0.66 | -45.79 | -23.08 |
| | 2-Step | -0.69 | -40.33 | -22.51 |
| | Class Cond. GMM | -11.01 | **-20.63** | 12.13 |
| 40 | pf-GMM | **-0.04** | -66.07 | **18.19** |
| | pf-GMM (Pred. Term Only) | -0.09 | -131.40 | -40.66 |
| | pc-GMM | -0.49 | -116.71 | -30.75 |
| | sup-GMM | -0.61 | -66.95 | -18.36 |
| | 2-Step | -0.69 | -55.06 | -26.88 |
| | Class Cond. GMM | -3.90 | **-23.39** | 16.91 |
| 50 | pf-GMM | **-0.09** | -46.73 | **23.15** |
| | pf-GMM (Pred. Term Only) | -0.25 | -135.90 | -56.48 |
| | pc-GMM | -0.51 | -112.77 | -40.68 |
| | sup-GMM | -0.63 | -69.19 | -22.89 |
| | 2-Step | -0.68 | -66.60 | -30.58 |
| | Class Cond. GMM | -7.56 | **-14.29** | 20.76 |
| 80 | pf-GMM | **-0.06** | 10.66 | 37.69 |
| | pf-GMM (Pred. Term Only) | -0.18 | -125.77 | -95.61 |
| | pc-GMM | -0.49 | -84.83 | -55.85 |
| | sup-GMM | -0.41 | -34.03 | -3.01 |
| | 2-Step | -0.07 | 9.21 | **37.69** |
| | Class Cond. GMM | -7.65 | **19.98** | 37.27 |

| K | Model | $\log P(Y\|\mathbf{X})$ | $\log P(\mathbf{X})$ | $\log P(\mathbf{X}_\mathrm{R})$ |
|---|---|---|---|---|
| 4 | pf-GMM | **-0.05** | -105.65 | **7.38** |
| | pf-GMM (Pred. Term Only) | -0.18 | -150.18 | -20.93 |
| | pc-GMM | -0.50 | -130.41 | -15.51 |
| | sup-GMM | -0.66 | -30.81 | -15.48 |
| | 2-Step | -0.70 | 5.79 | -26.23 |
| | Class Cond. GMM | -5.92 | **14.87** | -20.66 |
| 6 | pf-GMM | **-0.05** | -92.20 | **7.49** |
| | pf-GMM (Pred. Term Only) | -0.10 | -151.88 | -20.02 |
| | pc-GMM | -0.51 | -129.12 | -15.48 |
| | sup-GMM | -0.63 | 1.76 | -9.23 |
| | 2-Step | -0.69 | 16.63 | -13.77 |
| | Class Cond. GMM | -2.25 | **28.47** | 6.96 |
| 8 | pf-GMM | **-0.05** | -57.23 | **8.37** |
| | pf-GMM (Pred. Term Only) | -0.09 | -150.80 | -18.66 |
| | pc-GMM | -0.50 | -130.46 | -15.55 |
| | sup-GMM | -0.56 | 8.80 | 0.89 |
| | 2-Step | -0.68 | 23.28 | -10.75 |
| | Class Cond. GMM | -2.07 | **44.05** | 8.16 |

Figure 11: Discriminative and generative performance of all the methods for different dataset complexities (varying $D_\mathrm{R}$) and different model complexities (varying $K$) on simulated data (complex).

# E  Experimental and Baseline Details

## E.1  Discussion: Baselines

Our central purpose for choosing baselines was to span the space of outcomes with generative and discriminative models, and then position our model in that space.

1. 2-Step: This baseline uses a fully generative model for X, in the sense that it learns parameters without using any information about Y. This learned model is then used for predicting Y.

2. sup-GMM: This baseline uses a fully generative model in a semi-supervised setting, i.e. it uses information about both X and Y when fitting its parameters.

3. pf-GMM: This model explicitly trades off generative and discriminative quality as described in paper.

4. pc-GMM: This baseline also trades off generative and discriminative quality in a different way.

5. pf-GMM (Pred. Term Only): This baseline has the same model as pf-GMM but the objective is purely discriminative, i.e. $\log f(Y|X)$. This objective is hard to optimize, though. Therefore, we use this method to demonstrate the predictive-generative trade-off but not for the rest of the tasks.

6. Random Forest: This is a purely discriminative baseline. We expect it to be similar in performance to "pf-GMM (Pred. Term Only)". Random Forests is not a method of interest, though. We want a generative model that is able to cluster the data, and RF clearly cannot achieve it.

7. Sparse Clustering: This baseline is another 2-step method commonly used in the clustering literature to ignore certain features in the data. It has no generative model.

### E.2   Training Details for Baselines

1. 2-Step: We used the scikit-learn (Pedregosa et al., 2011) implementations of GMM for the generative model and logistic regression for the discriminative model. The GMM is trained using the Expectation-Maximization algorithm.

2. Random Forest: We used the scikit-learn (Pedregosa et al., 2011) implementation of Random Forest.

3. pc-GMM: We used the Adam optimizer (Kingma & Ba, 2017) to optimize the pc-GMM objective. We used the PyTorch (Paszke et al., 2019) library for automatic differentiation. We search for the best value of $\lambda$ and learning rate parameters using a grid search.

4. sup-GMM: We used the same implementation as pf-GMM since the special case of $p = 1$ is equivalent to sup-GMM.

5. Sparse Clustering: We used the implementation from the pysparcl (tsurumeso, 2024) library. We used 25 permutations to search for the best value of the $w$ parameter as proposed in (Witten & Tibshirani).

6. pf-GMM (Pred. Term Only): We used the same implementation as pc-GMM with a very high value of $\lambda$ to ignore the generative term.

### E.3   Simulated datasets for the GMM experiments

#### E.3.1   Synthetic (simple)

We follow the following generative process to generate a collection of datasets:

$$\boldsymbol{p} = [p_0, \cdots, p_{K_D-1}]$$

$$[\mathbf{X}_1, y] \sim \sum_{k=0}^{K_D-1} \theta_k^{\mathrm{R}} [\mathcal{N}(K \cdot \mathbf{6}, I), \mathbf{Bern}(p_k)]$$

$$\mathbf{X}_2 \sim \sum_{k=0}^{K_D-1} \theta_k^{\mathrm{Ir}} \mathcal{N}(K \cdot \mathbf{6}, I)$$

$$\mathbf{X} = [\mathbf{X}_1; \mathbf{X}_2]; \quad \mathbf{X}_1 \in \mathbb{R}^{D_{\mathrm{R}}}; \quad \mathbf{X}_2 \in \mathbb{R}^{D_{\mathrm{Ir}}}$$

where $\mathbf{Bern}(a)$ is a Bernoulli distribution with parameter $a$, and $\mathbf{6}$ is a constant vector. This process ensures that the target $y$ is correlated with the first $D_{\mathrm{R}}$ dimensions of the input $\mathbf{X}$ but not the last $D_{\mathrm{Ir}}$ dimensions. The gap between successive components is fixed to 6 so that the clusters are well separated and so that the relevant and irrelevant dimensions have equivalent signal-to-noise ratio a-priori.

In our experiments, we set:

$$K_D = 3$$
$$D_{\mathrm{R}} \in \{20, 30, 40, 50, 80\}$$
$$D_{\mathrm{R}} + D_{\mathrm{Ir}} = 100$$
$$\theta^{\mathrm{R}} = normalize(0.5 + [0, \dots, K-1])$$
$$\theta^{\mathrm{Ir}} = normalize(1 + [0, \dots, K-1])$$

where $normalize(\mathbf{x}) = \mathbf{x}/\|\mathbf{x}\|_1$ makes sure the vectors are valid probability distributions.

This gives us a knob to tweak: $D_{\mathrm{R}}$ (number of relevant dimensions). It is common for the input to only have a few relevant dimensions. Therefore, we vary $D_{\mathrm{R}}$ in { 10, 20, 30, 40, 50, 80, 100 }.

### E.3.2   Synthetic (complex)

The generative process for the complex dataset is simple, except for the following differences:

$$K_D = 4$$
$$p = [0.05, 0.95, 0.05, 0.95]$$
$$D_{\mathrm{R}} \in \{20, 30, 40, 50, 80\}$$

### E.4   Simulated datasets for the HMM experiments

$$[\mathbf{X}_1, y] \sim \sum_{k=0}^{K-1} \theta_k^{\mathrm{R}} [\mathcal{N}(K \cdot \mathbf{6}, I), \mathbf{Bern}(p_k)]$$

$$\mathbf{X}_2 \sim \sum_{k=0}^{K-1} \theta_k^{\mathrm{Ir}} \mathcal{N}(K \cdot \mathbf{6}K, I)$$

$$\mathbf{X} = [\mathbf{X}_1; \mathbf{X}_2]; \quad \mathbf{X}_1 \in \mathbb{R}^{D_{\mathrm{R}}}; \quad \mathbf{X}_2 \in \mathbb{R}^{D_{\mathrm{Ir}}}$$

In our experiments, we set:

$$K = 4$$
$$p = [0.05, 0.95, 0.05, 0.95]$$
$$D_{\mathrm{R}} \in \{2, 3, 4, 5, 8, 12, 20\}$$
$$D_{\mathrm{R}} + D_{\mathrm{Ir}} = 20$$
$$\theta^{\mathrm{R}} = \theta^{\mathrm{Ir}} = [.1, .2, .3, .4]$$
$$A^{\mathrm{R}} \leftarrow normalize_{row}\left(.1 + \begin{bmatrix} -z_1- \\ \dots \\ -z_K- \end{bmatrix} + \mathbf{I}_K\right)$$
$$A^{\mathrm{Ir}} \leftarrow normalize_{row}\left(.01 + \begin{bmatrix} -z_1'- \\ \dots \\ -z_K'- \end{bmatrix} + \mathbf{I}_K\right)$$
$$z_i, z_i' \sim \mathbf{Cat}(\mathbf{1}/K), z_i, z_i' \in \mathbb{R}^K$$

where $normalize_{row}(M)$ applies $normalize$ to each row of matrix $M$. Also, the numbers .1 and .01 are arbitrary choices to differentiate $A^{\mathrm{R}}$ and $A^{\mathrm{Ir}}$, and the simulation would work with different values too.

### E.5   HIV Experiments

Therapy for HIV involves administering cocktails of antiretrovirals from five classes namely, Non-nucleoside Reverse Transcriptase Inhibitors (nnRTIs), Nucleoside Reverse Transcriptase Inhibitors (nRTIs), Protease Inhibitors (PIs), Fusion Inhibitors (FIs), and Integrase Inhibitors (IIs) to bring the viral load below detection limits ($\leq 40$ copies/ml). We study 53 236 patients with HIV from the EuResist Integrated Database. Each person has a time-series of an average length of 16 steps where a time step is approximately 4 months between consecutive treatments. Our task is to predict whether a treatment will bring the viral load below detection limits in the next time-step. Each input contains 267 features including CD4+counts, genetic mutations, treatments in terms of drug classes, and lab results. Though it is common to have many genetic mutations, only a few of these may be relevant for inducing drug resistance thus increasing the viral load.

Table 4: Comparison of the baselines with pf-GMM.

|  |  | Class-specific-GMM | GMM | pf-GMM |
|---|---|---|---|---|
| | $Y$ AUC | **1.0** | 0.6 | **1.0** |
| Dataset 1 (Simple) | $\log P(\mathbf{X})$ | **-10.0** | -16.9 | -104.6 |
| | $\log P(\mathbf{X}_R)$ | -19.6 | -27.0 | **10.2** |
| | $Y$ AUC | 0.7 | 0.5 | **1.0** |
| Dataset 2 (Complex) | $\log P(\mathbf{X})$ | **10.4** | 5.6 | -106.3 |
| | $\log P(\mathbf{X}_R)$ | -21.4 | -26.8 | **10.1** |

# F    Discussion: Naive Baseline of Fitting Class-Specific GMMs Fails

A simple alternative to the prediction-focused approach might simply try to learn a separate GMM for each class label. In this case, the prediction $Y$ for an input $\mathbf{X}$ can be predicted based on the probability of the input under each class-specific GMM. First, we note that this approach which splits up the data by class does not allow us to identify structures that have commonalities between the classes. Second, this approach will often produce poor predictions, as we demonstrate below. In this section, we demonstrate using two datasets that a baseline that uses a separate GMM for each $Y$ label is insufficient in identifying and modeling the relevant dimensions in the dataset.

## F.1    Setup

**'Simple' Dataset with a unimodal class-specific density along relevant dimensions.**    This dataset contains 100 dimensions, where the first 20 dimensions are relevant and generated using a 3-cluster GMM, and the rest of the dimensions are irrelevant and generated using a separate 3-cluster GMM. Therefore, the data density along relevant dimensions has three modes, where the middle mode corresponds to the $Y = 1$ class and the other two modes correspond to the $Y = 0$ class (see the first row of Figure 12 for the histogram of a relevant dimension of this dataset). The data density along irrelevant dimensions also has three modes but they do not differ in the proportion of $Y$ labels (see the second row of Figure 12 for the histogram of an irrelevant dimension). We will show below that this demonstrates the case where the class-specific GMM can give good predictions without learning a good density for the relevant dimensions (i.e. for the wrong reasons).

**'Complex' Dataset with multimodal class-specific densities along relevant dimensions.**    This dataset contains 100 dimensions, where the first 20 dimensions are relevant and generated using a 4-cluster GMM, and the rest of the dimensions are irrelevant and generated using a separate 4-cluster GMM. In contrast to the previous dataset, the data density along relevant dimensions has four modes, where the first and the third modes correspond to $Y = 0$ class and the second and the fourth modes correspond to $Y = 1$ class (see the third row of Figure 12 for the histogram of a relevant dimension of this dataset). The data density along irrelevant dimensions also has four modes but again they do not differ in the proportion of $Y$ labels (see the fourth row of Figure 12 for the histogram of an irrelevant dimension). We will show below that the class-specific GMM will fail to give good predictions *and* learn a poor density for the relevant dimensions in this case.

**Evaluation**    We evaluate the models on $Y$ AUC, $\log P(\mathbf{X})$, and $\log P(\mathbf{X}_R)$. $Y$ AUC measures how well the model clusters can discriminate the label and $\log P(\mathbf{X})$ measures how accurate the model density is for *all* the dimensions of $\mathbf{X}$. Finally, $\log P(\mathbf{X}_R)$ measures how accurately the models estimate the density of *only* the relevant dimensions (known to us because we generate the data).

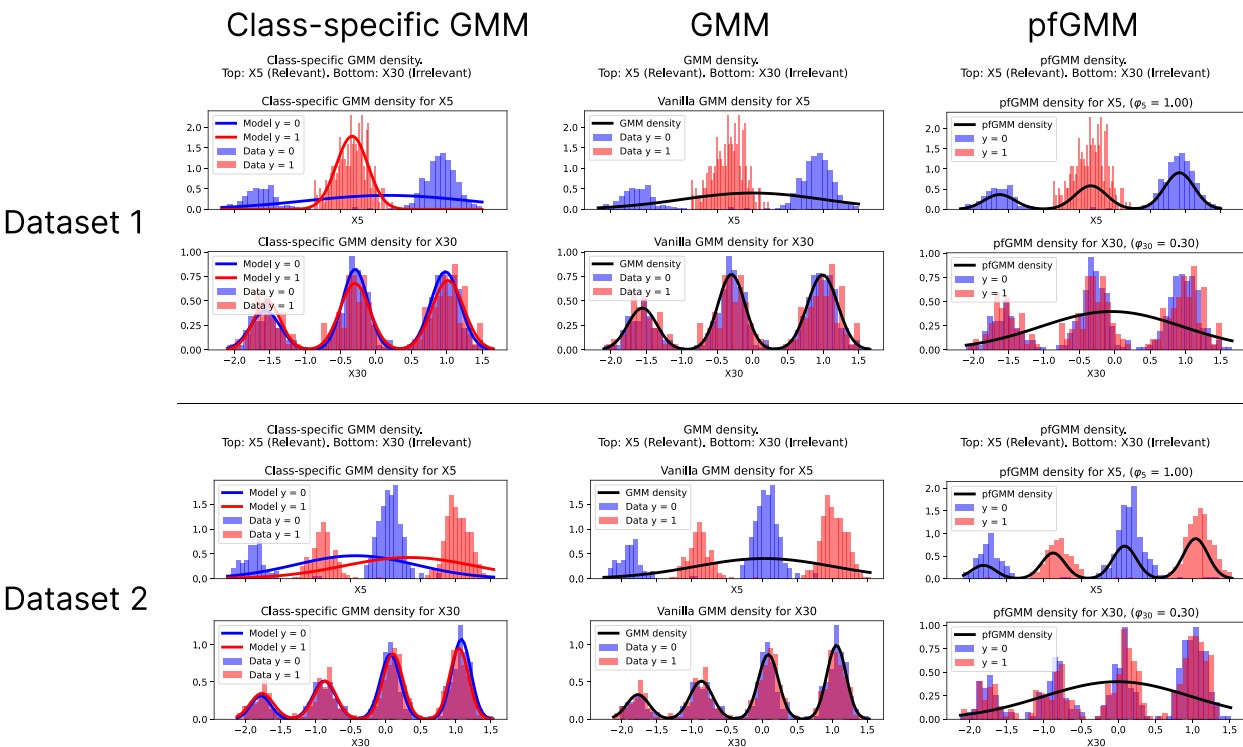

Figure 12: Comparison of the class-specific-GMM baseline (left column) on both datasets reveals that it is similar to a GMM (middle column) in ignoring relevant dimensions. In contrast, a pf-GMM (right column) can ignore irrelevant dimensions even when they outnumber the relevant dimensions.

### F.2 Results

**Class-specific GMM cannot filter out irrelevant dimensions**   In the left column of Figure 12, we see that the class-specific GMM baseline clusters on the irrelevant dimensions, and fails to learn a good density for the relevant dimensions, $\log P(\mathbf{X}_{\mathrm{R}})$, for both datasets. The densities learned by it are similar to densities learned by the simple GMM baseline (middle column of Figure 12). In contrast, the pf-GMM model learns a good density for the relevant dimensions, gives good predictions, and ignores the irrelevant dimensions (see right column of Figure 12 and Table 4).

**Class-specific GMM can be predictive of $Y$ without clustering the relevant dimensions well** In Table 4, we see that the class-specific GMM can give good predictions on the dataset where the relevant dimensions have a unimodal class-specific density (i.e. the first dataset in the Setup). This is because it can model the $P(\mathbf{X}_{\mathrm{R}}|Y = 1)$ density well and can tell when the data comes from it (as seen in the top row, left column of Figure 12). However, this good predictive performance is not indicative of a good density model for the relevant dimensions, as discussed in the previous paragraph. Furthermore, when the relevant dimensions have all multimodal class-specific densities (i.e. in the second dataset), the class-specific GMM fails to give good predictions *and* learns a poor density for the relevant dimensions.

**Class-specific GMM is qualitatively similar to a simple GMM**   The behavior of the class-specific GMM baseline is qualitatively similar to that of a simple GMM: when faced with many irrelevant dimensions, both these methods just cluster on the irrelevant dimensions (see Figure 12, middle column). Both these methods achieve a much better overall density $\log P(\mathbf{X})$ than the pf-GMM model because they learn a good density for the irrelevant dimensions and the irrelevant dimensions dominate the overall density. Furthermore, both these methods learn a poor density for the relevant dimensions, $\log P(\mathbf{X}_{\mathrm{R}})$. The class-specific GMM still outperforms the simple GMM quantitatively (Table 4) since it uses twice as many parameters. In contrast,

the pf-GMM model is inherently different in its behavior, as it learns to ignore irrelevant dimensions, learns a good density for the relevant dimensions by clustering on them, and uses these clusters to make predictions.

