# OpenReview forum: "Task-Relevant Feature Selection with Prediction Focused Mixture Models"
_TMLR — Accepted by TMLR_

### Review · Reviewer_jmg6 · 2024-01-08

**Summary Of Contributions:**

This work introduces a class of probabilistic generative models that account for a response variable in addition to the input/predictor covariates. Specifically, it develops a "prediction-focused Gaussian mixture model" (pf-GMM) and "prediction-focused hidden Markov model" (pf-HMM) for iid and temporal data, respectively. The models can be fit using conventional optimization approaches, and they enable both fitting the input data distribution (maximizing the data log-likelihood) and predicting the response variable, thereby bridging the capabilities of existing generative and discriminative models.

**Audience:**

Yes

**Broader Impact Concerns:**

No concerns.

**Claims And Evidence:**

Yes

**Requested Changes:**

Several requests for clarification and changes to the experiments are described above.

**Strengths And Weaknesses:**

## Strengths

The idea of task-focused generative modeling seems interesting, and I'm not aware of any established methods for fitting generative models like GMMs or HMMs that account for a response variables. The proposed approach seems reasonable, and it can be fit efficiently using standard optimization procedures (like EM). The paper is mostly well written and easy to follow.

I have some comments below the experiments, but I appreciated that the authors investigated the run-time of each method, the sensitivity to the $K$ parameter, and used a synthetic dataset to study the effect of a couple aspects of the data-generating process (the results shown in Figure 4).

## Weaknesses

One main weakness is the motivation for this work: in what scenarios is it important to fit a probabilistic model of the data distribution *and* predict the response variable? The experiments don't discuss any uses of the generative modeling capability. It's doubtful that the GMM-like models used here accurately estimate the data distribution - for example, I wouldn't expect that they could generate realistic samples, and the paper doesn't explore anything of this nature (for this we would probably require some kind of deep generative model). With respect to predicting the response variable, the proposed approach underperforms a simple random forest. Finally, the authors at times lean into the interpretability aspects of this approach, even writing "because inspection is our ultimate goal, simply creating a predictive model is not sufficient" (page 1). However, the experiments seem to barely explore the inspectability of these models, and the authors overlook a range of other approaches that create interpretable predictive models (e.g., feature selection or post-hoc feature attribution, both of which can accommodate more expressive nonlinear models).

Another weakness has to do with the presentation: the assumptions about the data distribution in Section 3 seem disconnected from the modeling approach in Section 4. It takes roughly 3 pages to explain the data setting, but the proposed methods correspond to a far simpler model of the data-generating process. Also, a couple questions and concerns about Section 3: 1) the reference to "Supplement Section 3.4" looks like a typo, 2) assuming that all data sources are independent seems unrealistic for most datasets (this perhaps doesn't matter because Section 4 is so different from Section 3), 3) it was unclear whether we would assume existing feature groupings (again, this bit is ignored in Section 4), and 4) I couldn't understand what the authors mean by "our approach extends to any exponential-family distribution" on page 4 (again, because Section 3 is not obviously related to the proposed method).

Next, the proposed models are interesting, but I wonder if the authors considered a simpler way to integrate a response variable into a GMM or HMM. For the iid case with a GMM, instead of the pf-GMM (modeling $Y$ jointly with a subset of covariates), or modeling $Y$ as a covariate (the "supervised GMM" baseline), or predicting $Y$ based on the mixture components, what about fitting two separate GMMs for data with $Y = 0$ and $Y = 1$? This simple probabilistic model would make it easy to calculate the posterior $P(Y \mid X)$, and fitting it is just as easy as a standard GMM. If I understand correctly, this baseline was not used in the experiments, and I wonder if the authors could add it.

Some other thoughts on the method:

- In Eq. 13, it seems like one weakness in the model is that features with more complex, non-Gaussian distributions may be pushed towards having $S = 1$ to benefit from the expressiveness of the mixture distribution. Is this intuition correct, and do the experiments address this point?
- Eq. 14 appears to have some typos, it omits $X_S$ and $Y$'s conditioning on $Z$

About the experiments:

- Figure 4B/E show the landscape of discriminative and generative modeling performance for each model, and that pf-GMM offers a reasonable trade-off. However, like I mentioned above it's not clear why the $\log p(X)$ term is important.
- Given the emphasis on interpretability and inspection of the model, I expected to see some of this in Section 8. The only bit I see is the very last paragraph, where the authors mention two mutations that are consistent with prior work. Is there any kind of more robust evaluation the authors can offer? For example, would it be more appropriate to share a full list of the features deemed relevant, and perhaps analyze the stability of this set as parameters like $K$ and $p$ are varied?
- The experiment are run using just two real dataset, and I'm not sure that's enough to draw the conclusions discussed in Section 8. The model seems fast to train, so perhaps the authors could add some additional datasets? And perhaps some more similar to the HIV dataset, because the Swiss Bank Notes one seems very small (6 features, 200 examples).

---

> ### Author Response · Authors · 2024-02-26
> **Response 1**
>
> Dear reviewer,
>
> Thank you for your thoughtful comments. We have incorporated your typographical recommendations and address your requested changes below. As requested by you, we also introduce a new real-world Psychiatry dataset.
>
>
> **Motivation for the work**
> There are several applications where it is important to learn a probabilistic model of the data distribution and subsequently use this to predict a response variable. For instance, the review by [Chen et al 2020] highlights the importance of probabilistic models for understanding the differences in survival time of patients with melanoma with different demographics for aiding in better planning of disease management. [Halpern et al 2016] describe the importance of probabilistic models for identifying groups of clinical conditions that can subsequently used to predict disease subtypes and clinical tags. Our approach would enable identifying only those conditions that are relevant for a characterizing a particular subtype and a particular clinical tag. Works such as [Prabhakaran et al 2019] and [Prabhakaran et al 2012] also show how probabilistic models can be used to learn clusters of similar chemical compounds or proteins in HIV treatment which can subsequently be used to predict response patterns for patients who receive those treatments.
>
>
> **Psychiatry dataset**
> 1. Description of the dataset (Also added to Section 7.1, page 14 of the paper):
>     We analyzed a dataset of electronic health records from two large academic medical centers and their affiliated community hospitals and outpatient clinics. We selected a cohort of 13,608 patients with at least one diagnosis of major depressive disorder ICD-10 diagnosis code during psychiatric care between 2017 and 2022. We remapped the ICD-10 codes to 423 CCSR codes, resulting in 423 features for a patient: each feature with a value of 1 if the diagnosis code was observed, and 0 otherwise. An independent standard normal observation noise with a standard deviation of 0.1 was added. Our prediction task is to ascertain a substance use disorder (SUD) outcome for the patient, and the goal of the clustering step is to discover subgroups relevant to the prediction task. This task would benefit from prediction-focused modeling because, without the prediction task, the clusters would correspond to common comorbidities of MDD. However, we wish to first discover pain-related comorbidities.
>
> 2. Motivation for prediction-focused modeling
>     The goal for pf modeling in this case is to *jointly* (a) filter out the disease codes irrelevant to a patient's substance use disorder (SUD), and (b) identify patient cohorts (i.e. clusters) based on the relevant codes. While it is clear that patient groups cannot be explained using only very few (e.g. 2) clusters if *all* the diseases are considered, it is reasonable to expect 2 patient groups when only considering SUD-related codes. This is evidenced by the following: although a 2-component GMM and 500-component GMM achieve AUCs of 0.6 and 0.75 at predicting the outcome, a 2-component pf-GMM achieves an AUC of 0.85. Furthermore, a 2-component GMM with the top 20 features from pf-GMM achieves an AUC of 0.93, suggesting that prediction-focused modeling is indeed picking out the relevant features.
> 3. Features considered relevant by pf-GMM:
>     We compare the top features of pf-GMM against the top features of Random Forest classifier, which acts as a consistency check that the features we are picking out are predictive of the outcome.
>
>     *Top relevant dimensions w.r.t pf-GMM*:
>     Trauma- and stressor-related disorders
>     Tobacco-related disorders
>     Alcohol-related disorders
>     Other specified and unspecified mood disorders
>     Bipolar and related disorders
>     Miscellaneous mental and behavioral disorders/conditions
>     Mental and substance use disorders in remission
>     Other specified substance-related disorders
>     Opioid-related disorders
>     Suicidal ideation/attempt/intentional self-harm
>
>     *Top relevant features w.r.t RF (for reference)*:
>     Trauma- and stressor-related disorders
>     Tobacco-related disorders
>     Other specified and unspecified mood disorders
>     Alcohol-related disorders
>     Bipolar and related disorders
>     Miscellaneous mental and behavioral disorders/conditions
>     Mental and substance use disorders in remission
>     Schizophrenia spectrum and other psychotic disorders
>     Other specified substance-related disorders
>     Suicidal ideation/attempt/intentional self-harm

---

> ### Author Response · Authors · 2024-02-26
> **Response Part 2**
>
> **Proposed baseline with different GMMs for each $Y=c$ label**
> Thank you for proposing the baseline, and we do expect it to be competitive with the baselines. However, we do not think it would be fair to compare against this baseline because (a) it uses twice as many components and parameters as the rest of the models, and (b) it scales the number of parameters with the number of classes for Y, which is something we wish to avoid, especially in the case of high-dimensional X values. If you need further clarification about this point, we are happy to provide it.
>
> **The utility of the generative model** (i.e. pf-GMM) is similar to that of a GMM: we can use it to simulate *relevant* dimensions of the data or use it for downstream applications that a GMM could be used for.
>
> **Ability to model relevant dimensions**:
> For a 100-dimensional dataset (the first 30 relevant dimensions generated using a 4-component GMM, and the rest 70 irrelevant dimensions generated using another 4-component GMM), we simulated data from one relevant (index 1) and one irrelevant dimension (index 51).
> We show below that the density of a relevant dimension (dimension 1, left plot) is indeed well modeled when we use pf-GMM. This is not the case for an irrelevant dimension (dimension 51, right plot).
> Image Link: https://hackmd.io/_uploads/HJqo2PF3T.png
>
> $\log p(X)$ is to quantify the generative quality of the methods.
>
> **Downstream tasks**:
> The learned mixture model can enable many downstream use cases. The prediction of a response variable is a straightforward application which we exploit to learn relevant clusters. But we can also use the probability density from the mixture to impute missing (relevant) features [Di Zio et al], and use the uncertainty provided by the learned density from the mixture model to better plan downstream decision-making tasks [Sharma et al].
>
>
> **Stability of features selected for different K/p values**:
> For the dataset described above, we list the top 10 features as we vary K and p. We see that the group of features identified as relevant is stable.
>
> For K in [2, 3, 4] and p=1e-2:
>
> |    |   K=2 |   K=3 |   K=4 |
> |----|-------|-------|-------|
> |  0 |    13 |    13 |    13 |
> |  1 |    27 |    27 |    27 |
> |  2 |    20 |    20 |    20 |
> |  3 |     1 |     1 |     1 |
> |  4 |    28 |    28 |    28 |
> |  5 |     8 |     8 |     8 |
> |  6 |     4 |     4 |     4 |
> |  7 |    26 |    26 |    26 |
> |  8 |    14 |    14 |    14 |
> |  9 |    19 |    19 |    19 |
>
> For p in [1e-3, 5e-3, 1e-2] and K=4:
>
> |    |   p=1e-3 |   p=5e-3 |   p=1e-2 |
> |----|-----------|-----------|----------|
> |  0 |        13 |        13 |       13 |
> |  1 |        27 |        27 |       27 |
> |  2 |        20 |        20 |       20 |
> |  3 |         1 |         1 |        1 |
> |  4 |        28 |        28 |       28 |
> |  5 |         8 |         8 |        8 |
> |  6 |         4 |         4 |        4 |
> |  7 |        26 |        26 |       26 |
> |  8 |        14 |        14 |       14 |
> |  9 |        19 |        19 |       19 |
>
>
>
> **Feature-selection**
> pf-GMM is primarily intended to be a generative mixture model. It's not just a classic feature selection method because we can also do many downstream tasks using the generative model that we obtain---the feature selection of pf-GMM is an added benefit that helps us interpret which features were indeed clustered. Nevertheless, we have compared the feature relevances of pf-GMM with importances obtained from Random Forests, and can add these to the supplement if you deem that useful.

---

> ### Author Response · Authors · 2024-02-26
> **Response Part 3**
>
> **Apparent discrepancy between Sections 3 and 4**:
> The proposed model is intentionally simpler than the assumed data distribution, i.e. the model's distribution cannot realize the true data distribution because we expect this model to be useful in *underspecified* setting. This setting is common in the real world: in the context of the psychiatry data, we do not need the model to be as complex as the true data to identify *useful* insights about the parts we care about.
>
> **Comparison to Random Forest**
> As addressed in the paper (Figure 4, page 13 and Section E.1, page 42), the Random Forest baseline is presented to only act as a discriminative "oracle"/upper bound. We do expect it to do better at predicting the response variable, but importantly, it does *not* provide the generative model that we require.
>
> **Independent data sources assumption**
> We agree that the independence of data sources appears to be a strong assumption. However, it is not that uncommon for this assumption to be approximately true in many datasets as we motivated in the HIV subtyping example in Section 3.3 (page 4). We also provide evidence from the psychiatry dataset in the following heatmap of pairwise correlations of the features. In the heatmap, we see that there are indeed several "groups" of features, in which features are strongly correlated to other features in the group, and are uncorrelated with most other features outside the group:
> Image Link: https://hackmd.io/_uploads/By_zhtY2T.png
>
>
> **Exponential-family distributions**
> By "our approach extends to any exponential-family distribution", we mean that our of prediction-focused mixture model's EM-style training algorithm will work with any emission distribution as long as it belongs to the exponential family. pf-GMM is just a case when the emission distribution is the Gaussian distribution. This is because only sufficient statistics are required for the M-step (like in a simple mixture model).

---

> > ### Comment · Reviewer_jmg6 · 2024-02-29
> > **Response**
> >
> > Thanks for responding to all the feedback. I'll quickly address a couple points from the response.
> >
> > **Motivation for probabilistic modeling.** Thanks for pointing out some papers that make use of probabilistic models for various medical applications. Many of the cases you describe sound like they leverage simplicity or interpretability in their models, not necessarily an accurate modeling of the generative distribution? I won't make this a big issue, but I'll maintain that the paper doesn't demonstrate concrete use cases for the generative/probabilistic aspect of the proposed approach; benchmarking methods via their $\log p(x)$ is reasonable, but it's not clear when this is useful or that it must be performed jointly with the response variable prediction.
> >
> > **Psychiatry dataset.** Thanks for adding the new dataset. It's encouraging but not surprising that the pf-GMM outperforms a purely generative GMM at classification. However, the pf-GMM's accuracy has a huge drop relative to a simple random forest (0.85 vs 0.99), so that brings back the concern that this isn't exactly the best of both worlds - the pf-GMM is in some cases a mediocre classifier. And thanks for highlighting the top features, this seems like a helpful addition that was missing in the original submission.
> >
> > **Simple baseline.** This baseline sounds pretty simple to add? If doubling the number of parameters is really an issue, it would presumably show up via either slow fitting or less efficient sample complexity. And it's ultimately not that many parameters. I don't agree that it's an unfair comparison, and given the simplicity it would be odd to omit.
> >
> > **Feature selection and attribution.** Adding the comparison to random forest feature importance scores could be helpful. It would be more helpful to add a section to the related work explaining that model inspection, which the introduction states is your ultimate goal, is achievable by other means that have been thoroughly explored in the literature. If there are certain capabilities that are unique to your approach that aren't enabled by existing feature selection/attribution methods, that would be a good place to point them out.
> >
> > **Section 3 vs Section 4.** Thanks for the clarification. I guess the rationale makes sense, but the paper dedicates a lot of space to describing a data setting that isn't really relevant in the end. It's up to you whether to adjust that, but it's confusing as a reader.

---

> > > ### Author Response · Authors · 2024-03-08
> > > **Response Part 1**
> > >
> > > Dear reviewer,
> > >
> > > Thank you for the feedback.
> > >
> > > **Regarding concrete downstream use cases of pf-GMM's mixture model:**
> > >
> > > In the context of our paper, interpretation of clusters for exploration and science-oriented tasks is a common use case of the mixture models in the literature [e.g., Kuyuk et al., Zhuang et al.]. Our work is motivated by this, indeed funded by a grant in close collaboration with psychiatrists to identify subtypes of major depression.  We are happy to add more about the clinical interpretations of the clusters we find and the value of those in hypothesis generation.
> > >
> > > However, we do have a logistical problem: Our access to the data is currently not working (unexpectedly, as of last week).  Our partner hospital is working to fix that, but it may take some time.  Our current resubmission addresses everything except adding this.  Please advise on whether we should wait and add some clinical interpretation of the clusters.
> > >
> > > More generally, we note that there are many examples of downstream use cases in the literature for mixture models (all of these are potential uses of the mixture model obtained from pf-GMM). Our prior work [Sharma et al] uses the density $p(X)$ from a mixture model to plan an elevator scheduling decision-making task. [Silva and Clayton] use the gaussian mixture model to impute missing values in a dataset of lateritic Nickel deposits. [El Attar et al] use gaussian mixture models for anomaly detection.
> > >
> > > We have also added these to the paper to motivate the generative model better (Section 2, Page 2)
> > >
> > > **Why jointly predict the response variable.**
> > >
> > > The joint prediction of the response variable has the benefit of picking out the relevant dimensions, even if they are outnumbered by the irrelevant dimensions.  For example, in our psychiatry collaboration, we expect that most of the codes in a patient's history are irrelevant to their depression (in our paper, only about 1/8th of the 423 features were sufficient to form predictive clusters).  However, we do not know which are relevant, a priori. For example, cancer codes may be relevant because having the cancer often affects a person's mental health---but it would be hard to manually curate which conditions play a part in a person's depression and which do not. The response variable, which represents a psychiatry outcome, helps us pick out depression-related codes.
> > >
> > > **Psychiatry dataset**
> > >
> > > Thank you for the feedback about the top features. Regarding the discriminative performance, we would like to point out that while pf-GMM may not be the best of both worlds, the discriminative task helps with the more exploration/science-oriented tasks that we highlighted above.  If our goal was purely discriminative, we would not attempt to do so with a relatively small generative model.  Here, training this relatively small generative model to have the best discriminative can result in learning a generative model that is valuable for e.g. understanding subtypes of depression.

---

> > > > ### Author Response · Authors · 2024-03-08
> > > > **Response Part 2**
> > > >
> > > > **Simple baseline**
> > > >
> > > > Fair point.  As we noted above, we currently do not have access to the hospital server.  However, we did run the baseline on synthetic examples below.  We have added this discussion to Section F in the Supplement (Page 46) and cited that discussion in the Baselines subsection in the main paper (Section 7, page 14).
> > > >
> > > > Please let us know whether the below addresses your concerns or whether we should also run on the real data.  (We also want to emphasize that prior work [Hughes et al 2017] has also analyzed this baseline for a different version of prediction-focused tasks and found it has major limitations.)
> > > >
> > > > **Synthetic Example**
> > > >
> > > > Setup:
> > > >
> > > > Dataset 1: 100 dimensions (first 20 are relevant and generated using a 3-cluster GMM, and the rest are irrelevant and generated using a separate 3-cluster GMM). This demonstrates the case where the baseline can give good predictions *without* clustering the relevant dimensions (i.e. for the wrong reasons).
> > > >
> > > > Dataset 2: 100 dimensions (first 20 are relevant and generated using a 4-cluster GMM, and the rest are irrelevant and generated using a separate 4-cluster GMM). This is a case where the baseline just fails because there is no way it can cluster the irrelevant dimensions *and* learn a good density for any class.
> > > >
> > > > Results (also presented in the figure):
> > > > 1. For dataset 1, the baseline has :
> > > >
> > > > Good y-predictions since it is able to model the $X_{rel}|Y=1$ density well and can tell when the data comes from it. However, this is purely accidental, as shown in dataset 2.
> > > >
> > > > Good overall $p(X)$ since it is modeling irrelevant dimensions well, which contribute to $p(x)$ a lot more.
> > > >
> > > > Bad $p(X_{rel})$ since it is not really modeling the relevant dimensions well.
> > > >
> > > > 2. For dataset 2, the baseline has :
> > > >
> > > > Bad y-predictions since now there is no possibility of modeling $X_{rel}|Y=1$ or $X_{rel}|Y=0$ well unless the model ignores the irrelevant dimensions.
> > > >
> > > > Good $p(X)$ and bad $p(X_{rel})$ for similar reasons as dataset 1
> > > >
> > > > Thus, the behavior of the baseline is similar to that of a simple GMM: when faced with many irrelevant dimensions, both these methods just cluster on the irrelevant dimensions. In contrast, pf-GMM consistently ignores the irrelevant dimensions in both cases and correctly clusters the relevant dimensions.
> > > >
> > > > Figure link: https://hackmd.io/_uploads/SkCzwsDa6.png
> > > >
> > > > ---
> > > >
> > > >
> > > > **Feature selection and attribution**
> > > >
> > > > Thank you for the suggestion. We have added a subsection in the Related Works section discussing the Feature selection and attribution methods (Section 2, Page 3).
> > > >
> > > > ---
> > > >
> > > > **Section 3 vs Section 4**
> > > >
> > > > Thank you for the feedback. We have provided an introduction to Section 3 (Data Setting) clarifying that the data distribution is presented to emphasize the complexity of the data compared to the model distribution, and that the model is designed to work in this "underspecified" setting.

---

> > > > > ### Comment · Reviewer_jmg6 · 2024-03-19
> > > > > **Response**
> > > > >
> > > > > **Concrete use cases for the mixture model:** sorry to hear about your situation with the psychiatry data. It would be nice to include clinical interpretation of the clusters, but if you can't do so for this data perhaps you can do it for the HIV dataset? It would also be nice to explicitly connect this notion of mixture modeling (seemingly a form of supervised clustering) with generative modeling, because the connection isn't very clear in the paper (i.e., what this has to do with scoring well on the $\log p(x)$ metric).
> > > > >
> > > > > **Feature selection and attribution:** this is a good addition.
> > > > >
> > > > > **Simple baseline:** your synthetic example seems reasonable for understanding the strengths and weaknesses of your approach compared to this baseline. But why not run the method on real datasets, at least the HIV or Swiss bank notes data if you still can't access the psychiatry dataset? It seems reasonable to include this comparison given the various baselines in figures 4, 5 and 6. Based on table 3, the trade-off between discriminative and generative performance seems favorable compared to the pf-GMM.

---

> > > > > > ### Comment · Editors_In_Chief · 2024-03-30
> > > > > >
> > > > > > Hi authors, just pinging to check if you saw this comment/question by a reviewer? We're at about the time to make a decision on this paper, and your attention here would be appreciated.
> > > > > >
> > > > > > Thanks,
> > > > > > Gautam

---

> > > > > > > ### Author Response · Authors · 2024-04-01
> > > > > > > **Response to Editors In Chief**
> > > > > > >
> > > > > > > Dear Gautam and Reviewers,
> > > > > > >
> > > > > > > Sorry for the delay. We recently regained access to the data and re-running experiments for the dataset and the baseline reviewer jmg619 recommended. We will get back soon with an updated manuscript with all changes incorporated.
> > > > > > >
> > > > > > > Thank you for being patient with us,
> > > > > > > The authors

---

> > > > > > ### Author Response · Authors · 2024-04-05
> > > > > > **Response to reviewer**
> > > > > >
> > > > > > Dear reviewer,
> > > > > >
> > > > > > Thank you for pointing us to the class conditional GMM baseline. It revealed more subtle things about how the pfGMM model behaves in different data settings. Our inspection also revealed when and the ways in which the class conditional baseline can appear to have good performance but still fail to cluster in a desired way, even when pfGMM stays consistent in its behavior. Since some of these discussions are specific to this baseline, we put it in the appendix (section F), but we can move it to the main paper if you think that would be useful.
> > > > > >
> > > > > > *Clinical interpretations of the clusters*
> > > > > > We have added clinical interpretations of the clusters obtained using our pfGMM method (for both Psychiatry and HIV datasets), providing insights into how the clusters can be used to understand different subgroups that are relevant to the outcome (Figure 17, page 17 ; section 8, page 18)
> > > > > >
> > > > > > *Baseline*
> > > > > > We have run the baseline on all the datasets and updated our plots. As noted above, we have also updated our results (section 8, page 18) to discuss this baseline in the main paper and add a deeper discussion in the supplement.
> > > > > >
> > > > > >
> > > > > > *Link to generative modeling*
> > > > > > We would emphasize that pfGMM is indeed a generative model (because it is a model-based clustering model). We have also added the model density plots for the synthetic and real datasets (Figure 5 page 14, Figure 17 page 17) so that it is conveyed in the paper explicitly. If we misunderstood your concern, we are happy to engage again.
> > > > > >
> > > > > > Thank you a lot for being patient with us while we made these changes.
> > > > > >
> > > > > > Best,
> > > > > > Authors

---

> > > > > > > ### Comment · Reviewer_jmg6 · 2024-04-05
> > > > > > > **Response**
> > > > > > >
> > > > > > > Thanks for your updates, I'm satisfied with the new experiments involving the class-specific GMM baseline. It seems like that method mostly performs well, especially on the generative term $\log p(x)$, but that the proposed approach sometimes performs better (e.g., the discriminative accuracy for the HIV dataset). The more thorough discussion in the appendix is a good idea. Thanks again for all the changes made during the review process.

---

### Review · Reviewer_ZjeR · 2024-01-22

**Summary Of Contributions:**

This papers describes a generative model for observed data consisting on a set of explaining attributes, which can be either relevant or irrelevant for prediction, and a set of observed labels. The labels can be categorical or sequential data. The authors describe alternative models to infer the relevant information from the observed data on which and expectation maximization based algorithm using variational inference is proposed. The method proposed is validated in synthetic data, and on two real-world datasets, showing improved results with respect to alternative methods.

**Audience:**

Yes

**Claims And Evidence:**

Yes

**Requested Changes:**

- Revise the writing of the paper. E.g., in the beginning of Section 3.1, in the notation, there is a bold capital letter referred to as a random variable, which is confusing.

- Notation [N] is confusing. I would suggest using {1,...,N}, which is understood better and does not use that much writing space.

- Explain why there is a need to use VI to estimate the model parameters, instead of maximum likelihood.

- Include the synthetic and add real-world experiments for the HMM model in the main part of the manuscript.

**Strengths And Weaknesses:**

Strengths:

        - Intuitive model definition.

        - Well written paper.

        - Interesting problem addressed.

Weaknesses:

        It seems that the authors describe a generative model, but do not carry out inference in such a generative. They suggest an alternative model on which they do carry out inference. That is not very well explained. In particular, why it is not possible to use the generating model to carry out inference and why there is a need to describe another model for inference. That is counterintuitive. Is not this inference model another way of writing the generative model assumed first?

        It is not clear VI is used for inference. It seems that you have an expression for the joint probability of the observed variables in the inference models. Thus, one can simply maximize the likelihood without having to approximate it via VI.

        The analysis in Section 6.2 seems to be valid only for data without correlations since the covariance matrices in Eq. (22) and (23) are diagonal. It seems also limited to only two Gaussians.

        No analysis similar to than of Section 6.2 is carried out for temporal data.

        It seems that no real-world experiments are carried out with the HMM model. There are only synthetic experiments in the appendix.

---

> ### Author Response · Authors · 2024-02-26
> **Response**
>
> Dear reviewer,
>
> Thank you for your thoughtful comments. We address the changes requested by you below.
>
> 1. **Why use VI instead of MLE?** We use VI to learn parameters instead of directly using MLE because the likelihood objective (Equation 7) is intractable to compute in practice: it scales exponentially with dimension $O(K.2^D)$ due to the sum over the $S$ variable. We have added this point right after the equation.
> 2. We have incorporated your notation-related recommendations ({1,..,N} instead of [N] ; clarified notation at the beginning of Section 3.1).
> 3. **HIV Results**: We would like to point you to Figure 5C for the HMM model results with a real-world HIV dataset.
>
> We are eager to hear if that answers your questions and addresses the changes you had in mind.

---

### Review · Reviewer_kVkz · 2024-02-12

**Summary Of Contributions:**

In this paper, the authors aim to fit a mixture model (such as GMM or HMM) that uses a user-specified number of components to a given data distribution. As is typical in this literature, the authors focus on maximizing the data likelihood for a given prediction quality of the labels.

For this setting, the authors introduce prediction-focused mixture models, including one case for GMM and another case for HMM. After specifying these models, the authors went on to formalize the questions, such as dataset characteristics for which a misspelled model can still select the relevant components.

Through a detailed case study, the authors derive a series of conclusions from the analysis, including:

- The prediction-focused GMM can identify relevant structure by changing the switch prior $p$ (this is a variable to distinguish relevant and irrelevant dimensions).
- The prediction-focused GMM identifies mixtures even when there are many irrelevant dimensions, and it is more robust to high SNR.

Lastly, the authors use synthetic data and an HIV patient database and another Swiss ban notes dataset to evaluate their proposed models.

In summary, the paper developed a model-based feature selection method for mixture models and hidden Markov models.

**Audience:**

Yes

**Claims And Evidence:**

Yes

**Requested Changes:**

- See weaknesses above. In particular, addressing both points under Weaknesses.
- Also, there is a section called "Formalizing the Question" in Section 6.3---I'm not sure if these are the focused questions of the paper or just follow-up questions after the discussion in 6.2. It would be helpful to clarify.
- In the introduction, even though you listed a few items as contributions, they don't really talk about the actual empirical analysis you showed. This can be clarified.
- The appendix is very long; it can benefit from having an organizational paragraph detailing the list of contents to follow. Likewise, in the main text, you can include a preview of the main sections before starting the related work paragraph.

**Strengths And Weaknesses:**

S1) The paper gave a detailed study of using prediction-based GMM and HMM models on both synthetic datasets and two real datasets. The paper is clearly written, providing insights into when the feature selection will work and when it will fail.

S2) The appendix includes a detailed derivation of the technical details, including the ELBO-based inference procedure.

W1) I wish there are more discussions about the choice of the datasets---they seem reasonable to me but I think more background concerning their motivations and their particular application to this type of methods would be helpful.

W2) I am also wondering how does the proposed method compare with standard EM type methods---also, it seems that the training algorithms for fitting the mixture models are not sufficiently discussed in the current manuscript.

---

> ### Author Response · Authors · 2024-02-26
> **Response**
>
> Dear reviewer,
>
> Thank you for your thoughtful comments! We address the changes requested by you below:
>
> 1. We have updated descriptions of datasets to better motivate prediction-focused modeling (Section 7.1, pages 13-14). We have also added a sentence highlighting empirical results in the contributions list (Section 1, page 2).
> 2. **EM-style methods**: Here, we would like to highlight the difference between EM-style methods (e.g. Gaussian mixture models) and EM-style inference. In our baselines, we indeed have two variants of Gaussian mixture models (2-Step and sup-GMM), and both these methods are trained using the EM algorithm. Furthermore, we use EM-style inference for training pf-GMM (as noted below Equation 19 in the paper).
> 3. **Training details**: We have added training details of the baselines in the supplement (Section E.2, page 42).
> 4. **Organizational paragraphs**: We have added the organization paragraphs in the main paper and in the supplement as you had recommended (pages 2 and 18). Thank you for the suggestion!
>
> We are eager to hear if that answers your questions and addresses the changes you had in mind.

---

### Author Response · Authors · 2024-02-26
**Author Response**

Dear Reviewers,

We appreciate your valuable feedback on our paper regarding prediction-focused modeling. We have carefully reviewed your comments and responded individually to each of you with clarifications.

We have uploaded a revised version of the paper in PDF format, which incorporates all the requested changes, including writing-related modifications and clarifications. We have also added a new real-world dataset to the paper.

We hope that the changes we've made have addressed your recommendations. If you have any further questions or need additional clarification, we are happy to provide answers.

Thank you again for taking the time to review our work.

Best regards,
Authors

---

### Decision · Action_Editor_Tyzk · 2024-04-07

**Recommendation:** Accept as is

**Comment:**

The paper provides a detailed study of task-relevant feature selection, with two prediction-aware mixture models proposed. These models are analysed both theoretically and empirically, with both synthetic and real-world data. There were some initial concerns on the motivation and comparison to relevant baselines; following the author discussion and revisions, all three reviewers found the paper's execution to be convincing, and were in favour of acceptance. The AE concurs with this view.

**Audience:**

The paper provides a mixture model suitable for setting where there is a clear downstream prediction task. This is motivated by (and demonstrated) applications in electronic health record analysis. This is a problem of interest to many ML researchers, and thus the findings of the paper are likely to have an interested audience. (One reviewer noted that the specific subject of the paper may be a bit niche, which is acknowledged as a possibility, but we have confidence that a reasonable subset of the TMLR audience would find the work of interest.)

**Claims And Evidence:**

The primary claims of the paper are:
- one may design probabilistic graphical models (pf-GMM, pf-HMM) that are effective at identifying relevant features for downstream prediction tasks, with well-defined training and inference strategies (Section 4, 5)
- the proposed models have a rigorous theoretical underpinning, and in particular, can automatically infer relevant structure in settings where prior methods fail (Section 6)
- the proposed models perform well on both synthetic and real-world data, both in terms of quality and ease of tuning, compared to existing baselines (Section 7, 8)

After revisions following the author discussion, which included new experiments for the latter, all three reviewers found the claims to be sufficiently supported.